# Ki-67 shapes the nucleolus by anchoring chromatin via its amphiphilic properties

Daja Schichler [1,2,6], Yuki Hayashi [1,6], Letitia Fernandez[1,4], Mariam Chupanova[1], Alberto Hernandez-Armendariz [1,2,4,5], Beate Neumann [3] & Sara Cuylen-Haering [1✉]

## Abstract

**The nucleolus, a membrane-less organelle essential for ribosome biogenesis, adopts variable shapes across cell types and in response to environmental conditions, yet the mechanisms regulating its morphology and functional implications remain unclear. Using a high-throughput screen, we identify the proliferation marker Ki-67 as a central regulator of nucleolar shape. Ki-67 localises to the chromatin-nucleolus interface, where its depletion induces nucleolar rounding and reduces chromatin enrichment both at the nucleolar rim and within internal invaginations. This effect is driven by Ki-67's amphiphilic properties conferred by two distinct affinity domains separated by a spacer. Given that chromatin loss is a common feature of rounded nucleoli in our screen, and acute chromatin digestion also induces rounding, we propose that the chromatin environment in and around the nucleolus plays a key role in determining nucleolar shape. Our study elucidates a novel Ki-67-mediated chromatin anchoring mechanism, tightly linking nucleolar shape to genome organisation and expanding our understanding of condensate morphology.**

**Keywords** Nucleolus; Heterochromatin; Ki-67; Biomolecular condensates; Amphiphilic properties
**Subject Categories** Chromatin, Transcription & Genomics; Organelles

## Introduction

Membrane-less compartments are essential for cellular organisation and participate in diverse cellular functions. Unlike membrane-bound organelles, these compartments are dynamic and often exhibit liquid-like properties, allowing them to respond quickly to stimuli. In both cells and in vitro, membrane-less compartments typically adopt spherical shapes (Feric et al, 2016; Yao and Rosen, 2024; Bussi et al, 2023), driven by surface tension to minimise their surface area (Lyon et al, 2021). However, some membrane-less compartments, such as nucleoli or nuclear speckles, commonly display irregular shapes (Sleeman and Trinkle-Mulcahy, 2014). This irregularity may be influenced by several factors, such as their internal molecular properties, their interactions with the surrounding environment, or their viscoelastic properties. The precise molecular mechanisms and factors that determine the shape of specific membrane-less compartments remain, however, largely unknown.

Nucleoli are the largest membrane-less organelles in the nucleus, coordinating the process of ribosome biogenesis. To achieve this, they are organised into three distinct, nested subcompartments: the innermost fibrillar centre (FC), surrounded by the dense fibrillar component (DFC), and the outermost granular component (GC), which envelops dozens of FC-DFC subcompartments. This layered subcompartmentalisation is critical for spatially segregating different steps of ribosome biogenesis (Boisvert et al, 2007), ensuring that transcription, processing, and assembly occur efficiently within specialised regions of the nucleolus.

In addition, nucleoli play a key role in the cellular stress response. Under stress conditions, such as DNA damage or heat shock, nucleoli undergo dynamic changes to adapt to the altered cellular environment. For instance, DNA damage can lead to the inhibition of ribosome biogenesis through the suppression of RNA polymerase I activity. This triggers the rounding of nucleoli and the reorganisation of their nested structure, resulting in the appearance of FC-DFC subcompartments on the surface of the GC region, a structure known as nucleolar caps (Shav-Tal et al, 2005).

Another important function of nucleoli is their role in chromatin organisation. Nucleoli not only organise the ribosomal RNA (rRNA) gene clusters, but they are also surrounded by a rim of dense heterochromatin (Padeken and Heun, 2014). These heterochromatic regions, known as nucleolar-associated domains (NADs) (Németh et al, 2010), are enriched in telomeres and other repetitive DNA sequences and regions with low gene density (van Koningsbruggen et al, 2010). Therefore, nucleoli are considered to be one of the central structures for the organisation of inactive heterochromatin, contributing to the spatial arrangement and regulation of these genomic regions.

Despite the liquid-like properties of nucleoli (Mitrea et al, 2018; Stochaj and Weber, 2020; Lafontaine et al, 2021), their shape is highly variable

[1]Cell Biology and Biophysics Unit, European Molecular Biology Laboratory (EMBL), Heidelberg, Germany. [2]Collaboration for Joint PhD Degree between EMBL and Heidelberg University, Faculty of Biosciences, Heidelberg, Germany. [3]Advanced Light Microscopy Facility, European Molecular Biology Laboratory (EMBL), Heidelberg, Germany. [4]Present address: Max Planck Institute of Molecular Cell Biology and Genetics, Dresden, Germany. [5]Present address: Cluster of Excellence Physics of Life, TU Dresden, Dresden, Germany. [6]These authors contributed equally: Daja Schichler, Yuki Hayashi. ✉E-mail: sara.cuylen-haering@embl.de

between cell types (Thul et al, 2017). While some cell types exhibit nearly spherical nucleoli, others display highly irregular morphologies, characterised by alternating concave and convex regions. Such shape diversity is not only a feature of normal cellular differentiation but also a hallmark of certain diseases. In conditions like viral infections (Callé et al, 2008; Salvetti and Greco, 2014), neurodegeneration (Hetman and Slomnicki, 2019), and cancer (Zink et al, 2004), changes in nucleolar shape, size, and number have been observed and are often used as clinical diagnostic markers (Helpap, 1988; Derenzini et al, 2009; Stamatopoulou et al, 2018), suggesting that nucleolar shape is tightly regulated and responsive to the cellular environment. These alterations in shape may reflect underlying changes in ribosome biogenesis, stress signalling, or genome stability. However, the molecular mechanisms driving these shape changes remain poorly elucidated.

Here, we report the results of an RNA interference (RNAi) screen to identify proteins involved in regulating nucleolar shape. This screen identified several candidate proteins, including the cell proliferation marker Ki-67, that induce nucleolar rounding upon their depletion. In addition to inducing nucleolar rounding, Ki-67 depletion led to the loss of chromatin from both the nucleolar rim and its interior, the latter reflecting inward-folded chromatin invaginations. Quantitative live-cell imaging and protein engineering experiments revealed that Ki-67 localises at the chromatin-nucleolar interface, dependent on its two distinct affinity domains. This interaction promotes nucleolar chromatin enrichment and chromatin invaginations, maintaining an irregular nucleolar shape. Notably, loss of chromatin enrichment in the nucleolus was observed in most of the candidate proteins from our RNAi screen that induce nucleolar rounding. In most cases, Ki-67 expression levels were reduced, suggesting that Ki-67 may play an essential role in regulating chromatin distribution in the nucleolus. Since acute chromatin digestion was sufficient to induce nucleolar rounding, the chromatin network is most likely a central determinant of nucleolar morphology. Our findings highlight the role of chromatin in regulating nucleolar shape, with Ki-67-dependent chromatin anchoring facilitated by its amphiphilic structure emerging as one of the central molecular mechanisms underlying the irregular morphology of nucleoli.

## Results

### Identification of nucleolar shape regulators

To identify molecular factors regulating nucleolar shape, we conducted a live-cell imaging-based RNAi screen in a HeLa cell line expressing NPM1-EGFP as a nucleolus marker and H2B-mCherry as a chromatin marker. We targeted 614 genes using 2–3 individual siRNAs per gene, focusing on a library enriched for nucleolus-associated proteins (see Dataset EV1), including proteins that regulate nucleolar number (Farley-Barnes et al, 2018) and are associated with phase separation ability (Becher et al, 2018; Sridharan et al, 2019). Live cells were imaged using an automated wide-field microscope, and their cell-cycle stages were classified through supervised machine learning. We extracted interphase cells and quantified nucleolar aspect ratio as a proxy for shape (Fig. 1A). While nucleoli in cells treated with a non-targeting control siRNA were typically irregularly shaped, our screen identified many candidates that induce nucleolar rounding (Fig. 1B,C).

Remarkably, 15 of the 20 highest-scoring candidates were components of the small subunit (SSU) processome (Fig. 1C,D),

crucial for ribosome biogenesis (Singh et al, 2021). Other notable candidates included XRCC5 and XRCC6, a heterodimer involved in the DNA double-strand break repair. Additionally, depletion of RPABC1 (encoded by the POLR2E gene), a subunit common to all three RNA polymerases, the Bromodomain and WD repeat-containing protein 1 (encoded by BRWD1) (Filippakopoulos et al, 2012), and the well-known proliferation marker Ki-67 (Sun and Kaufman, 2018; Remnant et al, 2021) (encoded by MKI67) also induced nucleolar rounding.

Since knockdown of SSU processome components impairs rRNA processing (Tafforeau et al, 2013), the observed nucleolar rounding might be a consequence of stalled ribosome biogenesis and subsequent reorganisation (Boulon et al, 2010). To test this hypothesis, we performed a secondary screen using a dual-colour cell line labelling inner and outer nucleolar subcompartments (Appendix Fig. S1). As expected, knockdown of the RNA polymerase subunit RPABC1 led to nucleolar reorganisation with a characteristic cap-like structure. Similarly, knockdown of multiple SSU processome components induced nucleolar cap formation, suggesting that rounding is likely an indirect consequence of stalled ribosome biogenesis. While XRCC5 and XRCC6 depletion did not induce caps under our conditions, they likely affect nucleoli structure indirectly through genomic instability (Oberdoerffer and Sinclair, 2007). BRWD1 is thought to function as a transcription regulator that is involved in regulating cell morphology (Bai et al, 2011). Therefore, its effect might be indirect, through the regulation of other genes. Ki-67, on the other hand, localises to nucleoli during interphase (Stenström et al, 2020) and is dispensable for efficient rDNA transcription and pre-rRNA processing (Sobecki et al, 2016), suggesting that nucleolar rounding must occur via a different pathway. Notably, Ki-67 has been proposed to function as a surfactant on the chromosome surface during mitosis (Cuylen et al, 2016). Although it remains unclear whether its surfactant function persists in interphase (Cuylen-Haering et al, 2020; Hernandez-Armendariz et al, 2024), Ki-67's reported localisation and suggested function make it a compelling candidate for regulating nucleolar morphology during interphase. Based on this rationale, we investigated Ki-67's role in shaping nucleoli.

To avoid artefacts caused by overexpression of nucleolar marker proteins or fluorescent protein tags, we performed label-free holotomographic imaging and used supervised machine learning to segment nucleoli (preprint: Shabanov et al, 2021) (Appendix Fig. S2A). To determine whether nucleolar rounding was specific to Ki-67 depletion rather than a non-specific siRNA effect, we transfected two different Ki-67 siRNAs, both of which deplete Ki-67 to background levels (Appendix Fig. S2B,C), into wild-type HeLa cells and a cell line resistant to one of the siRNAs due to Cas9-engineered synonymous mutations of the target region (Cuylen et al, 2016). In wild-type cells, Ki-67 depletion led to rounder nucleoli, while control cells had irregularly shaped nucleoli (Fig. 1E, top row, and 1F, left graph). In the resistant cell line, only the siRNA targeting the non-mutated region caused nucleolar rounding, whereas the siRNA to which the cells were resistant resulted in irregular nucleoli, resembling the control (Fig. 1E, bottom row and 1F, right graph). The nucleolar rounding phenotype was also observed in unlabelled Ki-67 knockout (KO) cells (Appendix Fig. S2D,E). In summary, these results confirmed that the nucleolar rounding induced by transfection with Ki-67-targeting siRNAs was a specific effect of Ki-67 knockdown.

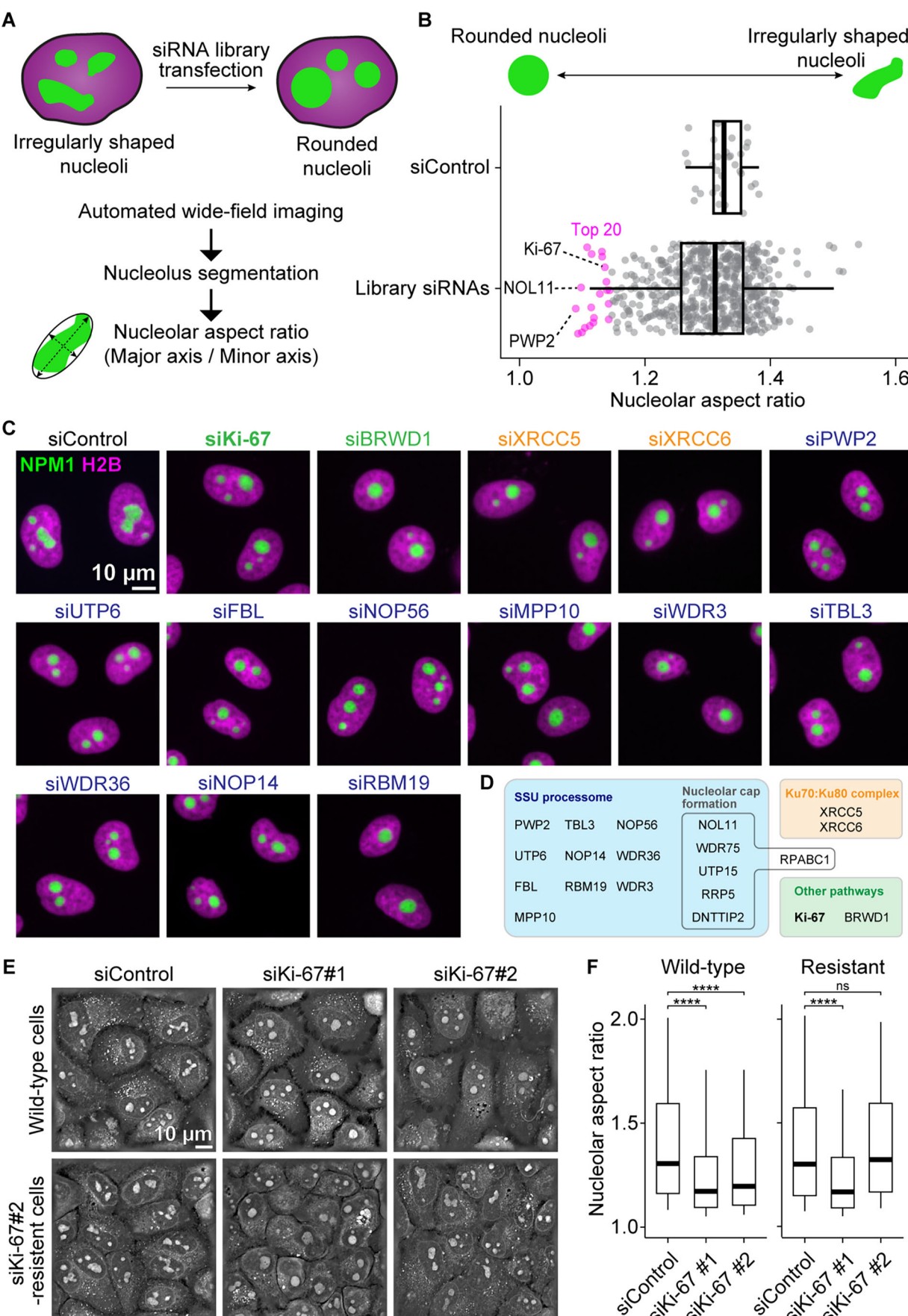

**Figure 1.  An RNAi screen for genes regulating nucleolar shape identifies Ki-67.**

(**A**) Experimental design of a live-cell RNAi screen to identify genes regulating nucleolar shape. HeLa cells expressing H2B-mCherry (magenta) and NPM1-EGFP (green) were seeded in multiwell plates with siRNAs targeting one gene per well and imaged after 72 h. Nucleoli were segmented using an automated image analysis pipeline, and their shape was quantified by their aspect ratio after fitting an ellipse around each nucleolus. (**B**) Outcome of the RNAi screen targeting 614 genes. Individual data points in a non-targeting control siRNA (siControl) indicate median nucleolar aspect ratio per well, providing a baseline for comparison. Data points from the library siRNAs indicate median values per gene based on 2 or 3 different siRNAs. Magenta-coloured data points highlight the top 20 genes inducing nucleolar rounding. Boxplots display the median (centre line), interquartile range (box), and whiskers extending to the 10th and 90th percentiles. (**C, D**) Raw data from the screen showing the effect of knocking down the top 20 genes identified in (**B**). Chromatin is labelled by H2B-mCherry (magenta), and nucleoli are marked by NPM1-EGFP (green). The molecular function of the top 20 genes has been manually annotated and classified into different colour-coded groups. (**E**) Label-free holotomographic live-cell imaging of HeLa cells 72 h after transfection. Wild-type HeLa cells (top panels) or HeLa cells resistant to siRNA Ki-67#2 (bottom panels) were transfected with siControl or two different Ki-67 siRNAs (siKi-67#1 and siKi-67#2). A single z-slice is shown. (**F**) Quantification of nucleolar aspect ratio in holotomographic images. The aspect ratio of nucleoli was measured for wild-type HeLa cells (left plots) or HeLa cells resistant to siKi-67#2 (right plots). Boxplots display the median (centre line), interquartile range (box), and whiskers extending to the 10th and 90th percentiles. Statistical comparisons were performed against the siControl sample for each cell line. For wild-type cells, siKi-67 #1, $P = 4.76 \times 10^{-15}$; siKi-67 #2, $P = 1.25 \times 10^{-8}$. For resistant cells, siKi-67 #1, $P = 9.09 \times 10^{-20}$; siKi-67 #2, $P = 0.628$. For (**F**), $n = 527$ nucleoli (siControl, wild-type), 432 nucleoli (siKi-67#1, wild-type), 444 nucleoli (siKi-67#2, wild-type), $n = 707$ nucleoli (siControl, resistant), 662 nucleoli (siKi-67#1, resistant), 724 nucleoli (siKi-67#2, resistant), 2 biological replicates. Statistical tests were performed with the Kruskal–Wallis test followed by Dunn's test, ns (not significant) $P > 0.05$, ****$P < 0.0001$. See also Appendix Figs. S1 and S2. Source data are available online for this figure.

## Ki-67 localises to the interface of chromatin and the nucleolus

To understand how Ki-67 regulates nucleolar shape, we revisited its localisation within nucleoli. Several studies have described its localisation to the region surrounding nucleoli, known as the nucleolar periphery or rim (Verheijen et al, 1989; Sobecki et al, 2016; Stenström et al, 2020). In addition, early studies using immunoelectron microscopy or immunofluorescence have demonstrated that Ki-67 also localises to the inner regions of nucleoli with limited overlap with the other nucleolar subcompartments (Kill, 1996; Verheijen et al, 1989; Cheutin et al, 2003). However, its exact localisation within these inner nucleolar regions remained unclear.

To investigate the precise localisation of Ki-67 in nucleoli, we generated a HeLa cell line endogenously tagged with EGFP at the Ki-67 N-terminus and stably expressing the nucleolar protein NPM1 tagged with a SNAP-tag. We then performed live-cell Airyscan super-resolution imaging of Ki-67 and NPM1, along with DNA staining. Although Ki-67 was enriched at the nucleolar rim, it was also clearly detectable within nucleoli (Fig. 2A–D). This internal Ki-67 signal did not overlap with the canonical nucleolar subcompartments (Fig. EV1), consistent with previous protein localisation studies (Kill, 1996; Verheijen et al, 1989). Since we also observed a significant chromatin signal in the nucleolar interior (Fig. 2A,E), we tested for colocalisation with Ki-67. Line profile analyses along intranucleolar chromatin foci indicated that Ki-67 signal peaks were positioned between DNA and NPM1 signal peaks (Fig. 2F, line 1). Similarly, at the nucleolar rim, the Ki-67 signal was most intense between the NPM1 and DNA signal peaks (Fig. 2F, line 2). These results suggest that Ki-67 localises at the boundary between the nucleolus (GC compartment) and chromatin.

To quantitatively confirm this observation across many cells, we segmented intranucleolar chromatin and divided the nucleolus into 50 concentric rings around the chromatin foci's centre points (Fig. 2G). We then measured Ki-67, DNA, and NPM1 signal intensities in each ring, enabling us to generate spatial intensity profiles across multiple cells (Fig. 2H). Adjacent to the chromatin foci (rings 1–15), Ki-67 displayed a broader peak compared to DNA, suggesting its localisation on the surface of intranucleolar chromatin foci. At the nucleolar boundary (rings 15–50), Ki-67 intensity peaked concomitantly with a sharp decrease in

NPM1 signal, preceding the peak in DNA signal. These results corroborate our line profile analyses, reinforcing the conclusion that Ki-67 preferentially localises at the boundary between chromatin and the nucleolus, both at the nucleolar rim and along intranucleolar chromatin foci.

To further investigate the nature of these Ki-67-coated intranucleolar chromatin structures, we performed 3D imaging on histone H2B throughout the nucleus. We found that these chromatin foci originated from chromatin invaginating into nucleoli (Fig. EV2). Notably, these chromatin invaginations were distinct from rDNA (Fig. EV3A), and instead they colocalised with heterochromatin markers (Fig. EV3B–E), consistent with previous reports that Ki-67 associates with heterochromatin (Sobecki et al, 2016; van Schaik et al, 2022). These results indicate that the chromatin signal within nucleoli represents invaginated chromatin fibres that extend deep into the nucleolar interior and are consistently coated with Ki-67.

## Ki-67 anchors chromatin into the nucleolar interior

Given that Ki-67 localises at the chromatin-nucleolus boundary, we next investigated whether Ki-67 plays a role in regulating the chromatin environment within and around nucleoli. To test this, we depleted Ki-67 using two different siRNAs in HeLa cells stably expressing histone H2B-mCherry as a chromatin marker and NPM1-EGFP as a nucleolus marker (Fig. 3A). Ki-67 depletion resulted in rounder nucleoli, as measured by nucleolar circularity (Fig. 3B), which accounts for both perimeter and area, while having minimal impact on nuclear morphology, nucleolar size, or nucleolar number (Appendix Fig. S3). Notably, we found that Ki-67 depletion not only decreased the H2B signal intensities in the nucleolar rim (Fig. 3C), in line with previous reports (Sobecki et al, 2016; van Schaik et al, 2022), but also led to reduced H2B signal intensities in the nucleolar interior (Fig. 3D), causing the nucleoli to appear as dark voids (Fig. 3A). The Ki-67 depletion phenotype, characterised by nucleolar rounding and loss of intranucleolar chromatin, was further confirmed in U2-OS and RPE-1 cells (Fig. EV4). Thus, our findings demonstrate the conserved role of Ki-67 in regulating nucleolar morphology and chromatin enrichment at the nucleolus.

Based on this finding and considering the localisation of chromatin and Ki-67 in the nucleolar interior between nucleolar subcompartments, we investigated whether chromatin tethering by

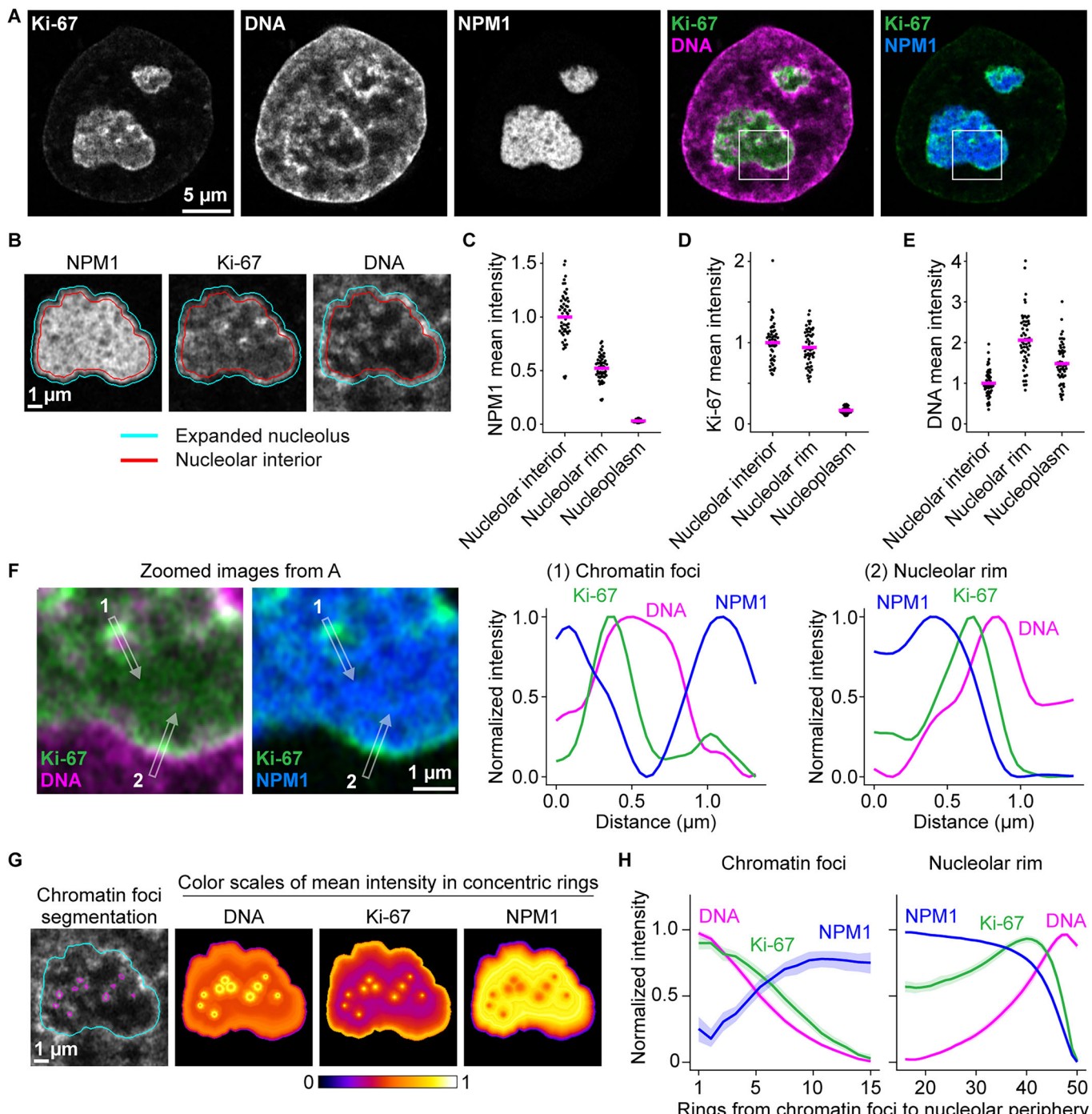

Ki-67 could support the separation of nucleolar subcompartments. Interestingly, Ki-67 depletion did not affect the internal organisation of the nucleolar subcompartments, as both the number and area of UBF foci remained constant (Fig. EV5), consistent with a report that Ki-67 is not required for ribosome biogenesis (Sobecki et al, 2016). These findings indicate that the separation of internal nucleolar subcompartments does not depend on Ki-67's chromatin recruitment to the nucleolar interior.

To further investigate Ki-67's role in anchoring chromatin to the nucleolus, we next tested whether Ki-67 overexpression causes

additional chromatin enrichment within and around nucleoli. We transiently overexpressed EGFP-Ki-67 in cells that already expressed endogenous EGFP-tagged Ki-67 and stably expressed SNAP-NPM1. After FACS-sorting to isolate cells with higher Ki-67 expression compared to the endogenous levels (Fig. 4A; Appendix Fig. S4A), we quantified the mean intensities of DNA and Ki-67 in the nucleolar interior and the rim (Fig. 4B,C). The DNA signal intensities in the nucleolar interior and the rim increased with EGFP-Ki-67 expression. At very high levels, Ki-67 recruited substantial amounts of chromatin to the nucleolar interior,

◀ **Figure 2. Ki-67 localises at the chromatin-nucleolus interface.**

(A) Representative live-cell Airyscan imaging of HeLa cells. Cells endogenously tagged with EGFP-Ki-67 and stably expressing SNAP-NPM1 were stained with SNAP-silicon rhodamine (SiR). DNA was stained with SPY555-DNA dye. (B–H) Shows different analyses performed on this dataset. (B) Example nucleolar segmentations based on the NPM1 signal from the same cell as in (A). The nucleolar segmentation was expanded (cyan) and shrunk (red; nucleolar interior) by 5 pixels, corresponding to approximately 200 nm. The nucleolar rim was defined by subtracting the shrunk segmentation from the expanded segmentation. (C–E) Quantification of NPM1 (C), Ki-67 (D), and DNA (E) signal intensities in the nucleolar interior, its rim and nucleoplasm. Mean intensities in each region were normalised to the mean intensity in the nucleolar interior. Bars represent mean values from 53 nuclei (from 3 biological replicates). (F) Signal distribution of NPM1, Ki-67, and DNA at the intranucleolar chromatin foci and the nucleolar periphery. Images show the regions indicated by white squares in (A). Line profile measurements were performed around chromatin foci (1) and the nucleolar rim (2). Signal intensities were normalised to the respective maximum and minimum fluorescence intensities. (G) Quantification of signal distributions of Ki-67, DNA, and NPM1 from the intranucleolar chromatin foci. Example segmentations of intranucleolar chromatin foci (magenta) and expanded nucleolus (cyan) are highlighted in the leftmost image from the same cell as in (A). The nucleolus was divided into 50 concentric rings from the centre of chromatin foci to the edge of the expanded nucleolus. The colour scale indicates relative signal intensity. (H) Quantification of the signal intensity in concentric rings. Mean intensities in each concentric ring were normalised by min–max scaling, applied independently to rings 1–15 (around the chromatin foci) and rings 16–50 (around the nucleolar rim). Line and shaded area indicate mean ± 95% confidence interval from 53 nuclei (from 3 biological replicates). See also Figs. EV1, 2 and 3. Source data are available online for this figure.

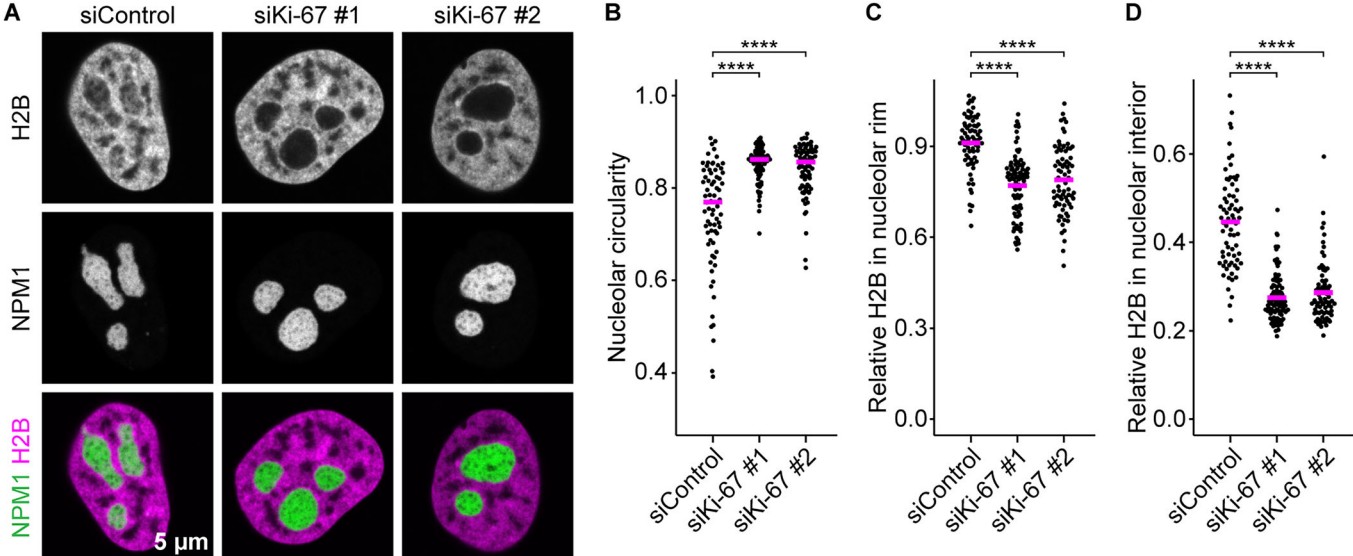

**Figure 3. Ki-67 depletion induces nucleolar rounding and chromatin removal from nucleoli.**

(A) Live-cell imaging of HeLa cells stably expressing H2B-mCherry and NPM1-EGFP. Cells were transfected with a non-targeting control siRNA (siControl) or two different Ki-67 siRNAs, followed by confocal imaging 72 h post-transfection. A single z-slice is shown. (B) Quantification of nucleolar shape from NPM1 signals. Nucleoli were segmented based on NPM1 signals to measure the median nucleolar aspect ratio per nucleus. Bars indicate median values. Statistical comparisons were performed against the siControl sample: siKi-67 #1, $P = 1.85 \times 10^{-9}$; siKi-67 #2, $P = 1.57 \times 10^{-7}$. (C, D) Quantification of chromatin enrichment in the nucleolar interior and its rim. Relative H2B signal intensities were calculated by dividing mean intensities in the nucleolar interior (C) or the nucleolar rim (D) by mean intensities in the nucleoplasm. Bars indicate mean values. Statistical comparisons were performed against the siControl sample. For (C), siKi-67 #1, $P = 2.97 \times 10^{-13}$; siKi-67 #2, $P = 1.41 \times 10^{-9}$. For (D), siKi-67 #1, $P = 4.23 \times 10^{-20}$; siKi-67 #2, $P = 1.41 \times 10^{-16}$. For (B–D), $n = 72$ nuclei (siControl), 87 nuclei (siKi-67#1), 75 nuclei (siKi-67#2), 2 biological replicates. Statistical tests were performed with Kruskal–Wallis test followed by Dunn's test, ****$P < 0.0001$. See also Figs. EV4 and EV5, Appendix Fig. S3. Source data are available online for this figure.

resulting in low chromatin densities in the rest of the nucleus (Fig. 4A). Analysis of DNA and NPM1 signals in concentric rings from the nucleolar centre to the periphery confirmed this observation (Appendix Fig. S4B,C). While DNA enriched at the rim in both mock and Ki-67 overexpression conditions, the normalised peak observed for Ki-67 overexpression was broader and extended into the nucleolar interior, possibly due to saturation at the rim. Ectopic expression of EGFP-T2A-Ki-67, which produces Ki-67 without a fluorescent protein tag, induced similar phenotypes, confirming that the observed nucleolar irregularity and chromatin enrichment are driven by Ki-67 overexpression rather than the tag (Appendix Fig. S4D). Together, these results

demonstrate that Ki-67 overexpression enhances chromatin recruitment to the nucleolar interior and the rim.

If chromatin incorporation contributes to nucleolar shape regulation, an increase in Ki-67 expression would be expected to induce higher nucleolar irregularity. Indeed, nucleolar circularity showed a clear negative correlation with Ki-67 levels (Fig. 4A,D), while nucleolar size and number remained largely unchanged (Appendix Fig. S4E,F). Notably, this effect was also observed in U2-OS and RPE-1 cells, indicating that the phenotype induced by Ki-67 overexpression is not cell-type specific (Fig. EV6). Taken together, these results demonstrate that Ki-67 recruits chromatin to both the nucleolar interior and the rim and modulates nucleolar shape in a dose-dependent manner.

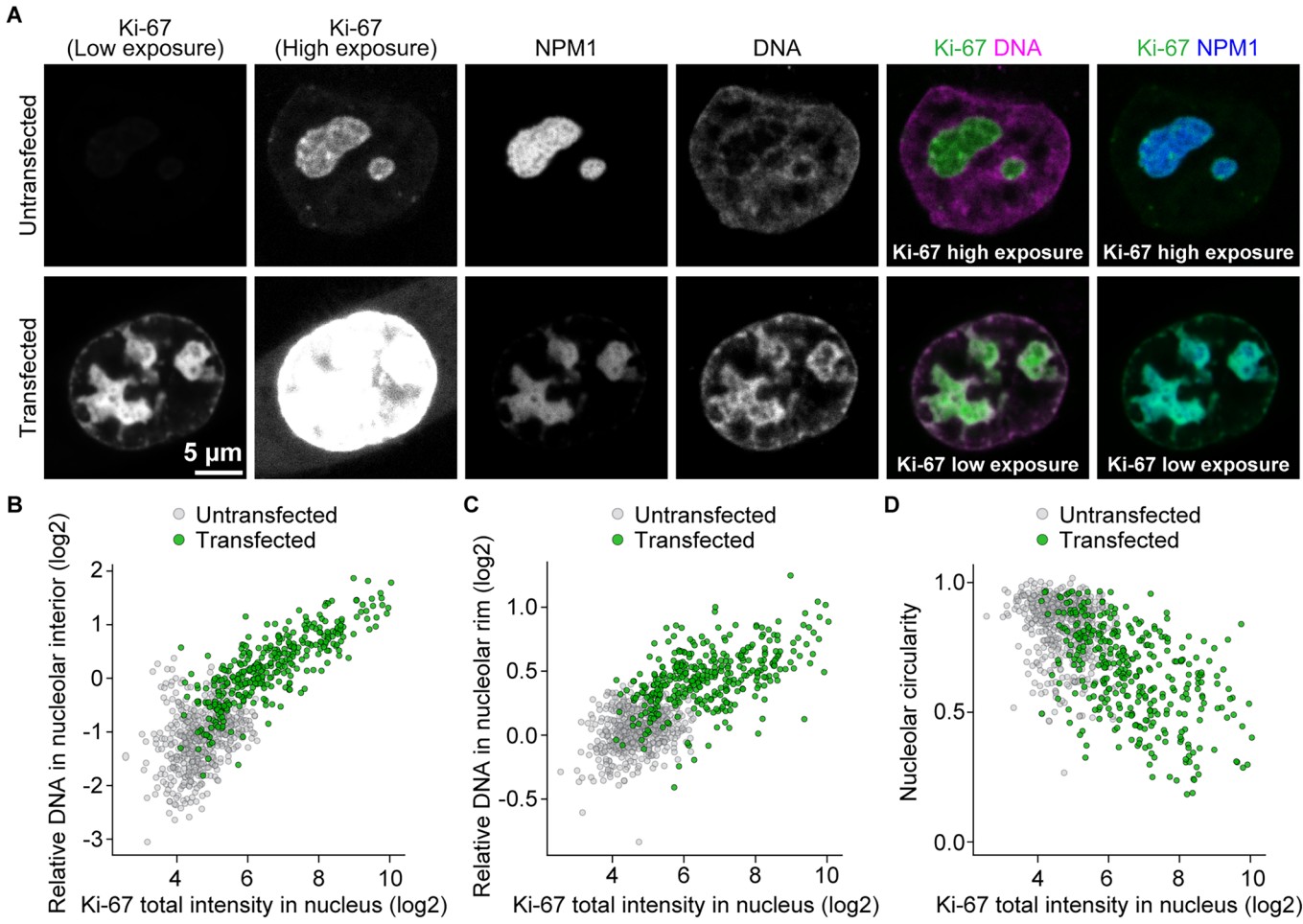

**Figure 4. Ki-67 overexpression induces nucleolar irregularity and excessive chromatin loading.**

(A) Live-cell imaging of Ki-67-overexpressing HeLa cells. Cells endogenously tagged with EGFP-Ki-67 and stably overexpressing SNAP-NPM1 were transfected with an EGFP-Ki-67 plasmid and isolated by FACS. SNAP-NPM1 was labelled with SNAP-SiR, and DNA was stained with SPY555-DNA. Two different Ki-67 signal intensities are shown by adjusting brightness and contrast settings. (B, C) Ki-67 expression level-dependent chromatin enrichment in the nucleolar interior (B) and its rim (C). Relative DNA signal intensities, calculated as described for the H2B signal intensities in Fig. 3C,D, are plotted against the total EGFP-Ki-67 intensity in the nucleus. (D) Ki-67 expression level-dependent increase in the irregularity of nucleolar shape. Median circularity of segmented nucleoli from NPM1 signals per nucleus is plotted against the total intensity of EGFP-Ki-67 in the nucleus. For (B–D), n = 517 nuclei (Untransfected); n = 355 nuclei (Transfected), 2 biological replicates. See also Fig. EV6, Appendix Fig. S4. Source data are available online for this figure.

## Nucleolar rounding is reversible and a direct consequence of Ki-67 removal

Since the siRNA-mediated protein knockdown generally takes 2–3 days to achieve efficient protein depletion, it was unclear whether the observed nucleolar rounding and chromatin removal from nucleoli were a direct consequence of Ki-67 depletion or resulted from secondary effects such as alterations in gene expression by altering chromatin organisation (Sobecki et al, 2016). To test this, we employed an auxin-inducible degron system (Natsume et al, 2016; Yesbolatova et al, 2019) for the rapid degradation of Ki-67. We generated a knock-in cell line endogenously expressing Ki-67 tagged with EGFP and a miniDegron-tag (Morawska and Ulrich, 2013) at its N-terminus and stably expressing the F-box protein OsTIR1 and the nucleolar marker FBL-TagRFP. Treatment with indole-3-acetic acid (IAA) rapidly decreased Ki-67 fluorescence to

background levels within 2 h (Fig. 5A, top panels and 5B). Removal of IAA from the culture medium restored Ki-67 fluorescence levels to approximately 90% at 14 h after washout (Fig. 5A, bottom panels and 5B, last timepoint), providing an opportunity to assess the correlation between acute changes in Ki-67 levels and nucleolar rounding and intranucleolar chromatin levels.

The addition of IAA immediately decreased chromatin levels in the nucleolar interior and induced nucleolar rounding within 2–3 h (Fig. 5C). While intranucleolar chromatin levels reached their minimum after 5 h, rounding continued to decrease until 15 h post-treatment (Fig. 5D). Importantly, the changes in intranucleolar chromatin levels and nucleolar roundness began to recover after the IAA washout. Both parameters required approximately 7 h for recovery, by which time Ki-67 levels had reached about 54% of the pre-depletion levels. Neither parameter fully recovered to the initial baseline value within the time imaged, possibly due to incomplete

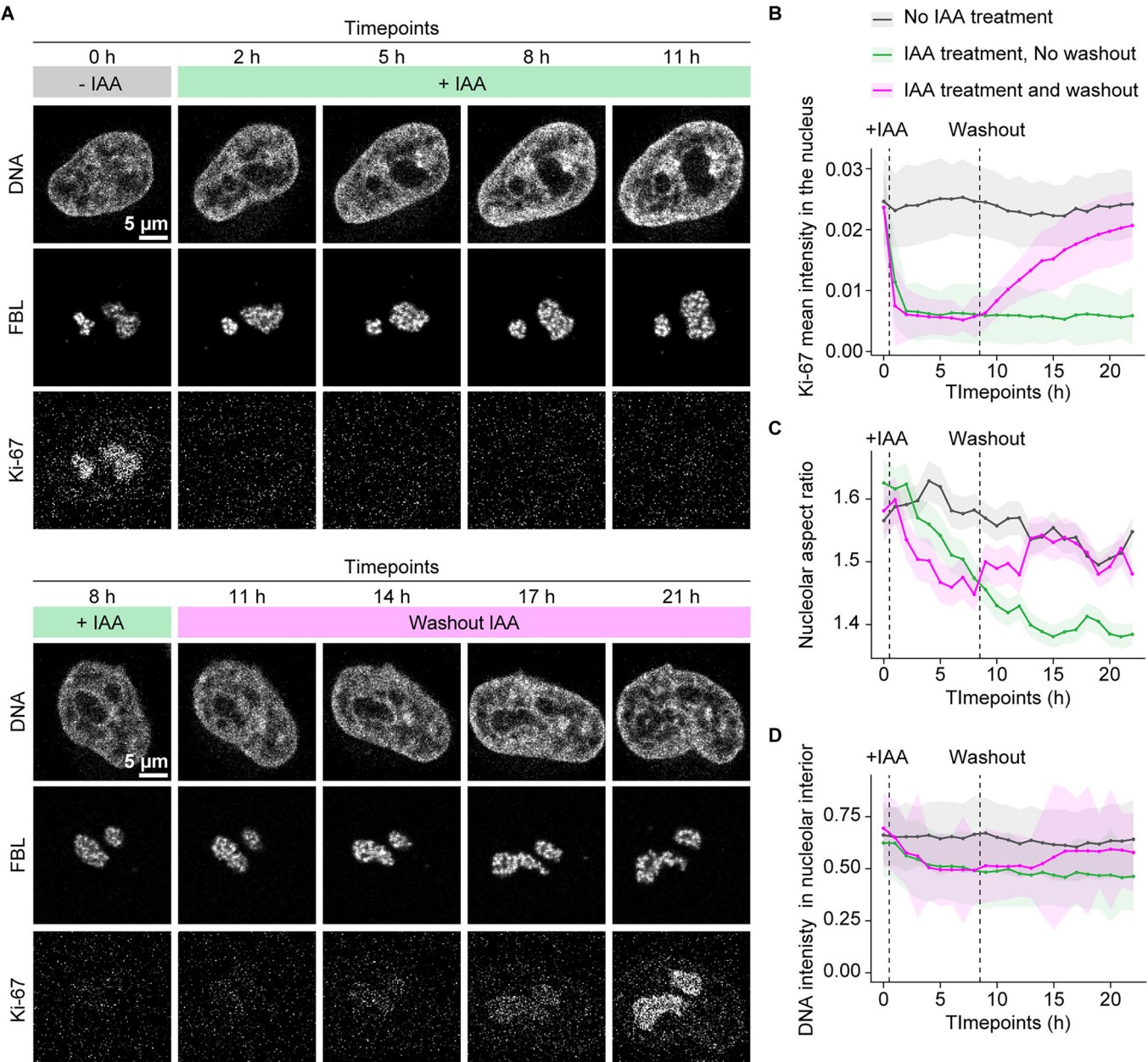

**Figure 5. Nucleolar rounding and chromatin enrichment in the nucleolus are reversible and direct consequences of Ki-67 removal.**

(A) Acute degradation and recovery of Ki-67 and its effect on nucleolar shape and chromatin enrichment in the nucleolus. HeLa cells endogenously expressing EGFP-AID-Ki-67 and stably overexpressing FBL-TagRFP and F-box protein OsTIR1 cells were treated with 3-indole-acetic acid (IAA) 0.5 h after the start of time-lapse imaging (top panels). For the recovery assay, IAA was washed out 8.5 h after the start of time-lapse imaging (bottom panels). DNA was stained with SiR-DNA. (B) Quantification of Ki-67 levels under control (no IAA), IAA treatment, and IAA treatment with subsequent washout. Panel A shows a representative image of the two IAA conditions. The mean intensity of Ki-67 in the nuclei of the indicated conditions was measured. Bars and shade indicate mean ± SD. (C, D) Quantification of nucleolar shape and intranucleolar chromatin levels. Nucleolar aspect ratio (D), and DNA levels in the nucleolar interior (E) of indicated conditions were measured. Bars and shade indicate mean ± SEM (C) and mean ± SD (D). For (B–D), $n = 193–391$ nuclei (No IAA treatment), $n = 171–381$ nuclei (IAA treatment, No washout), $n = 161–298$ nuclei (IAA treatment, Washout) per time point, 2 biological replicates. See also Appendix Fig. S5. Source data are available online for this figure.

recovery of EGFP-miniDegron-Ki-67 expression. As an IAA-insensitive control, we used a cell line expressing wild-type Ki-67 and FBL-TagRFP (Appendix Fig. S5A). The nucleolar shape and intranucleolar chromatin of IAA-insensitive control cells were largely unaffected by imaging conditions with or without IAA treatment (Appendix Fig. S5B,C). Importantly, the observed alterations of

nucleolar shape and chromatin enrichment upon Ki-67 depletion and their recovery were evident in the absence of cell division (Fig. 5A). Thus, the rapid changes in nucleolar roundness and chromatin enrichment following Ki-67 protein depletion and recovery provide strong evidence that Ki-67 directly regulates both nucleolar morphology and chromatin enrichment within the nucleolar interior.

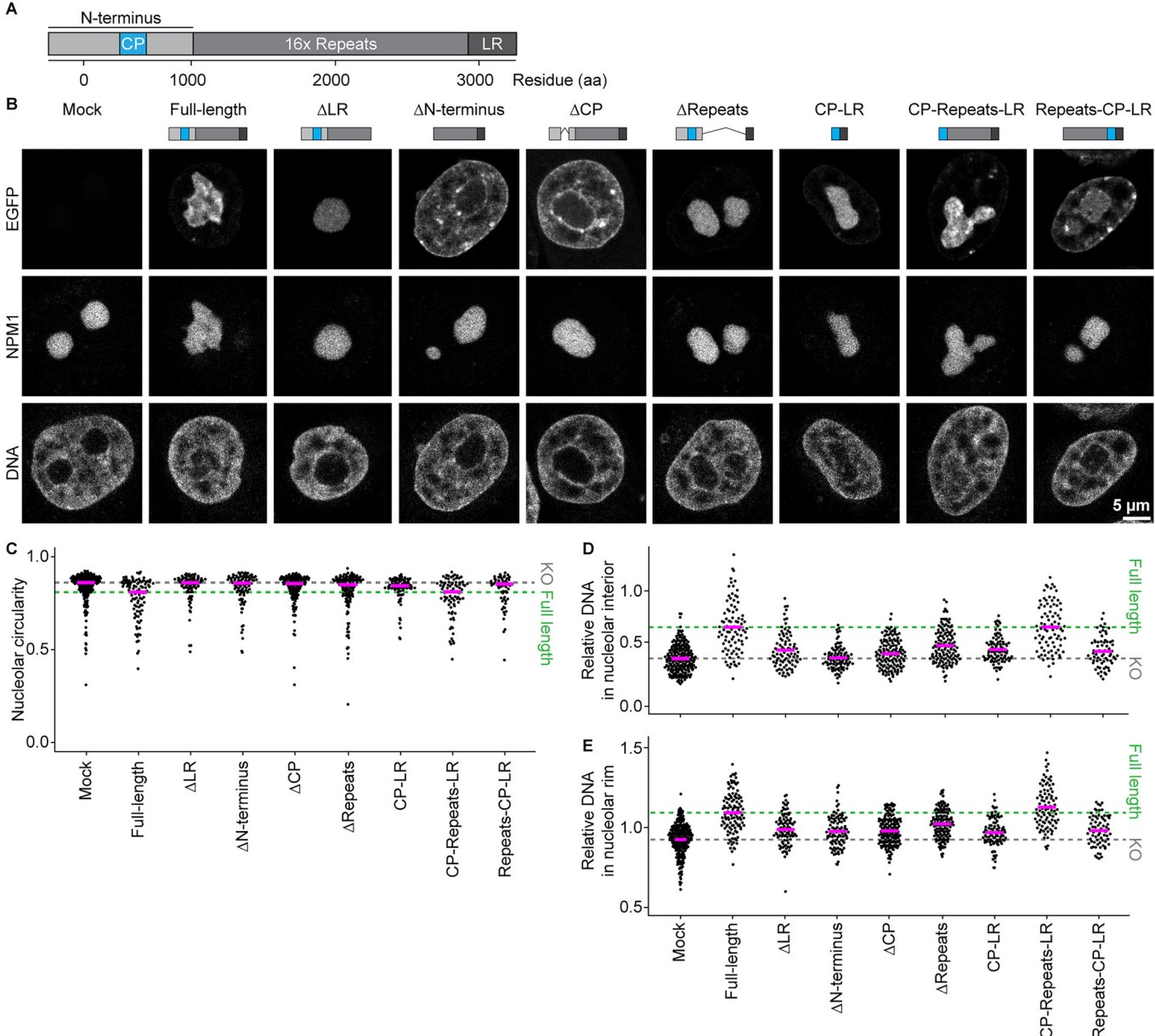

**Figure 6. Ki-67's dual affinity domains, separated by the repeat region, regulate nucleolar shape and chromatin enrichment.**

(A) Schematic representation of Ki-67 domains. CP: positively charged patch of 186 amino acids, LR: leucine/arginine-rich DNA-binding domain. (B) Live-cell imaging of Ki-67 KO HeLa cells transfected with Ki-67 domain mutants. Cells stably expressing mTurquoise2-NPM1 and transfected with EGFP-tagged Ki-67 mutant plasmids were collected by FACS and imaged on a confocal microscope. DNA was stained with SiR-DNA. (C) Quantification of nucleolar circularity in Ki-67 KO cells expressing different Ki-67 domain mutants. The nucleolus was segmented based on NPM1 signals, and the median nucleolar circularity per nucleus was calculated for cells expressing Ki-67 mutants at levels comparable to endogenous Ki-67. Dashed lines indicate median values of full-length Ki-67 (green) and mock-transfected cells (grey). (D, E) Quantification of chromatin enrichment in the nucleolus of cells expressing different Ki-67 domain mutants. Relative DNA signal intensities in the nucleolar interior (D) and rim (E) were measured, as described in (Fig. 3D,E), for cells expressing Ki-67 mutants at endogenous Ki-67 levels. Dashed lines indicate median values of full-length Ki-67 (green) and mock-transfected cells (grey). For (C–E), n = 220 nuclei (mock), n = 87 nuclei (full-length), n = 88 nuclei (ΔLR), n = 94 nuclei (ΔN-terminus), n = 139 nuclei (ΔCP), n = 34 nuclei (LR), n = 128 nuclei (ΔRepeats), n = 87 nuclei (CP-LR), n = 86 nuclei (CP-Repeats-LR), n = 69 nuclei (Repeats-CP-LR), 3 biological replicates. See also Fig. EV7. Source data are available online for this figure.

## Ki-67's amphiphilic properties are necessary for its localisation and function in interphase

To investigate how Ki-67 mediates nucleolar chromatin incorporation and concurrent shape changes, we tested which specific domain of Ki-67 is required for these functions. Ki-67 is a large (360 kDa) disordered protein that consists of mainly three regions (Fig. 6A): a C-terminal leucine/arginine-rich region (LR domain) implicated in DNA binding (MacCallum and Hall, 2000; Saiwaki et al, 2005), a middle region characterised by 16 repeats (Repeats),

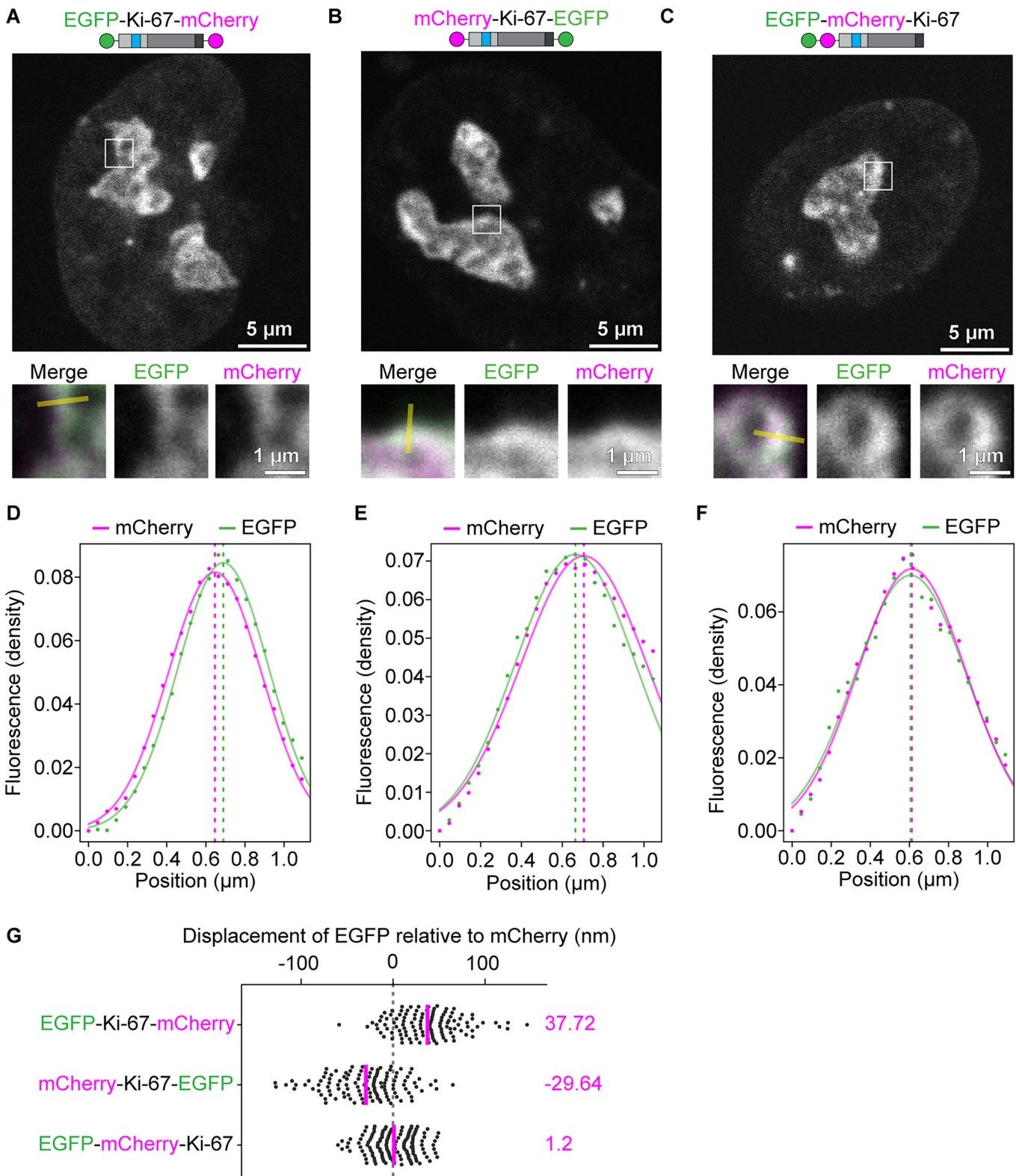

and an N-terminal domain containing a highly positively charged patch (CP domain) involved in clustering of chromosomes during anaphase (Hernandez-Armendariz et al, 2024). The N-terminus also includes two short structured domains: a forkhead-associated (FHA) domain implicated in phosphopeptide binding (Pennell et al, 2010; Sun and Kaufman, 2018) and a Protein phosphatase 1 (PP1) binding motif (Booth et al, 2014).

We transiently expressed wild-type EGFP-Ki-67 and different truncation mutants in Ki-67 KO cells stably expressing mTurquise2-NPM1 as a nucleolus marker (Fig. 6B), and FACS-sorted them

**Figure 7. Ki-67 adopts a partially extended, oriented structure within the nucleolus.**

(A–C) Live-cell imaging of dual fluorescence-tagged Ki-67. HeLa cells were transfected with EGFP-Ki-67-mCherry (A), mCherry-Ki-67-EGFP (B), or EGFP-mCherry-Ki-67 (C) and imaged using a confocal microscope. Insets display EGFP and mCherry signals at the nucleolar rim. The yellow line indicates the line profile from outside to inside the nucleolus. (D–F) Quantification of fluorescence signal densities along the line profile. Fluorescence intensity values were extracted along the indicated line profiles and fitted with Gaussian functions (solid lines). The positions of peak fluorescence intensities derived from the Gaussian fits are marked with dashed lines. (G) Quantification of EGFP signal displacement relative to mCherry signals at the nucleolar periphery. Line profiles were performed in cells transiently expressing EGFP-Ki-67-mCherry, a reversed order of fluorescent proteins (mCherry-Ki-67-EGFP), or N-terminal tagging of both fluorescent tags (EGFP-mCherry-Ki-67). Displacement was measured as the distance between EGFP and mCherry peaks after Gaussian fitting. For (G), $n = 116$ lines from 52 nuclei (EGFP-Ki-67-mCherry), $n = 119$ lines from 52 nuclei (mCherry-Ki-67-EGFP), $n = 135$ lines from 53 nuclei (EGFP-mCherry-Ki-67), 3 biological replicates. Source data are available online for this figure.

using the cell line with endogenous EGFP-Ki-67 as a reference (Fig. EV7A). In our analysis, we only considered cells with similar expression levels to endogenous EGFP-Ki-67 (Fig. EV7B–E). As expected, mock-transfected Ki-67 KO cells exhibited rounded nucleoli lacking chromatin (Fig. 6B, Mock), and full-length Ki-67 localised at the chromatin-nucleolus boundary, restoring the wild-type phenotype with irregular nucleolar shape and chromatin enrichment (Fig. 6B–E, Full-length). Similar to our previous experiments (Fig. 4), higher expression levels of full-length Ki-67 above the wild-type levels decreased nucleolar circularity and increased DNA intensity within nucleoli in a dose-dependent manner (Fig. EV7B–E, full-length).

Previous studies have indicated that the C-terminal LR domain is important for chromatin binding, while the N-terminus is required for nucleolus localisation (Takagi et al, 1999; Saiwaki et al, 2005). In line with this, our Ki-67 mutant lacking the LR domain (ΔLR) localised homogeneously within the nucleolus, while a Ki-67 mutant lacking the N-terminus region (ΔN-terminus) localised to chromatin (Fig. 6B). Given our previous finding that the CP-domain of Ki-67 is critical for its phase separation with RNA (Hernandez-Armendariz et al, 2024), we hypothesised that this domain might anchor Ki-67 to the nucleolus. Indeed, only the removal of the CP domain (ΔCP) relocalised Ki-67 from nucleoli to chromatin. Importantly, in contrast to full-length Ki-67, none of the mislocalised mutants (ΔLR, ΔN-terminus, ΔCP) were able to restore the irregular shape of nucleoli or the enrichment of chromatin to the nucleolus (Fig. 6C–E). These findings demonstrated that the LR and CP domains are essential for Ki-67's localisation at the boundary between chromatin and nucleoli and for the regulation of the nucleolar shape as well as chromatin enrichment.

To test whether CP and LR are sufficient for Ki-67 localisation and function, we tested a construct lacking the middle repeat region (ΔRepeats) as well as a direct CP-LR fusion. Surprisingly, both constructs localised homogenously to nucleoli and could only partially restore the irregular shape and chromatin enrichment (Fig. 6B–E). These findings indicate that while the CP and LR play a crucial role, they are insufficient on their own to fully rescue the Ki-67 wild-type phenotype. In contrast, a minimal Ki-67 construct with the repeats between CP and LR (CP-Repeats-LR) was able to localise correctly and restore nucleolar irregularity and chromatin enrichment (Fig. 6B–E), suggesting that the repeat region is also required for proper localisation and function. Interestingly, however, a shuffled minimal Ki-67 mutant (Repeats-CP-LR) did not restore these parameters efficiently even at higher expression levels than endogenous Ki-67 (Fig. EV7B–E). It is thus tempting to speculate that proper spacing between the CP and LR regions is necessary for the full functionality of Ki-67.

Based on the hypothesis that Ki-67 requires a spacer between the nucleolus and chromatin binding domains, we wondered whether it

adopts an extended conformation at the chromatin-nucleolus border. To test this, we expressed a Ki-67 version with EGFP at one end and mCherry at the other (Fig. 7A,B), or both fluorescence tags at the Ki-67 N-terminus (Fig. 7C). We acquired high-resolution confocal images of nucleoli and measured the fluorescence signal intensity of both tags along line profiles from the nucleolar periphery inward. By fitting Gaussian functions (Fig. 7D–F), we determined peak positions in both channels and calculated the displacement between them (Fig. 7G), indicating molecular extension and orientation. For EGFP-Ki-67-mCherry, the displacement was $37.7 \pm 35.2$ nm, indicating that Ki-67's N-terminus protrudes towards the nucleolar interior. The reverse construct (mCherry-Ki-67-EGFP) showed a displacement of $-29.6 \pm 38.4$ nm, while the control construct (EGFP-mCherry-Ki-67) had only $1.2 \pm 24.8$ nm, confirming that the previous measurements reflected molecular extension. Interestingly, this extension is comparable to Ki-67's extension on the chromosome surface during mitotic exit (Cuylen-Haering et al, 2020).

In summary, we have shown that Ki-67 localises specifically to the chromatin-nucleolar boundary with a clear orientation and an extension of ~33 nm. The dual affinity domains, CP and LR, separated by a repeat-containing region, are essential for Ki-67 boundary localisation and regulation of nucleolar shape and nucleolar chromatin enrichment.

## Ki-67-dependent chromatin recruitment regulates nucleolar shape

Given that our previous analysis showed a strong link between irregular nucleolar shape and increased chromatin incorporation within nucleoli, we hypothesised that the chromatin network within and around nucleoli might be the major determinant of nucleolar shape.

To test this, we revisited our initial screening hits and aimed to test whether nucleolar rounding induced by knockdown of the top 20 candidate genes correlates with chromatin depletion both within nucleoli and at the rim. We depleted candidate genes with two to three different siRNAs in cells stably expressing H2B-mCherry and SNAP-NPM1 and imaged cells using automated confocal spinning disk microscopy. Strikingly, the depletion of all 20 candidate proteins led to a reduction in chromatin levels within the nucleolar interior resembling the effects of Ki-67 depletion (Figs. 8A and EV8A). In addition, the decrease in chromatin levels within the nucleolar rim was evident in the majority of these conditions (Fig. 8B). This finding supports the hypothesis that round nucleoli are generally depleted of chromatin.

To investigate whether Ki-67 plays a crucial role in chromatin depletion under these conditions, we knocked down candidate

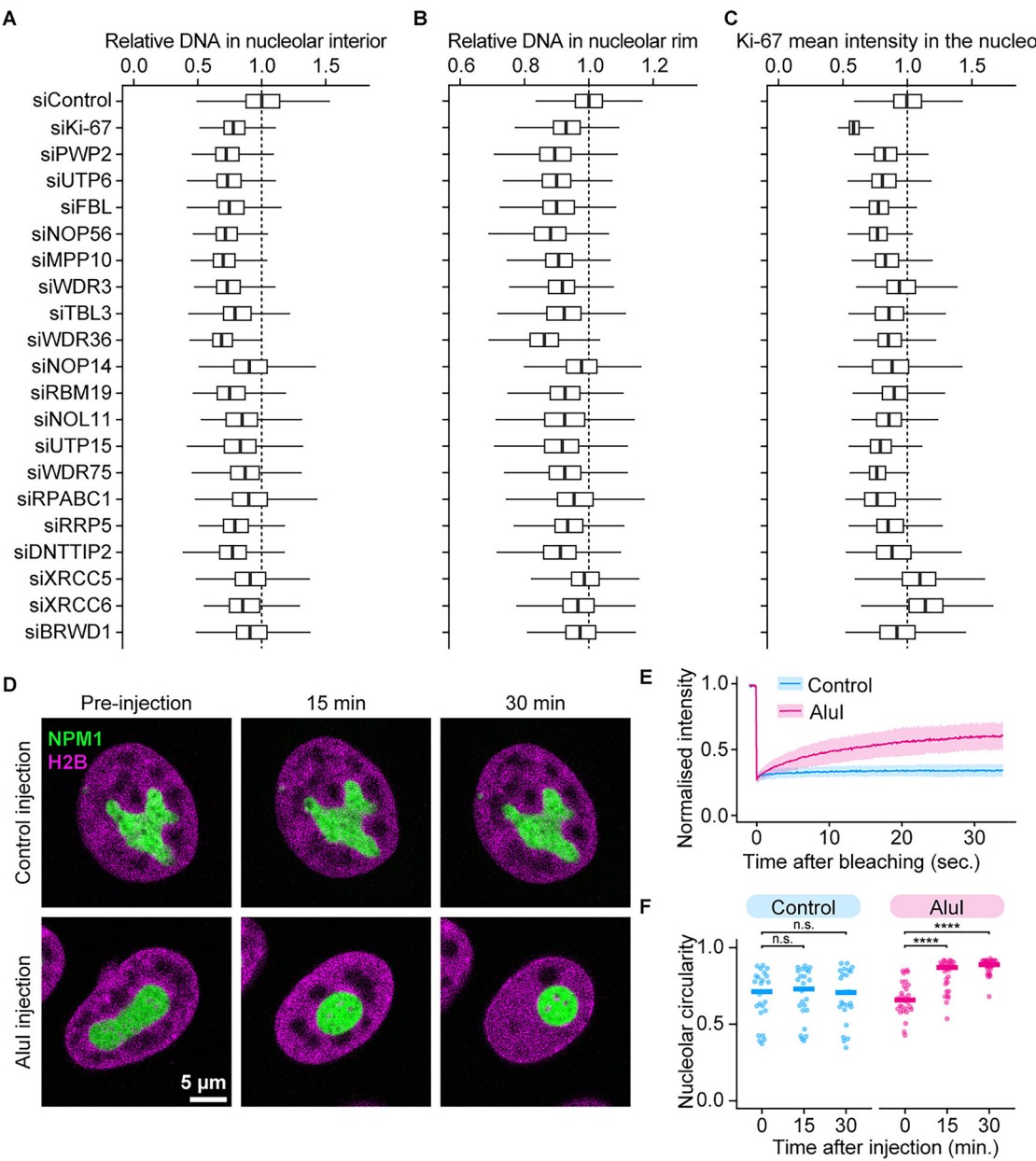

**Figure 8. Chromatin in and around nucleoli is a key determinant of nucleolar shape.**

(A, B) Quantification of chromatin levels in the nucleolus following the depletion of the top 20 candidates causing nucleolar rounding (Fig. 1). Images of HeLa cells stably expressing NPM1-EGFP and H2B-mCherry were acquired using spinning disk microscopy. Relative H2B signal intensities in the nucleolar interior and the nucleolar rim were calculated as described in (Fig. 3D,E). (C) Quantification of nucleolar Ki-67 levels following the depletion of the top 20 candidates causing nucleolar rounding (Fig. 1). Images of endogenous EGFP-Ki-67 HeLa cells stably expressing SNAP-NPM1 were acquired using a wide-field microscope. The mean intensity of Ki-67 signals was measured in nucleoli, and mean values per nucleus were calculated. (D) Effect of AluI injection on the nucleolar shape. HeLa cells stably expressing SNAP-NPM1 and H2B-mNeonGreen were injected with a control buffer or an AluI-containing buffer. (E) Effect of AluI injection on the chromatin mobility. H2B signals were photo-bleached 30 min after AluI injection. Fluorescence recovery curves of H2B signals following AluI injection were compared to those of undigested cells. Lines and shades indicate mean ± SD. (F) Quantification of the nucleolar shape 30 min after AluI injection. The median nucleolar circularity per nucleus is shown. Bars indicate the median. Statistical comparisons were performed against the initial time point (0 min) in each sample: For control, 15 min, $P = 0.863$; 30 min, $P = 0.863$. For AluI injection, 15 min, $P = 1.03 \times 10^{-6}$; 30 min, $P = 9.44 \times 10^{-12}$. For (A–C), boxplots display the median (centre line), first and third quartiles (box), and whiskers extend to 1.5× the interquartile range. For (A), $n > 1000$ cells per candidate, 3 biological replicates. For (B–D), $n > 500$ cells per candidate, 3 biological replicates. For (E), $n = 15$ nucleus (undigested) and $n = 14$ (AluI injection), 2 biological replicates. For (F), $n = 25$ nuclei (Control injection) and $n = 29$ (AluI injection), 2 biological replicates. Statistical tests were performed with the Wilcoxon Test, ns (not significant) $P > 0.05$, ****$P < 0.0001$. See also Fig. EV8. Source data are available online for this figure.

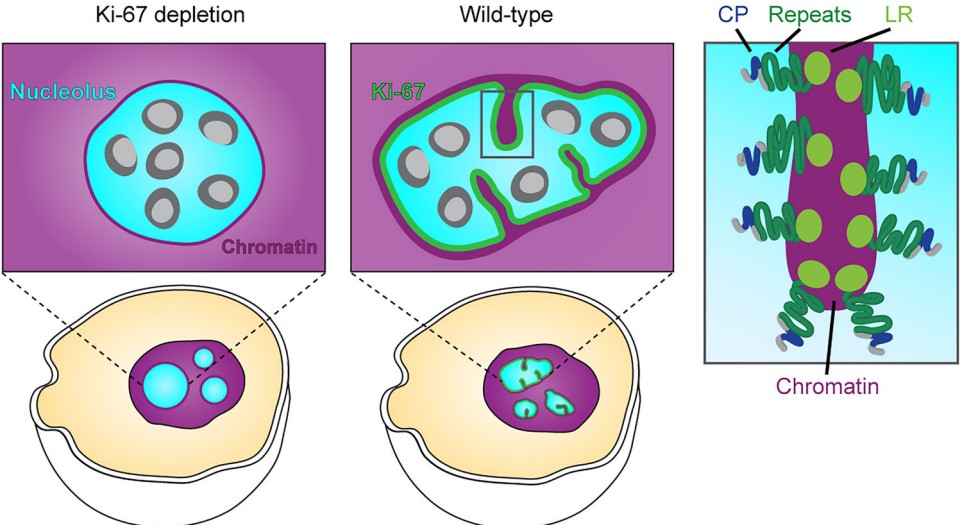

**Figure 9. Ki-67-mediated chromatin anchoring to the nucleolus shapes nucleolar morphology.**

Working model of nucleolar shape regulation by Ki-67. Ki-67 localises to the chromatin-nucleolus interface, where it anchors chromatin to the nucleolar surface and promotes chromatin invagination into the nucleolar interior. This localisation and function rely on the amphiphilic nature of Ki-67, characterised by a positively charged patch (CP) that interacts with the nucleolus and a leucine-arginine-rich (LR) domain that binds to chromatin, separated by a repeat domain. Through regulating the chromatin network around the nucleolus, Ki-67 prevents nucleolar rounding and thus determines nucleolar shape.

genes in a cell line stably expressing endogenous EGFP-Ki-67 and stably overexpressing SNAP-NPM1, followed by imaging with automated wide-field microscopy. Interestingly, in 18 of the 20 knockdowns, Ki-67 levels were lower than in the control (Figs. 8C and EV8B), raising the possibility that Ki-67 may regulate the chromatin environment even in these conditions. However, nucleolar rounding and chromatin depletion from nucleoli can occur even when Ki-67 levels remain high. For instance, depletion of the DNA damage proteins XRCC5 and XRCC6 increased Ki-67 levels but still led to rounded nucleoli and a marked reduction of intranucleolar chromatin. In summary, these findings suggest that chromatin is the major determinant of nucleolar shape.

As an alternative test of this conclusion, we aimed to acutely disrupt the chromatin network within and around nucleoli and observe the effect on nucleolar shape. We performed acute chromatin digestion in a cell line expressing a nucleolar marker (SNAP-NPM1) and a chromatin marker (H2B-mNeonGreen) by microinjecting the restriction enzyme AluI (Fig. 8D), which has been shown to efficiently digest chromatin in living cells (Schneider et al, 2022). We confirmed successful chromatin digestion by observing an increase in H2B-mNeonGreen mobility (Fig. 8E). Remarkably, within 15 min of AluI microinjection, nucleoli became noticeably rounded (Fig. 8D,F). Together, these findings reinforce the hypothesis that the chromatin network plays a central role in determining nucleolar shape.

## Discussion

Here, we uncover a chromatin-based mechanism that regulates nucleolar shape (Fig. 9): Ki-67 recruits chromatin to the nucleolar surface and drives the formation of chromatin invaginations into the nucleolar interior, causing nucleoli to adopt an irregular shape.

This chromatin tethering function relies on Ki-67's dual affinity for chromatin and nucleoli: The N-terminal CP domain has affinity for nucleoli, while the C-terminal LR domain, separated from the CP by a repeat region, interacts with chromatin. This dual interaction localises Ki-67 at the chromatin-nucleolus interface in a preferred molecular orientation, promoting chromatin enrichment at the nucleolus and thereby driving the formation of chromatin invagination from the nucleolar surface. Given that chromatin depletion from the nucleolus is commonly observed in rounded nucleoli (Fig. 8A,B), we propose that the chromatin environment of nucleoli plays a crucial role in regulating nucleolar shape. When chromatin is tightly tethered to the nucleolus, it affects the rigidity and flexibility due to the viscoelastic properties of chromatin, which may deform or stabilise the shape of nucleoli. In addition, nucleolar transcriptional activity has been proposed as the primary determinant of the non-spherical shape of nucleolus (Riback et al, 2023) as nucleoli round up upon RNA Polymerase I transcription inhibition. Interestingly, under these conditions, Ki-67 relocalises from the nucleolus to the nucleoplasm (Sobecki et al, 2016; Kill, 1996), raising the possibility that the rounding upon transcription inhibition may, at least in part, result from a redistribution of chromatin following Ki-67 relocalisation. Together, these observations support a model in which nucleolar shape emerges from the interplay between transcription-dependent nucleolar assembly and Ki-67-mediated chromatin organisation.

Previous data revealed that Ki-67 functions as a biological surfactant (Cuylen et al, 2016), separating mitotic chromosomes by localising to their surface and adopting an extended structure (~90 nm), mediated by Ki-67's amphiphilic nature towards both chromatin and cytoplasm. Given its ability to promote non-spherical nucleolar shapes and to solubilise chromatin within nucleoli, the surfactant model provides a useful framework for understanding Ki-67's interphase behaviour.

In interphase, Ki-67 adopts a more compact (~30 nm) yet still amphiphilic conformation (Fig. 7) and localises in discrete patches at the chromatin-nucleolus interface (Fig. 2A). In addition, its interphase mobility, as measured by fluorescence recovery after photobleaching (FRAP), is markedly reduced compared to mitotic Ki-67 with a large immobile fraction (Saiwaki et al, 2005), suggesting that its interactions are more stable compared to mitosis. This reduced mobility and patchy localisation likely reflects increased chromatin affinity through dephosphorylation (Hernandez-Armendariz et al, 2024) and multivalent interactions, including RNA binding (Ma et al, 2022; Hernandez-Armendariz et al, 2024).

Together, these observations suggest that Ki-67 retains surfactant-like behaviour during interphase, but the nature of this activity is slightly distinct from that in mitosis. The protein forms a semi-solid, persistent interfacial layer rather than a highly dynamic mitotic surfactant. At present, detailed physical parameters such as 2D lateral dynamics, nucleolar surface tension, and intranucleolar chromatin fusion remain to be quantified, preventing a complete mechanistic model.

As an interfacial stabiliser, Ki-67 not only shapes nucleolar morphology but also influences the organisation of surrounding chromatin, which is closely associated with heterochromatin (Padeken and Heun, 2014). Supporting this view, previous studies have shown that Ki-67 contributes to the maintenance of heterochromatin. During interphase, Ki-67 depletion leads to chromatin decompaction, particularly in perinucleolar regions, and a significant decrease in H3K9me3 levels within heterochromatic regions (Sobecki et al, 2016). Conversely, overexpression of Ki-67 caused ectopic heterochromatin formation (Sobecki et al, 2016). Consistent with these observations, Ki-67 preferentially localises to late-replicating genomic regions at the nuclear periphery and around the nucleolus, which are enriched in H3K9me3 (van Schaik et al, 2022). Moreover, Ki-67 is required for tethering heterochromatic elements to the nucleolus, such as satellite DNA (Matheson and Kaufman, 2017), a LacO array near rRNA repeats on chromosome 13 (Booth et al, 2014), centromeric regions marked by CENP-A (Sobecki et al, 2016), and the inactive X chromosome (Sun et al, 2017).

Our study extends these observations by directly demonstrating that Ki-67 controls the spatial organisation of chromatin not only around but also within nucleoli. Ki-67 depletion induces the loss of chromatin signals from the nucleolar interior (Fig. 3), whereas its overexpression promotes chromatin accumulation in the nucleolus (Fig. 4). Given that intranucleolar chromatin exhibits heterochromatin signatures (Appendix Fig. S5) and originates from chromatin invaginations (Fig. EV2), our findings extend the current model of heterochromatin and NADs beyond the nucleolar surface into the nucleolar interior, providing new mechanistic insight into how Ki-67 shapes nucleolar-associated heterochromatin architecture.

Despite the clear role of Ki-67 in heterochromatin formation and spatial positioning of genomic regions, its impact on gene expression remains less well defined. While several studies reported widespread gene deregulation following Ki-67 depletion (Sobecki et al, 2016; Sun et al, 2017; Mrouj et al, 2021), others observed minimal changes in transcription, with gene repression largely unaffected in HCT116 cells (Garwain et al, 2021; van Schaik et al, 2022). This discrepancy may reflect differences in cell type, particularly whether the p21 pathway is active, as p21 induction following Ki-67 loss can cause secondary effects on gene expression due to cell-cycle perturbations (Sun et al, 2017).

In addition, lamina-associated and nucleolus-associated heterochromatin can dynamically exchange (Kind et al, 2013; van Koningsbruggen et al, 2010; Politz et al, 2016), suggesting that loss of nucleolar tethering may redirect chromatin to the nuclear periphery, where it can remain transcriptionally silent. Consistent with this, upon Ki-67 depletion, regions previously bound by Ki-67 show increased lamin interactions (van Schaik et al, 2022). This spatial redistribution may help preserve the transcriptionally repressive state of these regions, which could explain the relatively modest changes in gene expression observed upon Ki-67 loss. Thus, Ki-67 appears to be one of several overlapping mechanisms that preserve heterochromatin by regulating its spatial positioning.

The precise mechanism by which Ki-67 regulates heterochromatin remains incompletely understood. Its effects may be mediated through previously described direct interactions with chromatin-associated proteins such as HP1 (Scholzen et al, 2002; Kametaka et al, 2002) or the CAF-1 complex (Matheson and Kaufman, 2017; Smith et al, 2014). Our data, including the observed amphiphilic nature of Ki-67 and the excessive chromatin incorporation into nucleoli upon overexpression, support a model in which Ki-67 primarily serves a structural role. We propose that Ki-67 acts as a scaffold that tethers chromatin to the repressive environment of the nucleolus, thereby promoting compaction and silencing indirectly through spatial sequestration. Whether and how Ki-67 contributes more directly to the formation or maintenance of heterochromatic marks remains to be fully explored.

Building on this structural role, Ki-67's surfactant-like activity at the chromatin–nucleolus interface further promotes chromatin invagination into the nucleolus, leading to the formation of irregularly shaped nucleoli with increased surface area. This expanded interface may not only facilitate transcriptional repression but may also enhance the directional export of mature ribosomal subunits to the cytoplasm, supporting the elevated ribosome production necessary for cell growth and protein synthesis. Intriguingly, rapidly proliferating cells, such as stem/ progenitor cells and cancer cells, generally express high levels of Ki-67 (Zhang et al, 2014; Li et al, 2015) and are characterised by large, irregular nucleoli (Helpap, 1988; Zink et al, 2004), features associated with their high proliferative and metabolic demands (Derenzini et al, 2000). Conversely, Ki-67 protein is typically undetectable in senescent cells (Lawless et al, 2010; Ashraf et al, 2023). A hallmark of cellular senescence is a change in cell morphology (González-Gualda et al, 2021), which includes the nucleolus adopting a more rounded shape (Jo et al, 2024) along with loss of H3K9me3 and displacement of centromeric and pericentromeric alpha-satellite sequences (Dillinger et al, 2017)— regions that strongly interact with Ki-67 (van Schaik et al, 2022). These changes align with our findings that loss of Ki-67 causes nucleolar rounding and disrupts chromatin tethering to nucleoli. These observations suggest that nucleolar shape may be functionally linked to both chromatin organisation and cellular growth states, including their impact on ribosome production and gene expression.

Our study further highlights the importance of interfacial proteins in tightly connecting and shaping two membrane-less compartments, nucleoli and heterochromatin. In light of the growing recognition of the interfaces between biological condensates and cellular structures, including chromatin, RNA, and other

membrane-less compartments (Böddeker et al, 2022; Snead et al, 2022; Choi et al, 2024; Strom et al, 2024; preprint: Schneider et al, 2025; Rajshekar et al, 2025), amphiphilic proteins like Ki-67 might be crucial in controlling the properties and fluxes (Riback et al, 2020, 2023; King et al, 2024; Dai et al, 2024) between membrane-less compartments, thereby influencing their organisation and function. Further studies of interfacial assemblies in cells, as well as on synthetic co-condensates in vitro (Kelley et al, 2021), will not only enhance our understanding of the regulatory mechanisms that control phase separation in cells but will also inform the rational design of engineered condensates, with potential applications in biosynthesis and drug delivery.

# Methods

### Reagents and tools table

| Reagent/resource | Reference or source | Identifier or catalogue number |
|---|---|---|
| **Experimental models** | | |
| Wild-type cells | Originally from S. Narumiya (Kyoto University, Japan) | Lab ID c-1 |
| NPM1-EGFP, H2B-mCherry | This study | Lab ID c-36 |
| siKi-67 no. 2 resistant (homozygous) | Cuylen et al, 2016 | Lab ID c-61 |
| Ki-67 KO | Cuylen et al, 2016 | Lab ID c-63 |
| mEGFP-Ki-67 (homozygous) | Cuylen et al, 2016 | Lab ID c-77 |
| mEGFP-Ki-67 (homozygous), H2B-mcherry | Gift from Daniel W. Gerlich | Lab ID c-80 |
| FBL-TagRFP (RIEP) | This study | Lab ID c-161 |
| mEGFP-miniDegron-Ki-67 (homozygous), OsTir1, FBL-TagRFP (RIEP) | This study | Lab ID c-163 |
| Ki-67-KO, mTurquoise2-NPM1, FBL-mScarlet (RIEP) | This study | Lab ID c-349 |
| mEGFP-Ki-67 (homozygous), FBL-Halo, SNAP-NPM1 | This study | Lab ID c-391 |
| mEGFP-Ki-67 (homozygous), H2B-mCherry, SNAP-NPM1 | This study | Lab ID c-394 |
| FBL-Halo, SNAP-NPM1 (RIEP) | This study | Lab ID c-318 |
| Halo-UBF (homozygous), SNAP-NPM1, H2B-mNeonGreen | This study | Lab ID c-534 |
| Halo-UBF (homozygous), SNAP-NPM1 | This study | Lab ID c-567 |

| Reagent/resource | Reference or source | Identifier or catalogue number |
|---|---|---|
| Halo-UBF (homozygous), FBL-mEGFP, SNAP-NPM1 | This study | Lab ID c-395 |
| U2-OS EGFP-NPM1 (RIEP) | This study | Lab ID c-309 |
| RPE-1 SNAP-NPM1, FBL-Halo (RIEP) | This study | Lab ID c-322 |
| **Recombinant DNA** | | |
| px335 | Addgene | #42335 |
| pUC57_UBTF_donor_N-term_Halo | This study | Donor vector to insert a Halo tag to UBF's N-terminus, Lab ID p-668 |
| pCR2.1_Ki-67-3'UTR_mini-AID-EGFP_Ki-67 N-term | This study | Donor vector to insert a mini-AID tag to Ki-67's N-terminus, Lab ID p-272 |
| mEGFP-Ki-67 | Hernandez-Armendariz et al, 2024 | Full-length Ki-67, Lab ID p-343 |
| mEGFP-Ki-67ΔRepeats | Hernandez-Armendariz et al, 2024 | Full-length Ki-67 lacking its 16xRepeats (aa 1003–2929), Lab ID p-607 |
| mEGFP-Ki-67ΔN-term | Hernandez-Armendariz et al, 2024 | Full-length Ki-67 lacking its N-terminal region (aa 1–1002), Lab ID p-665 |
| mEGFP-Ki-67ΔCP | Hernandez-Armendariz et al, 2024 | Full-length Ki-67 lacking its charged patch (CP, aa 496–681), Lab ID p-483 |
| mEGFP-Ki-67ΔLR | Hernandez-Armendariz et al, 2024 | Full-length Ki-67 lacking its LR domain (aa 2930–3256), Lab ID p-428 |
| mEGFP-CP-LR | This study | Charged patch (CP, aa 496–681) fused to LR-domain (aa 2930–3256), Lab ID p-757 |
| mEGFP-CP-Repeats-LR | Hernandez-Armendariz et al, 2024 | Charged-Patch (CP, aa 496–681) fused to C-terminus of Ki-67 containing 16xRepeats and the LR-domain (aa 1003–3256), Lab ID p-818 |
| mEGFP-Repeats-CP-LR | This study | 16xRepeats (aa 1003–2929) fused to the charged patch (CP, aa 496–681) fused to the LR domain (aa 2930–3256), Lab ID p-819 |
| Halo-Ki-67 | This study | Full-length Ki-67, Lab ID p-391 |
| EGFP-mCherry-Ki-67 | Cuylen et al, 2016 | Lab ID p-249 |
| mCherry-Ki-67-EGFP | Cuylen et al, 2016 | Lab ID p-216 |
| EGFP-Ki-67-mCherry | Cuylen et al, 2016 | Lab ID p-219 |
| **Antibodies** | | |
| Rabbit anti-Ki-67 | Abcam | ab16667 |
| Rabbit anti-H3K9me3 | Merck | 07-442 |
| Mouse anti-HP1α | Merck | 05-689 |
| Rabbit anti-H4K20me3 | Abcam | ab9053 |
| Rabbit anti-HP1β | Abcam | ab10478 |

| Reagent/resource | Reference or source | Identifier or catalogue number |
|---|---|---|
| Goat anti-rabbit IgG, Alexa Fluor 488 | Thermo Fisher Scientific | A11034 |
| Goat anti-rabbit IgG, Alexa Fluor 594 | Thermo Fisher Scientific | A11037 |
| Goat anti-mouse IgG, Alexa Fluor 594 | Thermo Fisher Scientific | A11032 |
| **Oligonucleotides and other sequence-based reagents** | | |
| siControl (SilencerSelect negative control) | Thermo Fisher Scientific | s44426 |
| siKi-67 #1 (SilencerSelect) | Thermo Fisher Scientific | s8798 |
| siKi-67 #2 (SilencerSelect) | Thermo Fisher Scientific | s8796 |
| Silencer Select siRNA library | Thermo Fisher Scientific | See Supplementary Table 1 |
| sgRNA Ki-67 no. 1 | Cuylen et al, 2016 | 5'-AATGTGGCCCACGAGACGCC-3' |
| sgRNA Ki-67 no. 2 | Cuylen et al, 2016 | 5'-TGAGTATAATCCGTAGGGGA-3' |
| sgRNA UBTF no. 1 | This study | 5'-AGCCACCTCCTCGGTCGTGC-3' |
| sgRNA UBTF no. 2 | This study | 5'-AGCCGACTGCCCCACAGACC-3' |
| rDNA DNA-FISH primary oligo pool (custom) | IDT | Custom oligo pool; see "Methods" |
| DNA-FISH secondary imager probes | IDT | Custom; see "Methods" |
| **Chemicals, enzymes and other reagents** | | |
| DMEM | Thermo Fisher Scientific | 41965039 |
| Foetal bovine serum (FBS) | Thermo Fisher Scientific | 10270106 |
| Penicillin-streptomycin | Sigma-Aldrich | 15140122 |
| Sodium pyruvate | Thermo Fisher Scientific | 11360039 |
| Geneticin (G418) | Thermo Fisher Scientific | 11811031 |
| Puromycin | Merck Millipore | 540411 |
| Blasticidin S hydrochloride | Sigma-Aldrich | 15205 |
| Hygromycin | Thermo Fisher Scientific | 10687010 |
| FluoroBrite DMEM | Thermo Fisher Scientific | A1896701 |
| GlutaMAX | Thermo Fisher Scientific | 35050038 |
| X-tremeGENE 9 DNA transfection reagent | Roche | 6365779001 |

| Reagent/resource | Reference or source | Identifier or catalogue number |
|---|---|---|
| Polyethylenimine Max (PEI) | Polysciences | 24765 |
| Lipofectamine RNAiMAX reagent | Thermo Fisher Scientific | 13778030 |
| Halo-TMR ligand | Promega | G8251 |
| Halo-JF646 ligand | Promega | GA1120 |
| SNAP-SiR ligand | New England Biolabs | S9102S |
| SPY595-DNA | Spirochrome | SC301 |
| SPY555-DNA | Spirochrome | SC201 |
| Indole-3-acetic acid (IAA) | Abcam | ab146402 |
| TetraSpeck Fluorescent Microspheres (100 nm) | Thermo Fisher Scientific | T14792 |
| DAPI | Thermo Fisher Scientific | 62248 |
| AluI restriction enzyme | Thermo Fisher Scientific | FD0014 |
| Glycerol | Merck | 1.04092.2511 |
| Magnesium acetate (Mg(OAc)2) | Sigma-Aldrich | M5661 |
| CF680R-conjugated 10k-MW dextran | VWR | 80116 |
| Formamide | Sigma-Aldrich | F9037 |
| Dextran sulfate | Merck | S4030 |
| RNase A | Thermo Fisher Scientific | EN5031 |
| 2x SSC | Sigma-Aldrich | S6639 |
| Opti-MEM | Thermo Fisher Scientific | 31985070 |
| Nuclease-free water | Thermo Fisher Scientific | AM9938 |
| DEPC-treated water | Thermo Fisher Scientific | |
| Triton X-100 | Roth | 3051.2 |
| BSA | Sigma-Aldrich | A7906 |
| **Software** | | |
| Ilastik | ilastik | version 1.4.1 |
| CellProfiler3 | McQuin et al, 2018 | version 3.1.8 |
| CellProfiler4 | Stirling et al, 2021 | version 4.2.5 |
| Fiji | Schneider et al, 2012 | version 1.54p |
| Huygens Professional | Scientific Volume Imaging (SVI) | version 24.10.04p 64b |
| R | R Foundation for Statistical Computing | version 4.4.2 |
| Prism | GraphPad | version not specified |

| Reagent/resource | Reference or source | Identifier or catalogue number |
|---|---|---|
| ZEN Blue | Zeiss | Microscope acquisition software |
| ZEN Black | Zeiss | Microscope acquisition software |
| Benchling | https://benchling.com | sgRNA design |
| OligoMiner | Beliveau et al, 2018 | DNA-FISH probe design |
| **Other** | | |
| 384-well imaging plates | Cellvis | P384-1.5H-N |
| 96-well imaging plates | Cellvis | P96-1.5H-N |
| 8-well chamber slide (Lab-Tek) | Thermo Fisher Scientific | 155411 |
| 35 mm high-wall µ-Dish (polymer bottom) | ibidi | 81156 |
| 8-well glass-bottom slide | ibidi | 80807 |
| FACSAria Fusion Flow Cytometer | BD Bioscience | |
| Neon Transfection System | Thermo Fisher Scientific | |
| ImageXpressMicro XL screening microscope | Molecular Devices | |
| IXplore SpinSR spinning disc microscope (CSU-W1-TS / SoRa scanner) | Olympus; Yokogawa | |
| LSM780 confocal microscope | Zeiss | |
| LSM980 confocal microscope (Airyscan) | Zeiss | |
| 3D Cell Explorer microscope | Nanolive | |
| FemtoJet microinjector | Eppendorf | 5181000017 |
| InjectMan NI2 micromanipulation device | Eppendorf | 5247000013 |
| Femtotips injection capillaries | Eppendorf | 5242952008 |
| Microcapillary Microloader | Eppendorf | 5242956003 |

## Cell lines and cell culture

All of the cell lines used in this study have been regularly verified as negative for mycoplasma contamination. Their sources and authentication are summarised in the Reagents and Tools table. All cell lines used in this study without indication were HeLa-Kyoto cells that have been previously described (Schmitz et al, 2010). Cells were cultured in Dulbecco's modified medium (DMEM; Thermo Fisher Scientific, 41965039) containing 10% (v/v) foetal bovine serum (FBS; Thermo Fisher Scientific, 10270106), 1% (v/v) penicillin-streptomycin (Sigma-Aldrich, 15140122), 1 mM Sodium Pyruvate (Thermo Fisher Scientific, 11360039). For drug selection, cells were cultured in the above medium supplemented with antibiotics according to the expression constructs: 300 µg/mL

geneticin G418 sulphate (Thermo Fisher Scientific, 11811031), 0.5 µg/mL puromycin (Merck Millipore, 540411), 6 µg/mL blasticidin S hydrochloride (Sigma-Aldrich, 15205), and 300 µg/mL hygromycin S at a final concentration (Thermo Fisher Scientific, 10687010). For live-cell imaging, cells were cultured in FluoroBrite DMEM (imaging medium; Thermo Fisher Scientific, A1896701) containing 10% (v/v) FBS, 1% (v/v) penicillin-streptomycin, 1 mM sodium pyruvate, and 1% (v/v) GlutaMAX (Thermo Fisher Scientific, 35050038). All live-cell imaging experiments were performed under the condition of constant humidity and 37 °C temperature with 5% $CO_2$.

## Plasmid construction

All constructs are listed in the Reagents and Tools table. For lentivirus transfer vectors, protein-coding sequences and protein tags were inserted into a backbone vector containing an internal ribosome entry site (IRES) and antibiotic genes by Gibson assembly. To generate the donor vector for EGFP-miniDegron-Ki-67 knock-in, the miniDegron sequence (Morawska and Ulrich, 2013) was inserted into an EGFP knock-in donor vector for Ki-67 N-terminus (Cuylen et al, 2016) by Gibson assembly. For the donor vector for the Halo-tag knock-in at the UBF N-terminus, homology arm sequences (700 bp around the start codon of UBTF gene) and the Halo-tag sequence were synthesised and cloned into pUC57 vector (Biomatik). To generate single-guide RNAs (sgRNAs)/Cas9 nickase (Cas9 D10A) expressing plasmids, sgRNAs were designed on Benchling (https://www.benchling.com/) and assembled into px335 vector (Addgene #42335). The following sgRNA sequences were used: sgRNA Ki-67 no. 1: 5'- AATGTGGCCCACGAGACGCC-3', sgRNA Ki-67 no. 2: 5'- tgagtataatccgtagggga-3', sgRNA UBTF no. 1: 5'-AGCCACCTCCTCGGTCGTGC-3', sgRNA UBTF no. 2: 5'-AGCCGACTGCCCCACAGACC-3'.

## Generation of stable cell lines

Cell lines stably expressing fluorescence-labelled marker proteins were generated by random plasmid integration or a lentiviral vector system pseudotyped with a VSV-G or a mouse ecotropic envelope that is rodent-restricted (RIEP receptor system). Construction of RIEP receptor parental cell lines and subsequent generation of stable cell lines that express fluorescent marker proteins were performed as described previously (Samwer et al, 2017). Genome editing was performed by the CRISPR-Cas9 nickase approach (Koch et al, 2018). HeLa cells endogenously tagged with EGFP at the N-terminus of Ki-67 were previously described (Cuylen et al, 2016). For the generation of auxin-inducible degradation of Ki-67, two sgRNAs targeting Ki-67 and the donor vector containing miniDegron-EGFP having 1-kb homology arms were transfected into cells using X-tremeGENE 9 DNA transfection reagent (Roche, 6365779001). Following a 1-week culture, GFP-positive cells were sorted into a 96-well plate using FACSAria™ Fusion Flow Cytometer (BD Bioscience). The sorted cells were assessed by cell morphology, genotyping PCR, immunoblotting, and fluorescence microscopy to confirm the correct protein. Following the validation of homozygous knock-in cells, OsTiR1 and FBL-TagRFP were stably expressed in the miniDegron-EGFP knock-in cells by lentivirus transduction. For the generation of endogenously expressing Halo-UBF, cells were electroporated with the two Cas9

nickase–sgRNA-expressing plasmids, 5 µg each, and 7.5 µg Halo-tag knock-in donor plasmid using the Neon Transfection System (Thermo Fisher Scientific) with $3 \times 10$ ms pulses at 1300 V. Following a 1-week culture, cells were incubated with 50 nM Halo-TMR (Promega, G8251) for 30 min before sorting. Cells showing TMR positive signals were sorted into a 96-well plate using FACSAria™ Fusion Flow Cytometer (BD Bioscience). The sorted cells were assessed by cell morphology, genotyping PCR, immunoblotting, and fluorescence microscopy to confirm the correct protein.

## RNAi screen

For the screen on nucleolar shape (Fig. 1), 614 genes were targeted by either two or three Silencer select siRNAs (Thermo Fisher Scientific, see Dataset EV1). This target gene list included 63 nucleolar proteins localising to the chromosome surface during mitosis (Ohta et al, 2010), 139 proteins involved in regulating nucleolar number (Farley-Barnes et al, 2018), 112 proteins whose solubility changes in response to increased ATP levels (Sridharan et al, 2019), and 310 proteins that undergo solubility changes during mitosis (Becher et al, 2018). Some proteins were present in more than one library. Using solid-phase reverse transfection (Erfle et al, 2008), 384-well imaging plates (Cellvis, P384-1.5H-N) were coated with siRNA transfection mixes. Cells stably expressing NPM1-EGFP and H2B-mCherry were seeded on the screening plates 72 h prior to imaging using a Multidrop Reagent Dispenser (Thermo Scientific). Images were acquired at 4 different positions in each well on an ImageXpressMicro XL screening microscope (Molecular Devices) using a CFI P-Apo 20× Lambda objective (Nikon).

For the follow-up experiments on nucleolus-associated chromatin levels and Ki-67 levels (Fig. 7A–C), siRNAs transfection and cell seeding were prepared with the same approaches as above using a subset of target genes in 96-well imaging plates (Cellvis, P96-1.5H-N). For the nucleolar chromatin analysis (Figs. 8A,B and EV8A), Cells stably expressing NPM1-EGFP and H2B-mCherry were acquired at four different positions in each well on an IXplore SpinSR spinning disc microscope (Olympus) with a CSU-W1-TS 2D 50/SoRa confocal scanner (Yokogawa) using UPLXAPO ×20 objective (Olympus). For the Ki-67 levels analysis (Figs. 8C and EV8B), cells expressing endogenously tagged EGFP-Ki-67 and stably expressing SNAP-NPM1 and H2B-mCherry were seeded into 384-well imaging plates. SNAP-NPM1 were labelled with 25 nM SNAP-SiR (New England Biolabs, S9102S). Images were acquired at 4 different positions in each well on an ImageXpress-Micro XL screening microscope using a CFI P-Apo 20× Lambda objective.

## Protein knockdown by siRNA transfection

Pre-designed SilencerSelect siRNAs (Thermo Fisher Scientific) were diluted in nuclease-free water (Thermo Fisher Scientific) to a concentration of 10 µM as a stock solution. For fluorescence imaging, the stock siRNA solution was further diluted to 80 nM in DEPC-treated water (Thermo Fisher Scientific) for a working solution. In an 8-well Lab-Tek chamber slide (Thermo Fisher Scientific, 155411), 50 µl of siRNA working solution was mixed with 50 µl of Opti-MEM containing 0.6 µl of Lipofectamine

RNAiMAX Reagent (Thermo Fisher Scientific, 13778030). The cell suspension was incubated for 20 min at RT and seeded to achieve 50–70% confluency on the day of imaging. To assess the nucleolar roundness and the chromatin enrichment in the nucleolus, cells were imaged 72 h after siRNA transfection on a Zeiss LSM780 using a Plan-Apochromat ×63/1.4 NA Oil DIC M27 oil-immersion objective. To examine the effect of Ki-67 depletion on the nucleolar small subcompartment, cells were imaged 48 h after siRNA transfection on an Olympus iXplore SPIN SR using a UPLSAPO ×100/1.35 NA Silicon oil-immersion objective. For holotomographic imaging, 2.5 µL of 10 µM siRNA stocks were diluted in 125 µL Opti-MEM and added to ibidi 35 mm high dishes (ibidi, 81156). Pre-mixed 7.5 µl of 125 µL of a transfection mix prepared from 125 µL per dish Opti-MEM and 7.5 µL/dish Lipofectamine RNAiMAX in 125 µL Opti-MEM was mixed with the diluted siRNA in the dishes. Twenty minutes after incubation at RT, cells suspended in the imaging medium were seeded to achieve 50–70% confluency on the day of imaging. Cells were imaged 72 h after siRNA transfection on 3D Cell Explorer microscope (Nanolive) at 37 °C with 5% $CO_2$ environment. The following siRNAs were used in this study; siControl (XWNeg9; s44426), sense strand 5'-UACGACCGGUCUAUCGUAGtt-3', antisense strand 5'-CUACGAUAGACCGGUCGUAtt-3'; siKi-67 #1 (s8798), sense strand 5'-GUACCAUAAUAAUAGGGAAtt-3'; antisense strand 5'-UUCCCUAUUAUUAUGGUACaa-3' siKi-67 #2 (s8796), sense strand 5'-CGUCGUGUCUCAAGAUCUAtt-3', antisense strand 5'-UAGAUCUUGAGACACGACGtg-3'.

## Live-cell Airyscan imaging

Cells were seeded on eight-well glass-bottom slides (ibidi, 80807) one day before imaging. For Halo-tag and SNAP-tag labelling, cells were incubated with the medium containing 100 nM SNAP-SiR and 1×SPY595-DNA (Spirochrome, SC301) for 1 h at 37 °C. After three washes with fresh medium, the cells were cultured in the imaging medium. Airyscan imaging was performed on a Zeiss LSM980 confocal microscope equipped with an Airyscan detector and a Plan-Apochromat ×63/1.4 NA Oil DIC M27 objective operated by ZEN Blue. Images were acquired using the Airyscan SR mode with 0.15 µm interval for 5 µm thickness. To correct for chromatic aberration, images of 100-nm TetraSpeck Fluorescent Microspheres (Thermo Fisher Scientific, T14792) were acquired using the same acquisition settings. Chromatic aberration was analysed from the bead images using Huygens Professional. The mean chromatic aberration values of the beads were used to correct for chromatic aberration in the cell images.

## Immunofluorescence staining

Cells were seeded in a LabTek slide one day before fixation. To label Halo-tag and SNAP-tag, living cells were incubated with the medium 100 nM Halo-TMR and 100 nM SNAP-SiR for 20 min at 37 °C. After three times washing with pre-warmed medium, cells were cultured in the fresh medium for 10 min. Then, cells were washed with phosphate-buffered saline (PBS) and fixed with 3.7% formaldehyde in PBS at room temperature (RT) for 15 min. After three times washing with PBS and quenching with 100 mM $NH_4Cl$ in PBS at RT for 10 min, cells were permeabilised with 0.2% (v/v) Triton X-100 in PBS at RT for 5 min. Followed by washing three

times with PBS, blocking was performed with 3% Bovine serum albumin (BSA) in PBS at RT for 1 h, and then the cells were incubated with the primary antibody in the blocking buffer at 4 °C overnight. The cells were washed three times with PBS and incubated with the secondary antibody in PBS at RT for 1 h. After twice washing with PBS, DNA was stained with 0.2 µg/ml DAPI (Thermo Fisher Scientific, 62248) in PBS at RT for 5 min. Images were acquired on a Zeiss LSM780 confocal microscope, operated by ZEN Black, with an EC Plan-Neofluar ×40/1.30 Oil DIC M27 oil-immersion objective or a Plan-Apochromat ×63/1.4 NA Oil DIC M27 oil-immersion objective. The following antibodies were used: rabbit anti-Ki-67 antibody (Abcam, ab16667), rabbit anti-H3K9me3 antibody (Merck, 07-442), mouse anti-HP1α antibody (Merck, 05-689), rabbit anti-H4K20me3 antibody (Abcam, ab9053), rabbit anti-HP1β antibody (Abcam, ab10478), goat anti-rabbit IgG conjugated with Alexa Fluor 488 (Thermo Fisher Scientific, A11034), goat anti-rabbit IgG conjugated with Alexa Fluor 594 (Thermo Fisher Scientific, A11037), goat anti-mouse IgG conjugated with Alexa Fluor 594 (Thermo Fisher Scientific, A11032).

## Transient expression of dual fluorescence protein-tagged Ki-67

Wild-type cells were seeded onto a LabTek slide and immediately transfected with plasmids encoding EGFP-Ki-67-mCherry, mCherry-Ki-67-EGFP, and EGFP-mCherry-Ki-67 using polyethylenimine Max (PEI; Polysciences, 24765). Following two days of culture, images were acquired with a LSM780 confocal microscope and an EC Plan-Neofluar 40×/1.30 Oil DIC M27 oil-immersion objective operated by ZEN Black. Single-cell z-stack images were acquired with EGFP channel and selected a region of the nucleolar periphery, preferentially at positions with minimal curvature in the z-dimension. Then, using zoom ×35, both EGFP and mCherry channels were acquired. Chromatic aberration was determined by acquiring images of TetraSpeck Fluorescent Microspheres with identical imaging settings as for nucleoli and measuring the average offset of intensity peaks in the $x$ and $y$ dimensions. The alignment of the red and green channels of images of nucleoli was then corrected for chromatic aberration accordingly.

## Ki-67 overexpression in HeLa cells endogenously expressing EGFP-Ki-67

A plasmid encoding EGFP-Ki-67 was transfected into cells endogenously expressing EGFP-Ki-67 with PEI in a 6-cm dish and incubated for 1 day. Cells expressing higher levels of EGFP-Ki-67 compared to the endogenous EGFP-Ki-67 were sorted into 500 µL of the culture medium in a 1.5-mL tube and seeded in LabTek slides. Following a 1-day culture, SNAP-NPM1 and DNA were labelled with 100 nM SNAP-SiR and 1× SPY555-DNA (Spirochrome, SC201) in the imaging medium for at least 2 h. Imaging was performed on a Zeiss LSM780 microscope with an EC Plan-Neofluar ×40/1.30 Oil DIC M27 oil-immersion objective using a custom-made macro in Zen Black for autofocus acquisition. To assess spectral bleed-through of signals from overexpressed EGFP-Ki-67, fluorescence was acquired sequentially using two tracks. In Track 1, the spectrally distinct EGFP and SiR signals were acquired simultaneously: EGFP was excited with a 488 nm laser and

its emission was collected from 490-570 nm, while SiR was excited with a 639 nm laser and its emission was collected from 650 to 760 nm. In Track 2, SPY555-DNA was excited with a 561 nm laser, and its emission was collected from 580-630 nm. Control experiments using cells overexpressing EGFP-Ki-67 without fluorescence labelling of SNAP-NPM1 and DNA confirmed that bleed-through into the SiR and SPY555-DNA channels was minimal (Appendix Fig. S6).

## Ki-67 overexpression in U2-OS and RPE-1 cells

For overexpression Ki-67 in U2-OS cells, a plasmid encoding Halo-Ki-67 was transfected into cells stably expressing EGFP-NPM1 with PEI in a LabTek slide. Cells treated with only the transfection mix were used as a mock control. Following 2 days of culture after transfection, Halo-Ki-67 and DNA were labelled with 200 nM Halo-JF646 (Promega, GA1120) and 1× SPY555-DNA in cell culture medium for 1 h at 37 °C. Cells were washed with the imaging medium and cultured in the imaging medium. Imaging was performed on a Zeiss LSM780 microscope with a Plan-Apochromat ×63/1.4 NA Oil DIC M27 oil-immersion objective.

For the overexpression of Ki-67 in RPE-1 cells, a plasmid encoding EGFP-Ki-67 was transfected into cells stably expressing SNAP-NPM1 with PEI in a LabTek slide. Cells treated with only the transfection mix were used as a mock control. Following 2 days of culture after transfection, SNAP-NPM1 and DNA were labelled with 100 nM SNAP-SiR and 1 x SPY555-DNA in cell culture medium for 1 h at 37 °C. Cells were washed with the imaging medium and cultured in the imaging medium. Imaging was performed on a Zeiss LSM780 microscope with a Plan-Apochromat ×63/1.4 NA Oil DIC M27 oil-immersion objective.

## Rescue the Ki-67 depletion phenotype with transient expression of Ki-67 mutants

Ki-67 mutant plasmids were transfected into Ki-67 KO cells in a 6-cm dish using PEI as described above. After culturing for 1 day, GFP-positive cells were isolated into 500 µl of the culture medium in a 1.5-mL tube using FACS. The sorting gate was determined by the EGFP levels of endogenously tagged EGFP-Ki-67 signal intensity. The sorted cells were spun down by 90×$g$ for 3 min, resuspended in the imaging medium, and then seeded to LabTek slides. Following 1-day culture, DNA was stained with 0.2 µM SiR-DNA in the imaging medium for at least 2 h. Imaging was performed on a Zeiss LSM780 microscope with an EC Plan-Neofluar ×40/1.30 Oil DIC M27 oil-immersion objective using a custom-made macro in Zen Black for autofocus acquisition.

## Auxin-induced acute Ki-67 depletion and drug washout

Cells were seeded onto a 96-well imaging plate (Cellvis, P96-1.5H-N) within the imaging medium containing 1,250 cells. Following a 1-day culture, DNA was stained with 0.2 µM SiR-DNA in the imaging medium 2 h before imaging. After equilibration of the imaging plate on the microscope stage at 37 °C for 1 h, time-lapse imaging was performed on a Zeiss LSM780 microscope with EC Plan-Neofluar ×40/1.30 Oil DIC M27 oil-immersion objective operated by Zen Black and a MyPic macro for autofocusing and acquiring multiple positions. Images were acquired with 3 z-slices by 0.525 µm intervals between slices at seven

positions per well every 30 min. For the first time point, all wells remained untreated. After completion of the first time point acquisition for all wells at around 30 min after the beginning of time-lapse imaging, cells were treated with indole-3-acetic acid (IAA; abcam, ab146402) by replacing half the medium (100 μL out of a total 200 μL per well) with imaging medium containing 500 μM IAA and 0.2 μg/ml SiR-DNA for the final concentration. For IAA non-treated cells, the 100 μL of the medium was replaced with 100 μL imaging medium only containing 0.2 μg/ml SiR-DNA for the final concentration. Around 8.5 h after the beginning of time-lapse imaging, in one of the IAA-treated wells, IAA was washed out by replacing 150 μL of the medium with fresh imaging medium 10 times. Then, the 150 μL of the medium was replaced with 150 μL of imaging medium containing 0.2 μg/ml SiR-DNA. The time lapse was stopped after 22 h. For the IAA-insensitive cells, the same time-lapse imaging procedures were applied.

## Microinjection

Live-cell microinjection experiments were performed using a FemtoJet microinjector (Eppendorf, 5181000017) in conjunction with an InjectMan NI2 micromanipulation device (Eppendorf, 5247000013). All microinjections were performed using pre-pulled Femtotips injection capillaries (Eppendorf, 5242952008). The microinjection device was directly mounted onto a customised confocal Zeiss LSM780, as described above.

Cells stably expressing H2B-mNeonGreen and SNAP-NPM1 and endogenously Halo-tagged UBF were cultured in μ-Dish 35 mm high-wall imaging dishes with a polymer bottom (ibidi, 81156) to reach 80-90% confluency on the day of the injection. SNAP-NPM1 and DNA were labelled with 100 nM SNAP-SiR and 0.2 μM SiR-DNA. AluI restriction enzyme (Thermo Fisher Scientific, FD0014) was mixed with twice the volume of micro-injection buffer (50 mM HEPES pH 7.4 (homemade), 5% glycerol (Merck, 1.04092.2511), 1 mM $Mg(OAc)_2$ (Sigma, M5661)) supplemented with 1 mg/mL CF®680R-conjugated 10k-MW dextran (VWR, 80116). Using an Eppendorf Microcapillary Microloader (Eppendorf, 5242956003), 4-6 μL diluted AluI in the microinjection buffer was loaded into a Femtotip. Microinjection was performed using 150 hPa injection pressure, 0.2 s injection time and 30 hPa compensation pressure. Images were acquired before injection, 15 min and 30 min after the injection on a Zeiss LSM780 using a Plan-Apochromat ×63/1.4 NA Oil DIC M27 oil-immersion objective. For FRAP assay of H2B after microinjection, photo-bleaching was performed in a 3 μm diameter circular region with 488 nm laser at 80% of laser power 30 min after injection. Images were acquired every 100 ms for 400 frames. The intensities were normalised using a reference region that was located away from the bleaching region within the cells, as well as a background region that was located outside the cells.

## DNA fluorescence in situ hybridisation (DNA-FISH)

FISH-probe libraries targeting human rDNA were designed using oligoMiner based on the human reference rDNA sequence (GenBank, U13369.1) and the GRCh38 human genome assembly. We used OligoMiner (Beliveau et al, 2018) with a genomic target length between 30 and 40 nt, a melting temperature between 42 and 47 °C. Selected probes were filtered by the 42 °C LDA model with 0.5 stringency and a maximum of 5 off-target 18 bp kmers. Up to

200 probes spanning the entire rDNA were selected, and a 20-nt docking sequence was added to the 5' end. The library was ordered as an oligo pool (IDT).

For the FISH procedure, cells stably expressing EGFP-NPM1 were seeded onto poly-L-lysine (PLL)-coated coverslips and cultured overnight. Cells were fixed with 3.7% formaldehyde in PBS for 15 min at room temperature (RT), quenched with 100 mM $NH_4Cl$ for 10 min, and permeabilised with 0.5% Triton X-100 for 15 min. For DNA denaturation, coverslips were treated with 0.1 M HCl for 15 min and incubated in buffer H1 (50% formamide (Sigma, F9037), 10% (w/v) dextran sulfate (Merck, S4030), 0.4 mg/ml RNase A (Thermo Fisher Scientific, EN5031) in 2x SSC (Sigma, S6639)) for 1 h at 37 °C. Primary hybridisation was performed by inverting coverslips onto a 50 μL drop of buffer H1 containing the primary oligo pool (~2 nM per probe) on a clean glass slide. The assembly was sealed, denatured at 75 °C for 5 min, and incubated overnight at 42 °C in a humidified chamber. Following primary hybridisation, coverslips were washed twice in 50% formamide/2× SSC at 37 °C, followed by washes in 0.2% Tween-20/2× SSC and 2× SSC. Secondary hybridisation was then carried out for 2 h at 30 °C in buffer H2 (25% formamide, 10% (w/v) dextran sulfate, 0.1% Tween-20 in 2× SSC) containing secondary imager probes (20 nM). Finally, coverslips underwent washes in 25% formamide/2× SSC, 0.2% Tween-20/2× SSC, and 2× SSC. DNA was counterstained with DAPI (1:5000 in PBS) for 5 min.

Images were acquired on an Olympus IXplore SpinSR micro-scope with a UPLSAPO100XS objective and an incubator chamber (EMBL, Heidelberg, Germany) maintaining 37 °C and 5% $CO_2$. Images were subsequently deconvoluted using Huygens software. The Pearson correlation coefficient was analysed within nucleoli, which were segmented based on NPM1 signals in CellProfiler, and data were visualised in R.

## Data analysis

### Image analysis of RNAi screen

Image analysis was performed using CellProfiler software (McQuin et al, 2018; Stirling et al, 2021). For the nucleolar shape screen (Fig. 1), the nuclei and the nucleoli were segmented by local adaptive thresholding. Using a control siRNA and siRNAs whose knockdown phenotypes are well-known (siRNAs for INCENP, KIF11, PLK1, and CDC20), chromatin morphologies were analysed to assess the RNAi efficiency and specificity. These supervised datasets were used for an automatic classification method (Support Vector Machine with Radial Basis Function Kernel) to classify chromatin morphology in all images. Classification results were overlaid on images for quality control and identified mononu-cleated interphase cells with high confidence. The genes targeted by siRNAs show that more than 20% of apoptotic cells were excluded from the following analysis. The median nucleolar aspect ratio was calculated in each well, followed by averaging those values for the same target genes except for the control siRNA. For the control siRNA, the median value of the nucleolar aspect ratio in each control well was used. Protein functions of the top 20 hit proteins were manually annotated based on a review of literature (Phipps et al, 2011), UniProt database (UniProt Consortium, 2025) and Complex portal (Balu et al, 2025).

For the analysis of the follow-up screen, we applied the same quality control methods and excluded cells other than interphase

cells in the following analysis. For the nucleolar chromatin analysis (Fig. 8A,B and EV8A), following segmentation of the nucleolus based on NPM1 signals, the segmentation was shrunk and expanded in arbitrary pixels to generate the nucleolar interior segmentation and the expanded nucleolus segmentation. The nucleolar rim segmentation was produced by subtracting the shrunk nucleolus segmentation from the expanded nucleolus segmentation. For the nucleoplasm segmentation, the expanded nucleolus segmentation was subtracted from the nucleus segmentation. To quantify the chromatin levels, H2B mean intensity was measured in each segmentation, followed by averaging those values per nucleus for the same target gene.

### Image analysis of confocal microscope images

If raw images contained a z-stack, a single slice was selected based on the highest mean intensity of the nucleolar marker fluorescence signals across slices. The single-slice images were analysed using CellProfiler ver. 3.1.8 (McQuin et al, 2018) or ver. 4.2.5 (Stirling et al, 2021). The nuclei and the nucleoli were segmented based on fluorescence signal intensity using an adaptive Otsu thresholding method. For experiments performed in RPE-1 and U2-OS cells, segmented nuclei exhibiting abnormal size or shape (defined as mean ± 2 SD of nuclear area and circularity across the dataset) and cells with insufficient fluorescence signal of nucleolar marker proteins were excluded from further analysis. Based on the segmented nucleoli objects, the nucleolar interior segmentations were generated by shrinking arbitrary pixels from the nucleoli objects to exclude perinucleolar chromatin. The nucleolar rim segmentations were generated by expanding arbitrary pixels from nucleoli objects to cover the perinucleolar chromatin and then subtracting the nucleolar interior masks. The nucleoplasm segmentation was generated by subtracting the expanded nucleoli masks from the nuclei mask. The nucleoli objects were linked to the parental nucleus object to calculate the per nucleus values. Fluorescence signal intensity within the objects and the object size and shape were measured. The data were further analysed in R. Using the relationship between the parental nucleus and the children's nucleoli, the average mean intensity of the nucleoli per nucleus was calculated. For the relative intensity measurements in the nucleolar interior and in the nucleolar rim, the average values of mean intensities in these segmentations were divided by the mean intensity in the nucleoplasm of the parental nucleus. For the nucleolar area, the summed nucleolar area in the parental nucleus was calculated. For the nucleolar aspect ratio and circularity, the median value was calculated per the parental nucleus. The circularity was calculated as $4 * \pi * area/perimeter^2$.

To segment FC regions, UBF signals acquired by the spinning disk microscope were analysed using Pixel Classification in ilastik (Berg et al, 2019). The classifier was manually trained on ten images and then used for batch processing of the whole datasets to generate probability maps. Then, in CellProfiler, the FC regions were segmented based on the probability map with a 0.5 threshold. The segmented FC regions were linked to their parental nucleolus and nucleus, followed by the measurement of the median FC area and the FC density (the number of FC divided by the nucleolar area) per nucleolus in each nucleus.

### Radial signal distribution analysis

For the Airyscan imaging (Fig. 2G,H), segmentations of nucleoli and nucleus were performed as described above. To segment chromatin foci in the nucleolar interior, the first nucleolar sub-region mask was generated by eroding 20 pixels from the nucleoli masks. Then, upper quantile values of the DNA intensity were measured within the nucleolar sub-region. The upper quantile values were subtracted from the raw DNA signal intensity in the image. After applying the Gaussian filter on this subtracted image, chromatin foci within the nucleolar sub-region were segmented by the adaptive Otsu method. The segmented chromatin foci were further filtered with their area and circularity above 15 pixels and above 0.7, respectively. Using a module called MeasureObjectIntensityDistribution, the expanded nucleolar mask was radially separated into 50 concentric rings from the chromatin foci in the nucleolar interior towards the edge of the expanded nucleolus, and the mean intensity was measured within each ring. The rings were divided into two regions: the chromatin foci (ring 1–15) and the nucleolar periphery (ring 16–50). In each nucleolus, normalised intensities by min–max scaling were calculated in R.

To measure radial signal distribution from the centre to the periphery of the nucleolus (Appendix Fig. S4B,C), the expanded nucleolar mask was radially segmented into 10 sectors from the centre to the edge of the expanded nucleolar mask. In each nucleolus, normalised intensities by min–max scaling were calculated in R.

### Segmentation of nucleoli from holotomographic imaging data

From 3D holotomography images of refractive index, a centre single slice with in-focus nucleoli was manually chosen and isolated. Background pixels and pixels belonging to nucleoli were manually annotated and trained in ilastik. Based on this training data, a neural network was developed by Akhmedkhan Shabanov (Preprint: Shabanov et al, 2021) and applied to segment nucleoli in the whole datasets. Nucleolar aspect ratio was measured in Fiji and analysed in R.

### Analysis of Ki-67 extension

To measure molecular extension, the fluorescence intensity of EGFP and mCherry signals was measured along line profiles of 2 pixels width drawn across the nucleolar periphery from the exterior to the interior across the nucleolar periphery in Fiji. The resulting intensity profiles were analysed in R. For each profile, background correction was performed by subtracting the minimum intensity value of each respective channel, EGFP and mCherry, from its raw data. These background-subtracted intensities for each channel were then normalised by dividing each value by the sum of all background-subtracted intensities within that channel for the profile. Subsequently, a single Gaussian function was fitted to each normalised channel profile to determine the Gaussian mean position. Data with badly fitted Gauss distributions were manually discarded after visual inspection of the plots. The displacement distance between EGFP and mCherry was then calculated as the difference between their respective Gaussian means.

### Colocalisation of Ki-67 and FBL or UBF

The nuclei and the nucleoli were segmented as described above. Pearson correlation coefficient of fluorescence signals between Ki-67 and UBF, FBL, or NPM1 in the nucleoli was measured in CellProfiler. The data was further analysed in R.

### Statistical analysis

No statistical methods were used to predetermine sample size. All experiments were repeated several times, and the indicated experiment numbers always refer to biological replicates. No formal blinding was performed, but to minimise potential human bias, most experiments were analysed automatically using image analysis software. Data were tested for normality and homogeneity of variance with Shapiro–Wilk and Levene's tests ($\alpha = 0.05$), respectively. For comparisons between two groups, the appropriate statistical test was chosen as follows: unpaired normally distributed data were analysed using a two-tailed Student's $t$ test (for equal variances) or a two-tailed Welch's $t$ test (for unequal variances). Unpaired non-normally distributed data were analysed using a two-tailed Mann–Whitney test $U$ test (for equal variances) or with a two-tailed Kolmogorov–Smirnov test (for unequal variances). For comparisons among multiple groups of normally distributed data with homogeneous variances, one-way analysis of variance (ANOVA) was performed. For comparisons among multiple groups of non-normally distributed data, a Kruskal–Wallis test followed by two-sided Dunn's multiple-comparison test was used. All statistical analyses were performed using R or GraphPad Prism.

For experiments involving Ki-67 depletion or overexpression, cells were excluded if they exhibited abnormal nuclear size or shape. Specifically, nuclei were retained if their area and circularity fell within mean $\pm 2$ SD of the overall population. In addition, cells with insufficient fluorescence signal of nucleolar marker proteins (e.g. NPM1) were excluded based on manually defined intensity thresholds to ensure reliable segmentation and quantification.

### Declaration of generative AI and AI-assisted technologies in the writing process

During the preparation of this work, the authors used ChatGPT and Claude in order to enhance clarity, improve readability and eliminate redundancy. After using this tool/service, the authors reviewed and edited the content as needed and take full responsibility for the content of the publication.

## Data availability

An earlier version of this manuscript was deposited on bioRxiv on 2025-07-24 (https://doi.org/10.1101/2025.07.23.666339). The source data of this paper will be deposited in BioStudies. Materials generated in this study will be made available on request upon completion of a Materials Transfer Agreement.

The source data of this paper are collected in the following database record: biostudies:S-SCDT-10_1038-S44318-026-00747-7.

## Peer review information

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

## Acknowledgements

We thank Daniel W. Gerlich for providing cell lines, the EMBL Advanced Light Microscopy Facility (ALMF) for imaging support, the Flow Cytometry Core Facility for cell sorting, Christian Tischer and Jean-Karim Hériche for assistance with the custom image analysis, and Kai S Beckwith, Natalia R Morero, and Jan Ellenberg for supporting an oligo probe design for DNA-FISH. This work was supported by the German Research Foundation (DFG project number 402723784) and the Human Frontier Science Program (CDA00045/2019). AH-A has received a PhD fellowship from the Boehringer Ingelheim Fonds; YH was supported by a fellowship from the EMBL interdisciplinary Postdoc (EIPOD) program (Marie Sklodowska-Curie Actions, COFUND grant agreement 664726).

## Author contributions

**Daja Schichler**: Conceptualisation; Supervision; Investigation; Writing—original draft; Writing—review and editing. **Yuki Hayashi**: Conceptualisation; Supervision; Funding acquisition; Investigation; Writing—original draft; Writing—review and editing. **Letitia Fernandez**: Investigation. **Mariam Chupanova**: Investigation. **Alberto Hernandez-Armendariz**: Funding acquisition; Investigation; Writing—review and editing. **Beate Neumann**: Methodology. **Sara Cuylen-Haering**: Conceptualisation; Supervision; Funding acquisition; Writing—original draft; Project administration; Writing—review and editing.

Source data underlying figure panels in this paper may have individual authorship assigned. Where available, figure panel/source data authorship is listed in the following database record: biostudies:S-SCDT-10_1038-S44318-026-00747-7.

## Funding

## Disclosure and competing interests statement

DS is now an AstraZeneca employee.

# Expanded View Figures

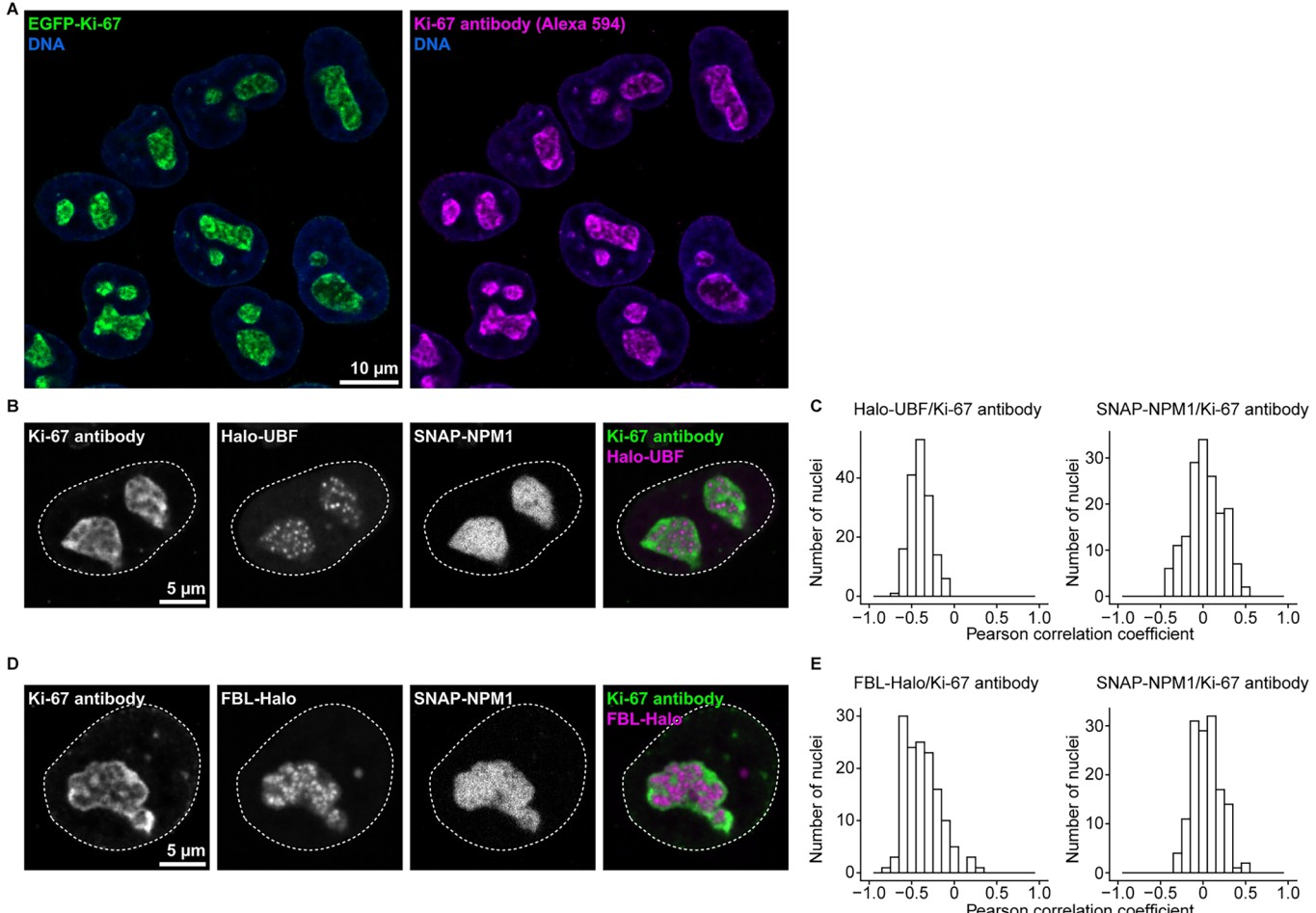

**Figure EV1.** **Ki-67 displays a distinct localisation pattern in the nucleolus, separate from nucleolar subcompartments, related to Fig. 2.**

(A) Immunofluorescence staining of Ki-67 in an endogenous EGFP-Ki-67 cell line. DNA was stained with DAPI. Notably, endogenous EGFP-Ki-67 signals substantially overlap with signals of anti-Ki-67 antibody. (B) Co-staining of Ki-67 and UBF. After labelling cells expressing SNAP-NPM1 and Halo-UBF with SNAP-SiR and Halo-TMR in live cells, immunofluorescence was performed for Ki-67 as in (A). Dashed lines indicate the nuclear boundary. (C) Colocalisation analysis between Ki-67 and UBF or NPM1 within the nucleolus. Pearson correlation coefficient between Ki-67 and UBF signals (left panel) or Ki-67 and NPM1 signals (right panel) was measured within nucleoli (segmented based on NPM1 signal). A Pearson correlation coefficient of +1.0 indicates a perfect positive linear correlation, −1.0 indicates a perfect negative correlation, and 0 indicates no correlation between the signals. (D) Co-staining of Ki-67 and FBL. After labelling cells expressing SNAP-NPM1 and FBL-Halo with SNAP-SiR and Halo-TMR in live cells, immunofluorescence was performed for Ki-67 as in (A). Dashed lines indicate the nuclear boundary. (E) Colocalisation analysis between Ki-67 and FBL or NPM1 within the nucleolus. Pearson correlation coefficient between Ki-67 and FBL signals (left panel) or Ki-67 and NPM1 signals (right panel) was measured within nucleoli, segmented based on NPM1 signal as in (C). For (C, E), $n = 144$ nucleoli and $n = 165$ nucleoli, respectively. 2 biological replicates. Source data are available online for this figure.

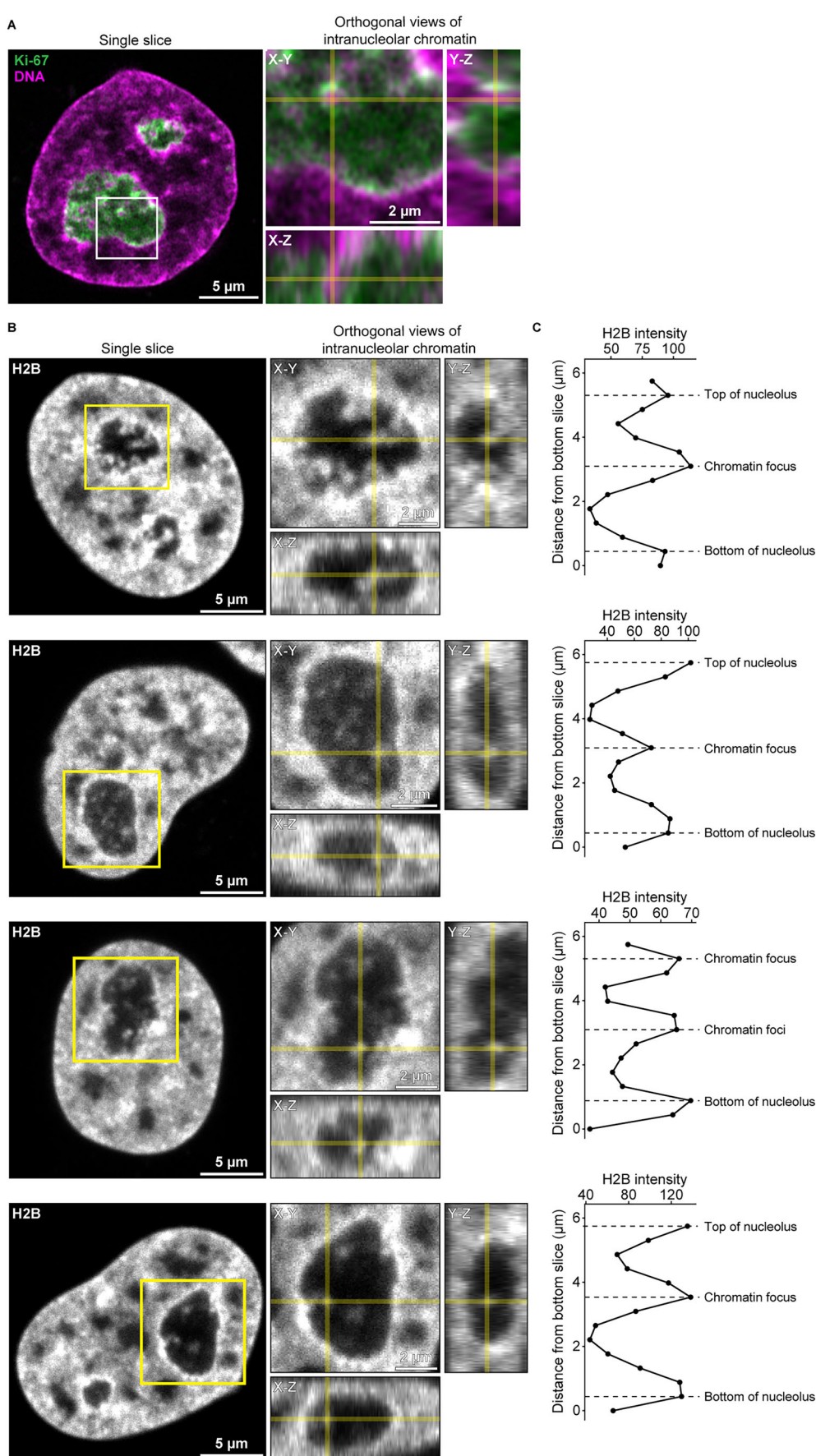

◀  **Figure EV2.  Chromatin invagination into the nucleolus, related to Fig. 2.**

(**A**) 3D confocal stack of HeLa cells endogenously tagged with EGFP-Ki-67 and DNA labelled with SPY555-DNA from Fig. 2A,F. Insets show orthogonal views at the chromatin focus analysed in Fig. 2F. (**B**) 3D confocal stack of HeLa cells expressing H2B-mCherry. Insets show nucleolar chromatin foci, which orthogonal views confirm to arise from chromatin invaginations. A yellow cross marks a representative focus. (**C**) Chromatin signal along the *z* axis at the chromatin foci within the nucleolus. H2B intensity profile measured through the centre of the yellow cross in (**B**). Source data are available online for this figure.

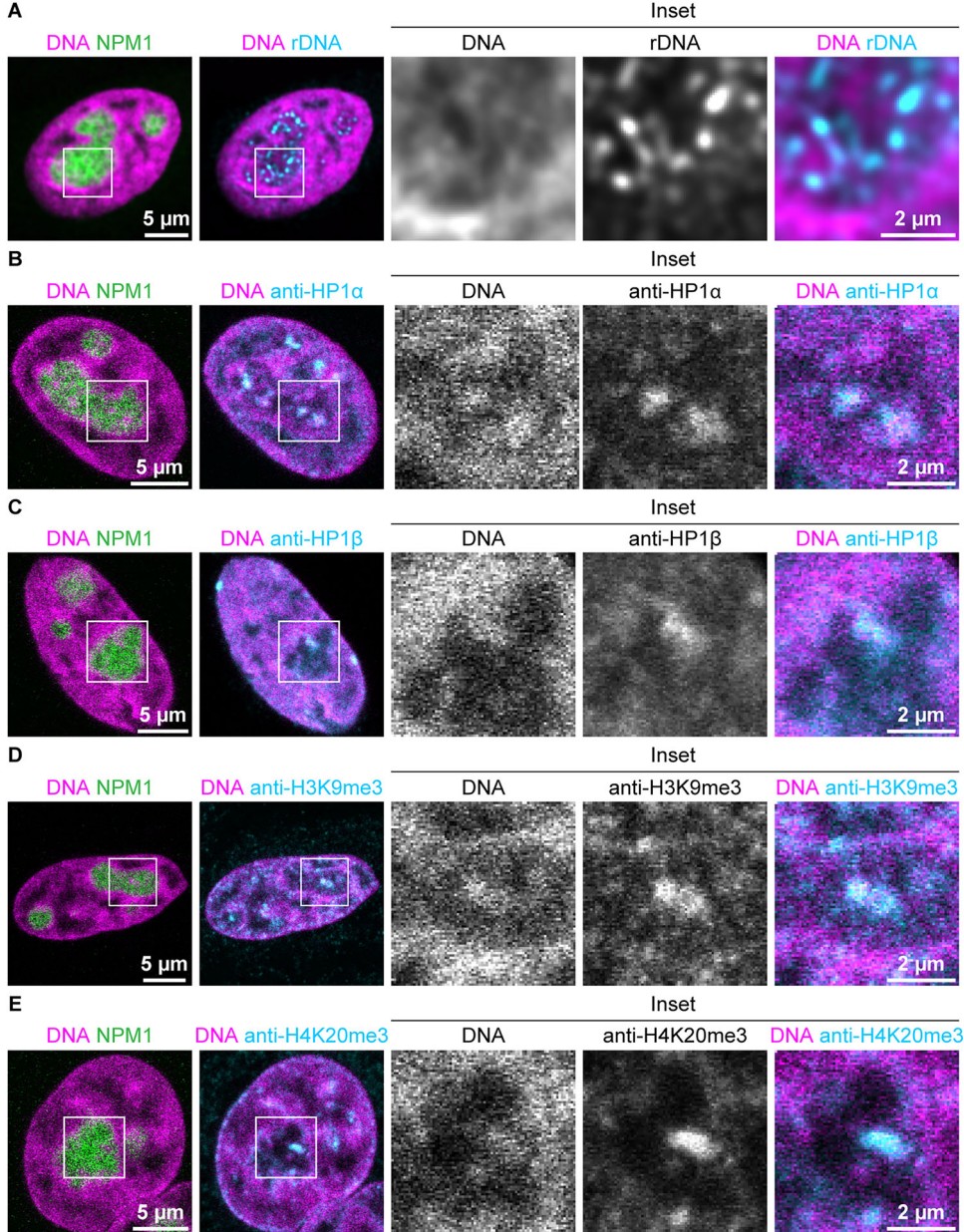

**Figure EV3.   Intranucleolar chromatin co-localises with heterochromatin markers, but not with rDNA, related to Fig. 2.**

(**A**) Distinct localisation of rDNA and chromatin in the nucleolus. DNA-FISH against rDNA was performed in HeLa cells. DNA was stained with DAPI. (**B–E**) Colocalisation of intranucleolar chromatin with heterochromatin markers. Immunostaining for HP1α, HP1β, H3K9me3, and H4K20me3 was performed in HeLa cells. DNA was stained with DAPI. Two biological replicates were performed. Source data are available online for this figure.

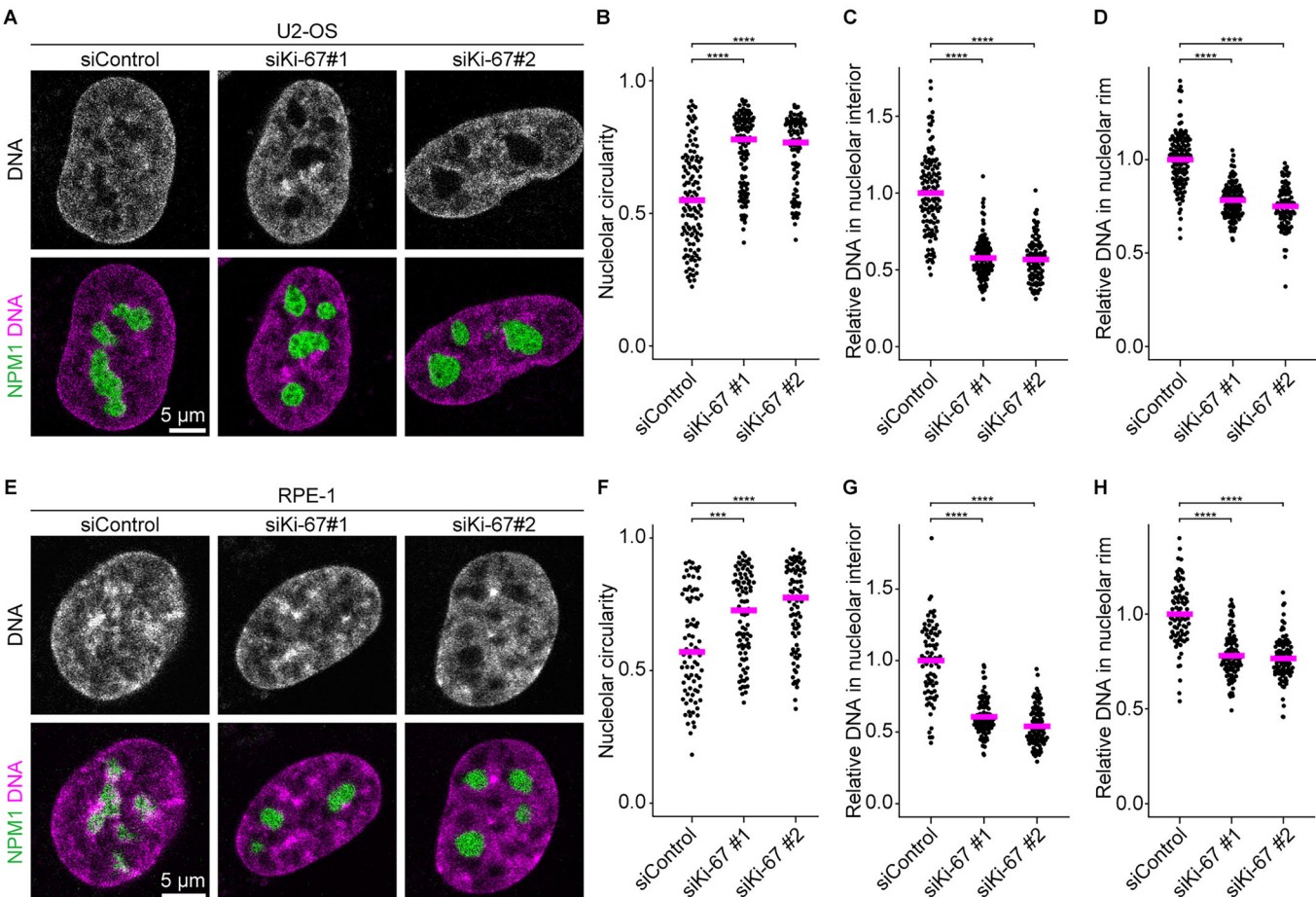

**Figure EV4. Ki-67 depletion induces nucleolar rounding and chromatin removal from nucleoli in U2-OS and RPE-1 cells, related to Fig. 3.**

(A) Live-cell imaging of U2-OS cells stably expressing EGFP-NPM1 following Ki-67 depletion. Cells were transfected with a non-targeting control siRNA (siControl) or two different Ki-67 siRNAs, followed by confocal imaging 48 h post-transfection. DNA was stained with SPY555-DNA before imaging. A single z-slice is shown. (B) Quantification of nucleolar roundness. Nucleoli were segmented based on NPM1 signals to measure their circularity. Circularity values of the largest nucleolus per nucleus are shown. Bars indicate median values. Statistical comparisons were performed against the siControl sample: siKi-67 #1, $P = 4.69 \times 10^{-13}$; siKi-67 #2, $P = 8.99 \times 10^{-11}$. (C, D) Quantification of chromatin enrichment in the nucleolar interior and its rim. Relative DNA signal intensities were calculated as Fig. 3C,D for the nucleolar interior (C) or the nucleolar rim (D). Bars indicate mean values. Statistical comparisons were performed against the siControl sample: For (C), siKi-67 #1, $P = 7.73 \times 10^{-29}$; siKi-67 #2, $P = 7.73 \times 10^{-29}$. For (D), siKi-67 #1, $P = 1.50 \times 10^{-23}$; siKi-67 #2, $P = 8.99 \times 10^{-30}$. (E) Live-cell imaging of RPE-1 cells stably expressing SNAP-NPM1 following Ki-67 depletion. Cells were transfected with a non-targeting control siRNA (siControl) or two different Ki-67 siRNAs, followed by confocal imaging 48 h post-transfection. SNAP-NPM1 and DNA were stained with SiR-SNAP and SPY555-DNA before imaging. A single z-slice is shown. (F) Quantification of nucleolar roundness. Nucleoli were segmented based on NPM1 signals to measure their circularity. Circularity values of the largest nucleolus per nucleus are shown. Bars indicate median values. Statistical comparisons were performed against the siControl sample: siKi-67 #1, $P = 2.32 \times 10^{-4}$; siKi-67 #2, $P = 1.08 \times 10^{-6}$. (G, H) Quantification of chromatin enrichment in the nucleolar interior and its rim. Relative DNA signal intensities were calculated as Fig. 3C,D for the nucleolar interior (G) or the nucleolar rim (H). Bars indicate mean values. Statistical comparisons were performed against the siControl sample: For (G), siKi-67 #1, $P = 7.35 \times 10^{-16}$; siKi-67 #2, $P = 8.10 \times 10^{-26}$. For (H), siKi-67 #1, $P = 6.11 \times 10^{-16}$; siKi-67 #2, $P = 4.30 \times 10^{-18}$. For (B–D), n = 122 nuclei (siControl), 116 nuclei (siKi-67#1), 92 nuclei (siKi-67#2), 2 biological replicates. For (F–H), n = 82 nuclei (siControl), 90 nuclei (siKi-67#1), 88 nuclei (siKi-67#2), 2 biological replicates. Statistical tests were performed with the Kruskal–Wallis test followed by Dunn's test, ns (not significant) $P > 0.05$, ***$P < 0.001$, ****$P < 0.0001$. Source data are available online for this figure.

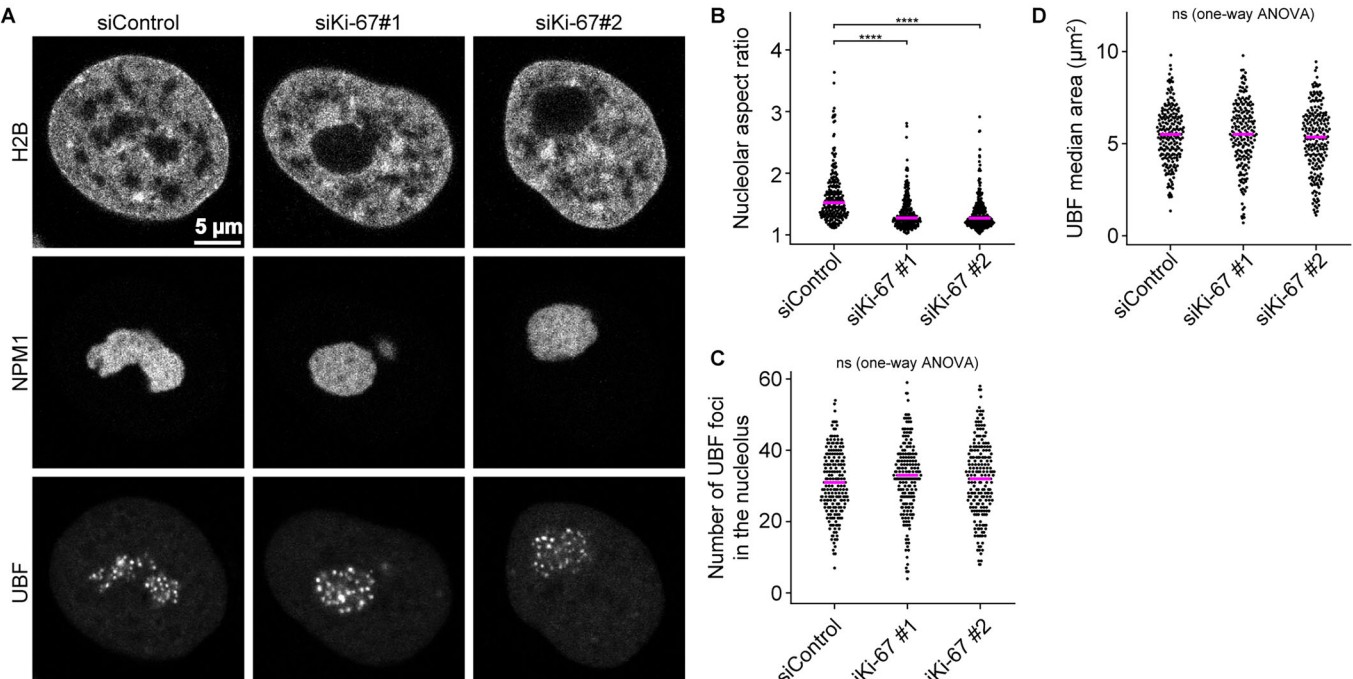

**Figure EV5. Ki-67 depletion does not disrupt internal nucleolar subcompartments, related to Fig. 3.**

(A) Effects of Ki-67 depletion on nucleolar subcompartments. HeLa cells expressing endogenously tagged Halo-UBF and stably overexpressing SNAP-NPM1 and H2B-mNeonGreen were transfected with a non-targeting control siRNA (siControl) or Ki-67 siRNAs. After 72 h of transfection, Halo-UBF and SNAP-SNAP were labelled with Halo-TMR and SNAP-SiR, respectively. Images were acquired using spinning disk microscopy. (B) Quantification of the nucleolar shape. Nucleoli were segmented based on NPM1 signals, and their aspect ratio was measured. Bars indicate the median. Statistical comparisons were performed against the siControl sample: siKi-67 #1, $P = 4.16 \times 10^{-16}$; siKi-67 #2, $P = 6.91 \times 10^{-17}$. Statistical tests were performed with the Kruskal–Wallis test followed by Dunn's test, ****$P < 0.0001$. (C, D) Quantification of the size and number of internal UBF subcompartments based on segmented UBF signals. Median area per the nucleus and the total number of subcompartments were measured. Bars indicate median values. No significant differences were detected among samples (one-way ANOVA, $P = 0.28$). For (B–D), $n = 207$ nuclei (siControl), 194 nuclei (siKi-67#1), 214 nuclei (siKi-67#2), 2 biological replicates. Source data are available online for this figure.

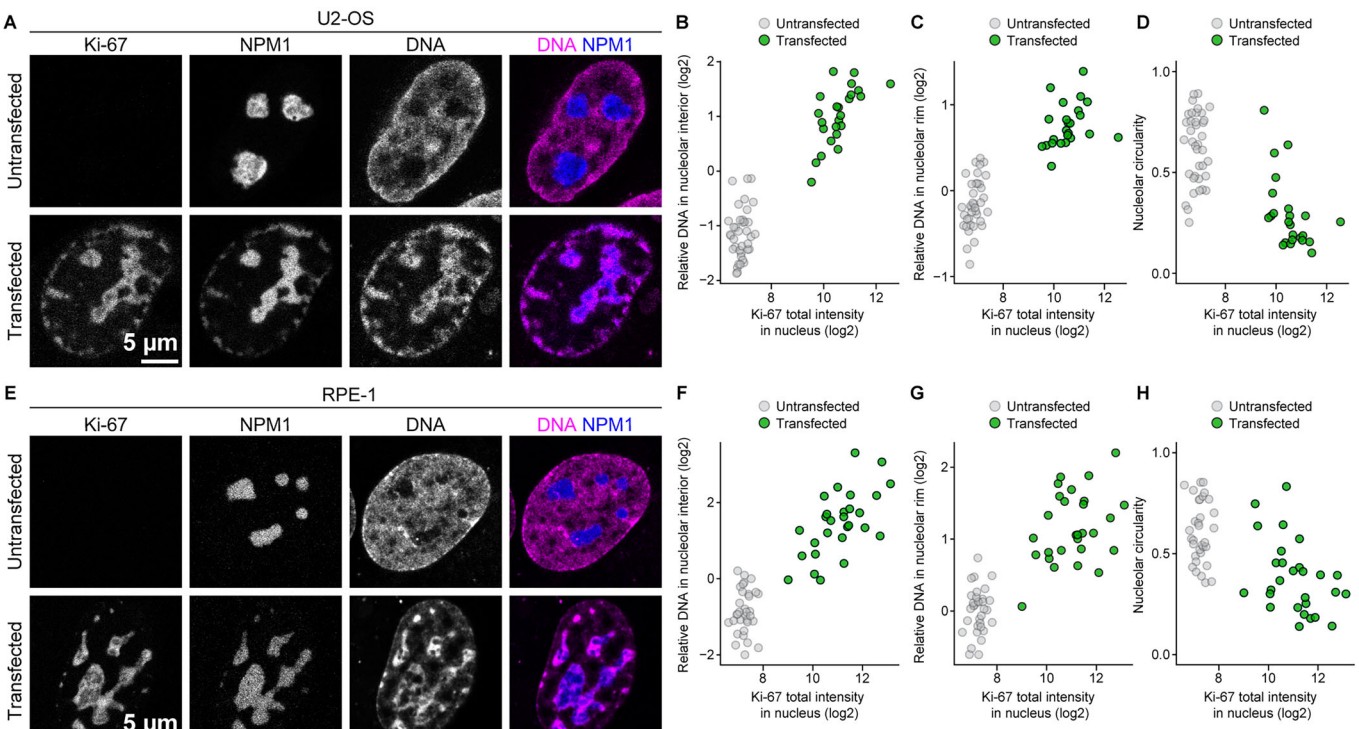

**Figure EV6. Ki-67 overexpression induces irregularly shaped nucleoli and excess chromatin loading into nucleoli in U2-OS and RPE-1 cells, related to Fig. 4.**

(A) Live-cell imaging of Ki-67-overexpressing cells in U2-OS cells. Cells stably expressing EGFP-NPM1 were transfected with a Halo-Ki-67 plasmid. Halo-Ki-67 was labelled with Halo-JF646, and DNA was stained with SPY555-DNA. (B, C) Ki-67 expression level-dependent chromatin enrichment in the nucleolar interior (B) and rim (C). Relative DNA signal intensities, calculated as described in Fig. 3D,E, are plotted against total nuclear Halo-Ki-67 intensity. (D) Ki-67 expression level-dependent increase in the irregularity of nucleolar shape. Median circularity of segmented nucleoli from NPM1 signals per nucleus is plotted against the total intensity of Halo-Ki-67 in the nucleus. (E) Live-cell imaging of Ki-67-overexpressing cells in RPE-1 cells. Cells stably expressing SNAP-NPM1 were transfected with an EGFP-Ki-67 plasmid. DNA was stained with SPY555-DNA. (F, G) Ki-67 expression level-dependent chromatin enrichment in the nucleolar interior (F) and its rim (G). Relative DNA signal intensities, calculated as described in Fig. 3D,E, are plotted against total nuclear EGFP-Ki-67 intensity. (H) Ki-67 expression level-dependent increase in the irregularity of nucleolar shape. Median circularity of segmented nucleoli from NPM1 signals per nucleus is plotted against the total intensity of EGFP-Ki-67 in the nucleus. For (B–D), $n = 38$ nuclei (Untransfected); $n = 24$ nuclei (Transfected), 2 biological replicates. For (F–H), $n = 33$ nuclei (Untransfected); $n = 28$ nuclei (Transfected), 3 biological replicates. Source data are available online for this figure.

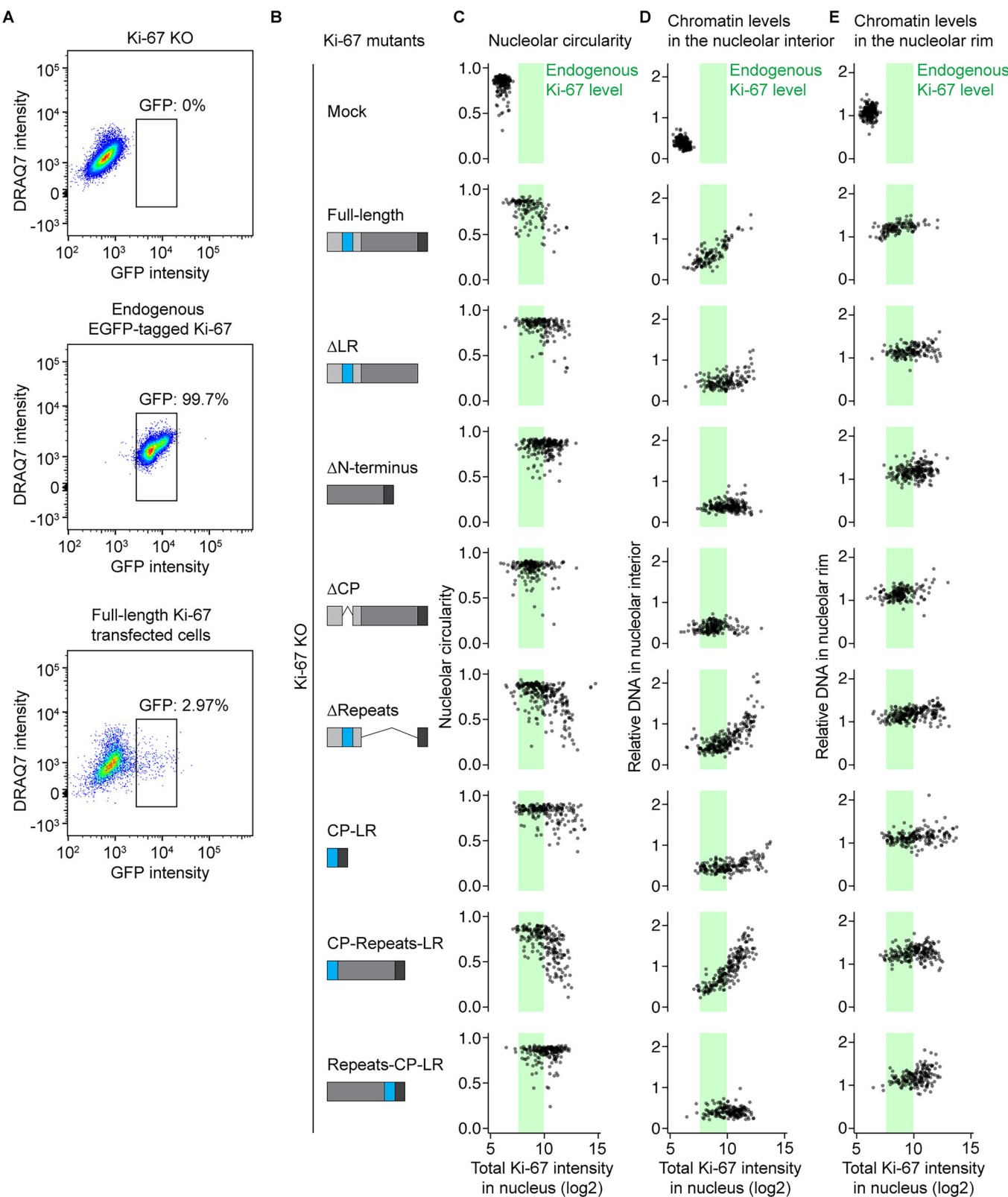

**Figure EV7. The effect of Ki-67 mutant expression levels on nucleolar shape and chromatin levels within nucleoli and their rim, related to Fig. 6.**

(A) Isolation of cells transfected with Ki-67 mutants at the endogenous Ki-67 levels by FACS. Ki-67 KO cells (left) and endogenously EGFP-Ki-67-expressing cells (middle) were used as a reference for FACS sorting. (B) Schematic of Ki-67 domain mutants. (C) Ki-67 expression level-dependent increase in the nucleolar irregularity. Median circularity of nucleoli based on NPM1 segmentation per nucleus is plotted against the total intensity of EGFP-Ki-67 in the nucleus. Green rectangles show the wild-type Ki-67 expression range determined by imaging of endogenously tagged EGFP-Ki-67 cells. (D, E) Ki-67 expression level-dependent chromatin enrichment in the nucleolar interior (D) and rim (E). Relative DNA signal intensities, calculated as described in Fig. 3D,E, are plotted against total EGFP-Ki-67 intensity in the nucleus. Green rectangles show wild-type Ki-67 expression. For (C–E), $n = 220$ nuclei (mock), $n = 126$ nuclei (full-length), $n = 162$ nuclei (ΔLR), $n = 228$ nuclei (ΔN-terminus), $n = 181$ nuclei (ΔCP), $n = 128$ nuclei (LR), $n = 250$ nuclei (ΔRepeats), $n = 189$ nuclei (CP-LR), $n = 212$ nuclei (CP-Repeats-LR), $n = 183$ nuclei (Repeats-CP-LR), 3 biological replicates. Source data are available online for this figure.

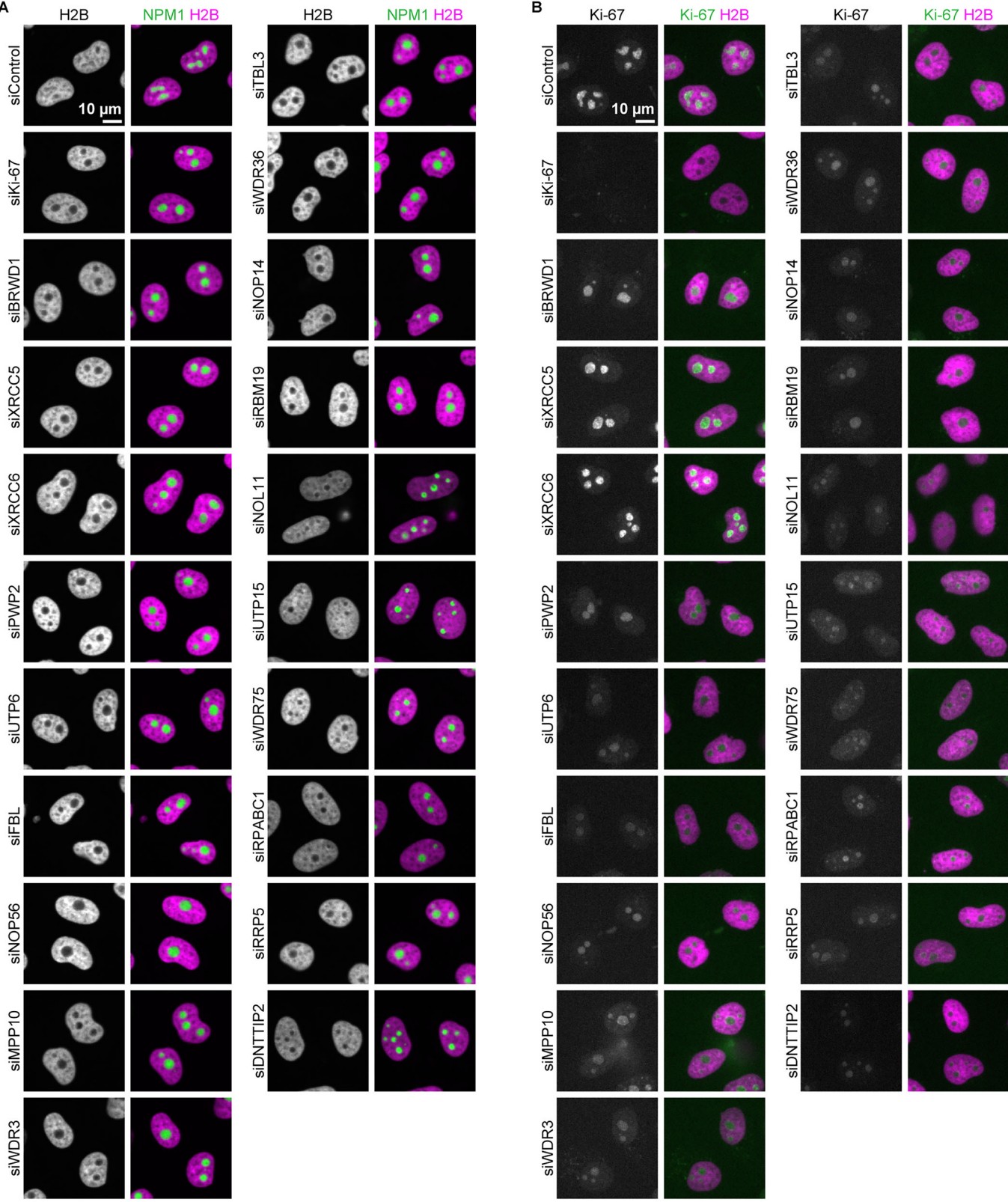

◀ **Figure EV8.** **Chromatin enrichment and Ki-67 expression in cells upon siRNA depletion of the top 20 candidate genes, inducing nucleolar rounding, related to Fig. 8.**

(A) Live-cell imaging of chromatin enrichment in the nucleolus following depletion of the top 20 candidate proteins. Cells expressing NPM1-EGFP and H2B-mCherry were transfected with a non-targeting control siRNA (siControl) or siRNAs targeting the top 20 candidate genes (Fig. 1B). Images were acquired 72 h after siRNA transfection. (B) Live-cell imaging of endogenous EGFP-Ki-67 following depletion of the top 20 candidate proteins. Cells endogenously tagged EGFP-Ki-67 and stably expressing SNAP-NPM1 and H2B-mCherry were transfected with a siControl or siRNAs targeting the top 20 candidate genes (Fig. 1B). Images were acquired 72 h after siRNA transfection. Source data are available online for this figure.

