## [Peer Review File · The EMBO Journal]

Ki-67 shapes the nucleolus by anchoring chromatin via its amphiphilic properties

Daja Schichler, Yuki Hayashi, Letitia Fernandez, Mariam Chupanova, Alberto Hernandez-Armendariz, Beate Neumann, and Sara Cuylen-Häring

Corresponding author(s): Sara Cuylen-Häring (sara.cuylen-haering@embl.de)

Review Timeline:

Submission Date:	21st Jul 25
Pre-consultation Date:	2nd Sep 25
Revision Plan Submission:	12th Sep 25
Editorial Decision:	10th Oct 25
Revision Received:	8th Jan 26
Editorial Decision:	23rd Feb 26
Revision Received:	26th Feb 26
Accepted:	27th Feb 26

Editor: Hartmut Vodermaier

Transaction Report:

Dear Dr. Cuylen-Haering,

Thank you for submitting your manuscript on Ki-67 roles in nucleolar shaping to The EMBO Journal. I am sorry that it has taken longer than usual to evaluate it, mainly due to the summer vacation period. We have now received feedback from three referees with expertise in membrane less organelles, nucleolar biology, and chromatin organization, which I am copying below for your information. As you will see, these referees are rather divided in their opinions, but only referee 3 is currently strongly supportive of publication with limited amounts of revision. On the other hand, referee 2 finds the novelty of the study somewhat too limited in the absence of deeper insights into functional consequences; while referee 1 is generally interested but raising several substantive concerns regarding the conclusiveness of key aspects of the study.

Faced with these divergent recommendations and criticisms, it would be helpful to hear if and how they could be experimentally addressed, before taking a final decision in this case. I am therefore inviting you to carefully consider the attached reports together with your coworkers, and to send me a tentative point-by-point response, ideally by early next week. Based on such a revision plan, which I would be happy to further discuss directly with you, we could then determine whether a major revision for The EMBO Journal would appear feasible/promising, or whether a less substantively revised/toned-down version might alternatively warrant publication in one of our sister journals such as EMBO Reports.

Looking forward to hearing from you,

With kind regards,

Hartmut

Referee #1 (Report for Author)

In Schichler, Hayashi et al., the authors define proliferation marker Ki-67 as one of the proteins regulating nucleolar shape in proliferating cells from a high-throughput screen of nucleolar shape. Using auxin-inducible degron and dose-controlled overexpression, the

authors show a clear causal link between Ki-67 and nucleolar shape. Following their previous works, they further establish that the amphiphilic properties of Ki-67 are necessary for its correct interphase localization. This is notable as it is unclear which proteins and mechanisms anchor perinucleolar chromatin, despite nucleolar-associated domains (NAD) being described for many years. This paper brings new knowledge on the potential mechanisms of NAD tethering to the nucleolus by Ki-67. Strikingly, the authors also suggest the presence of heterochromatin within the nucleolar interior, a notion that is both confusing due to the long history of DNA being difficult to identify in the nucleolus in EM and alarming regarding data interpretation. Overall, the manuscript will be of high interest given the well-established connection between nucleolar shape and human physiology, disease, and the general focus of elucidating the determinants of condensate form and function in cells. Currently, the role of Ki-67 and nucleolar shape is well established in this study; however, multiple problems in the manuscript remain, such as the notion that Ki-67 brings chromatin into the interior of the nucleolus, along with others discussed below. Thus, I am not in support of the manuscript being published in its current form.

1. The detection of chromatin signals within the nucleolus is intriguing but also raises concerns about potential imaging artifacts. Are the authors certain that the measured signals originate from within the GC, rather than from Z-bleed of chromatin located above or below the focal plane, given the irregular shape of nucleoli? While the observed differences between conditions are striking, Z-bleed artifacts would likely be less prominent in highly spherical nucleoli than in complex or multipart structures. A full 3D stack of the nucleus would be needed to prove whether these DNA signals are truly coming from the interior of the nucleolus (ideally with and without 3D deconvolution). Alternatively, these DNA and Ki-67 signals might represent chromatin invaginations into the nucleolus that are captured in the given optical section.

2. Related to the detection of chromatin within the nucleolar interior, the manuscript lacks sufficient discussion and literature context to support the interpretation of these findings. What types of heterochromatic DNA sequences do the authors propose could be enriched inside the nucleolus? The discussion currently cites Ki-67 interacting with heterochromatic, late-replicating sequences at the nucleolar rim. While this explains the presence of Ki-67 at the periphery, it does not account for the co-localization of both Ki-67 and DNA within the nucleolar interior. To our knowledge, the only DNA normally found inside the nucleolus is rDNA, which is already challenging to visualize even by EM, and truly heterochromatic sequences are not reported in the nucleolar interior. Indeed, the notion is perplexing based on the overwhelming difficulty of identifying DNA in EM images of

nucleoli, which has led to the location of pre-rRNA transcription being controversial (PMCID: PMC2173404) - why do the authors think this was missed? As it stands, the claim about chromatin present in the nucleolar interior is unsupported.

3. Since this study is largely based on quantitative microscopy, the authors should verify three things regarding possible imaging artifacts: (a) Axial (Z) bleed-through, which can cause peripheral chromatin signals to appear within the nucleolus in 2D sections, particularly for irregularly shaped or multipart nucleoli. (b) Detector linearity should be verified for quantitative measurements. Notably, the detector is a PMT, which has difficulty imaging more than 10-fold ranges of overexpression, and thus, there is concern that the 32-fold differences may be outside of the linear range. Alternatively (and ideally), the users may have made a conversion table from one laser power to another, allowing for a higher dynamic range. (c) The authors need to confirm if there is spectral bleed-through, which could lead to apparent signal overlap between channels. Furthermore, it is not evident in the methods if the imaging is done sequentially or simultaneously, and thus, there may be substantial (if simultaneous) bleed-through from overexpressed Ki67 into the DNA channels (Figure 3). Additionally, sequentially acquired channels can have bleed at high overexpression levels. For both, please specify whether detector linearity and spectral separation were validated under the same imaging conditions. The method section currently does not describe this or the details for acquisition parameters, such as the filters that were used.

4. A recent study - Riback et al Mol Cell 2023 - suggests that nucleolar activity (locations of FCs) maintains the non-spherical shape of nucleoli and rules out bulk chromatin mechanics employing lamin and hp1alpha knockdowns. The authors should comment on whether they view their results as reconcilable with this alternative determinant of nucleolar shape.

5. The notion of Ki-67 as a pickering agent in contrast to a traditional surfactant is perplexing. Are the authors proposing that Ki-67 is forming a solid-like phase that localizes to the surface? Is the model not that Ki-67 binds chromatin and the nucleolus in contrast to solid-like aggregates of Ki-67, lowering the surface tension on the surface of chromatin? This would explain the FRAP (binding to chromatin strongly and with more specificity during interphase) and fit the truncation data. The authors should reconsider or, at a minimum, explain how chromatin fits in this Pickering agent picture (is chromatin aggregating?). The physical picture is elusive to this reviewer.

6. To summarize, my core concern throughout the paper is the presentation and

interpretation of the DNA signals reported within the nucleolar interior. Beyond the potential imaging artifacts discussed above, the current discussion section does not sufficiently address these findings. As written, the discussion focuses on the rim localization of Ki-67 and its possible interface role, while largely overlooking the implications of DNA detected within the GC. Likewise, summary Figure 8 depicts Ki-67-bound chromatin more as invaginations from the nuclear interior rather than actual chromatin localized within the GC (likely a more accurate representation of the data). I encourage the authors to address this discrepancy directly. Importantly, even if the interpretation of DNA within the nucleolar interior were revised or omitted, the main findings of the study would remain valuable and compelling.

Other major and minor concerns discussed figure by figure:

1. Figure 1B, Figure 3I - the results currently do not take into account possible differences in nucleolar sizes. Upon depletion of Ki-67 or other target genes from the screen, the nucleoli can change in size significantly, which would shift their aspect ratio closer to 1 as smaller nucleoli are more spherical due to the diffraction limit. The mechanism proposed for Ki-67 would be expected to shift the relationship between nucleolar area and aspect ratio (e.g. aspect ratio of nucleoli at the same average area) It would be important to confirm Ki-67 depletion and overexpression experiment conclusions to determine if Ki-67 changes the number of nucleoli or the surface properties.

2. Figure 2F - the line scan for chromatin foci (1) is confusing. Ki-67 signal seems to be present asymmetrically on one side of the DNA region. At the same time NPM1 is strongly depleted in that area. The strongly asymmetric Ki-67 profile and strong depletion of NPM1 in that area rather suggest that this is, in fact, an invagination of the nuclear interior seen from a different optical slice. The nucleolar rim profile is, however, clear.

3. Figure 2G - we suggest using fire mode for colour scales, which would much improve the ability to see the difference in signal intensity. Figure 2H - please use microns on the X axis as the number of rings is an unclear measure (but presumably can be converted by the thickness per ring).

4. Figure 2H "Nucleolar Rim". The authors see a clear peak here for Ki-67; however, the rim and interior have identical fold changes in Figure 2D - why are these quantifications in conflict?

5. Figure 3F - see point about spectral bleed-through above. It would be useful to repeat this experiment with H2B-mCherry Ki-67-EGFP combination or, alternatively, not label

NPM1 and instead use SiR DNA dye with Ki-67-EGFP to ensure there is no channel bleed-through at high intensities. Notably, 3G has a characteristic signature of bleed-through for partitioning measurements (relative ratio = $r_{\text{bleed}} \cdot K_{\text{no}} \cdot [\text{Ki-67 total}] / ([\text{DNA in nucleoplasm}] + r_{\text{bleed}} \cdot [\text{Ki-67 total}])$).

6. Figure 4D - "No IAA treatment" control cells appear to also have a pronounced decrease in nucleolar aspect ratio over the timecourse of imaging, complicating the interpretation of results. This effect on control cells would suggest stress from improper incubator conditions or phototoxicity. Consider re-doing it with less channels to decrease phototoxicity. Figure S6 does not have the same issue, for example. Also, for Figure 4B-E please specify what kind of error the shading on the plots represents.

7. Figure 5C-E - please consider changing the plots for better visual readability. Using z-score and arranging the plots horizontally may help, as it is extremely hard to estimate the differences from the plots in their current form.

Referee #2 (Report for Author)

The nucleolus is a biomolecular condensate whose morphology reflects its function. The authors performed a siRNA screen using nucleolar shape as a readout. One of their top hits is Ki-67, a protein well known for its role as a chromosome surfactant and for its connection to the nucleolus (localization at the nucleolar periphery and importance for nucleolar genesis). This includes outstanding work both by the present authors and by others.

In this study, the authors confirm the localization of Ki-67 at the interface between perinucleolar chromatin and the nucleolus. They confirm that Ki-67 depletion results in nucleolar rounding, accompanied by depletion of intranucleolar chromatin. This phenotype is observed both upon long-term depletion (siRNAs) and acute depletion (auxin degran). It is reversible, although the effect on shape was less obvious to see, in my opinion, in short-term experiments.

Conversely, overexpression of Ki-67 leads to recruitment of chromatin into the nucleolus, at the expense of chromatin outside. Interestingly, this does not seem to perturb internal nucleolar organization. The authors also perform domain mapping, extending prior work that had already attributed functions to specific regions of Ki-67. Finally, the authors report that microinjection of a restriction enzyme (to digest nuclear DNA) also causes nucleolar rounding.

Technically, the study is rather fine (I was less convinced by Fig 6).

However, a major concern is that a significant fraction of the data presented has already been reported, in some cases nearly ten years ago (e.g. DOI: 10.7554/eLife.13722).

Examples include:

- 1) the localization of Ki-67 at the nucleolar-chromatin interface (Results section #2),
- 2) its essential role in nucleologenesis during mitosis (perichromosomal formation; this should be added to the Introduction), and, in fact,
- 3) even the effect of Ki-67 depletion on nucleolar rounding (Results section #1; see Fig. 7A in Sobecki et al., eLife 2016).

The authors themselves have already contributed substantially to defining the surfactant nature of Ki-67, confirming its localization to the nucleolar rim and its role in nucleologenesis (DOI: 10.15252/msb.20209469). There were also connections established previously between surrounding chromatin and the nucleolus (e.g. 10.7554/eLife.47533).

It is felt that, altogether, the novelty element of this submission is not sufficient to meet the threshold of The EMBO Journal. I would recommend publication in a different venue, after simplification of redundant data and/or improved referencing of prior studies.

Specific comments:

- 1) In Figure S2, the authors use holotomography to confirm, once more, nucleolar rounding in the absence of Ki-67. However, details on the technique (hardware, setup) are not provided.

If holotomography here is indeed equivalent to digital holographic microscopy (DHM), please reference Zorbas et al. (EMBO Reports 2024, DOI: 10.1038/s44319-024-00134-5), which used exactly the same AI approach (neural networks) for segmentation of the nucleolus.

That work extracted quantitative parameters such as "nucleolar optical volume" from OPL variations, providing a strong proxy for material state. Could similar quantifications be applied here to strengthen the analysis?

- 2) The claim that internal nucleolar organization is not affected by Ki-67 enrichment of intranucleolar chromatin seems counterintuitive. Such enrichment should logically alter

nucleolar architecture. Electron microscopy or DHM-based assessment of material state could help clarify this point.

Clear examples of redundancies:

-The conclusion of Fig. 1 of the present submission is already in Fig. 7A of Sobecki et al. 2016.

-The conclusion of Fig. 2 is also in Sobecki et al. 2016, which explicitly stated (p.12):
"...During interphase, we found that Ki-67 localised at the cortical periphery of the GC, visualised using PES1. Ki-67 formed a boundary between the perinucleolar heterochromatin (clearly visible as a 'ring' in the DAPI staining) and the GC (Figure 7A, Figure 7-figure supplement 1)."

-Etc.

Referee #3 (Report for Author)

This excellent manuscript begins with the description of a screen for modulators of nucleolar shape. Although novel, at first it seemed only moderately exciting to me. However, one of the hits (Ki-67) revealed an interesting link between nucleolar shape and chromatin, and the authors did a wonderful job connecting the dots by providing substantial new mechanistic insights, including convincing evidence for a role for Ki-67 in tethering chromatin to the nucleolus and thereby affecting nucleolar shape, plus beautiful super-resolution data showing that the long Ki-67 molecules are oriented within the nucleolus-chromatin interface.

Although Ki-67 had previously already been implicated in nucleoli and the positioning of (hetero)chromatin, this study provides important new pieces of the puzzle.

The writing is very clear and nicely compact, the figures are beautiful and effective, the experimental design is rigorous, and all data are supported by solid statistical analysis.

Altogether this is a very rounded story (pun intended). Although one can think of follow-up experiments to address new questions raised by the interesting findings, I do not think these are necessary for the current manuscript.

However, one concern is that all results are based on a single cell type (HeLa cells) only. I would recommend that a few key experiments are repeated in at least one other cell type. This will be important to verify that the findings are not restricted to a single cell lineage.

=====
Reviewed by Bas van Steensel, review task received: 28 July 2025;
completed 17 Aug 2025 (apologies for the delay; I was on vacation). It is my standard policy
to sign and date *all* of my manuscript review reports, regardless of my comments and
recommendations. All correspondence about this manuscript should go via the editor.
PLEASE DO NOT REMOVE THIS NOTE =====

Referee #1

In Schichler, Hayashi et al., the authors define proliferation marker Ki-67 as one of the proteins regulating nucleolar shape in proliferating cells from a high-throughput screen of nucleolar shape. Using auxin-inducible degron and dose-controlled overexpression, the authors show a clear causal link between Ki-67 and nucleolar shape. Following their previous works, they further establish that the amphiphilic properties of Ki-67 are necessary for its correct interphase localization. This is notable as it is unclear which proteins and mechanisms anchor perinucleolar chromatin, despite nucleolar-associated domains (NAD) being described for many years. This paper brings new knowledge on the potential mechanisms of NAD tethering to the nucleolus by Ki-67. Strikingly, the authors also suggest the presence of heterochromatin within the nucleolar interior, a notion that is both confusing due to the long history of DNA being difficult to identify in the nucleolus in EM and alarming regarding data interpretation. Overall, the manuscript will be of high interest given the well-established connection between nucleolar shape and human physiology, disease, and the general focus of elucidating the determinants of condensate form and function in cells. Currently, the role of Ki-67 and nucleolar shape is well established in this study; however, multiple problems in the manuscript remain, such as the notion that Ki-67 brings chromatin into the interior of the nucleolus, along with others discussed below. Thus, I am not in support of the manuscript being published in its current form.

We thank the reviewer for recognizing the significance of our findings on Ki-67 and for the helpful suggestions on how to further improve our manuscript. We are particularly grateful for the comments and questions regarding the origin of intranucleolar chromatin, which represents a novel and intriguing discovery. In response, we have performed additional experiments to rule out potential imaging artefacts and strengthen the manuscript (see below).

1. The detection of chromatin signals within the nucleolus is intriguing but also raises concerns about potential imaging artifacts. Are the authors certain that the measured signals originate from within the GC, rather than from Z-bleed of chromatin located above or below the focal plane, given the irregular shape of nucleoli? While the observed differences between conditions are striking, Z-bleed artifacts would likely be less prominent in highly spherical nucleoli than in complex or multipart structures. A full 3D stack of the nucleus would be needed to prove whether these DNA signals are truly coming from the interior of the nucleolus (ideally with and without 3D deconvolution). Alternatively, these DNA and Ki-67 signals might represent chromatin invaginations into the nucleolus that are captured in the given optical section.

We thank the reviewer for this valuable point. We recognize that our phrasing was suboptimal when referring to “chromatin foci,” which may have implied isolated structures. Our observations are in fact consistent with chromatin invaginations into the nucleolus, as illustrated in our model (Fig. 8). In cross-sections, these invaginations can appear as discrete foci, and we emphasized them because the associated Ki-67 signal is readily detected in their vicinity. However, in 3D reconstructions it is clear that they remain connected to the nuclear periphery. To rule out artefacts from irregular nucleolar shapes, we performed volumetric 3D imaging of chromatin throughout the entire nucleus. Orthogonal views and Z-axis line profiles confirm that these structures are true invaginations, not isolated fragments. For example, the Z-axis profile of one such region shows three distinct chromatin peaks – below, inside and above the nucleolus indicating a chromatin fiber traversing the nucleolus via an invagination (Response Figure 1). This rules out Z-bleed artefacts.

This 3D evidence strongly supports our conclusion that these are invaginated chromatin channels connected to the nuclear periphery and rules out Z-bleed artefacts. We will clarify this point and consistently use the term "chromatin invagination" throughout the revised manuscript. Although these invaginations are topologically part of the nucleolar periphery, they form striking, thin fibers that extend deep into the nucleolar interior.

Response Figure 1: Chromatin invagination into the nucleolus.

(A) A 3D confocal stack of HeLa cells expressing H2B-mCherry. Orthogonal views show chromatin foci within the nucleolus. A yellow cross marks a single chromatin focus.

(B) Chromatin signal along the z-axis. H2B intensity profile measured through the centre of the yellow cross in (A).

2. Related to the detection of chromatin within the nucleolar interior, the manuscript lacks sufficient discussion and literature context to support the interpretation of these findings. What types of heterochromatic DNA sequences do the authors propose could be enriched inside the nucleolus? The discussion currently cites Ki-67 interacting with heterochromatic, late-replicating sequences at the nucleolar rim. While this explains the presence of Ki-67 at the periphery, it does not account for the co-localization of both Ki-67 and DNA within the nucleolar interior. To our knowledge, the only DNA normally found inside the nucleolus is rDNA, which is already challenging to visualize even by EM, and truly heterochromatic sequences are not reported in the nucleolar interior. Indeed, the notion is perplexing based on the overwhelming difficulty of identifying DNA in EM images of nucleoli, which has led to the location of pre-rRNA transcription being controversial (PMCID: PMC2173404) - why do the authors think this was missed? As it stands, the claim about chromatin present in the nucleolar interior is unsupported.

We thank the reviewer for this valuable point and for highlighting the need for careful discussion and literature context regarding DNA detected within the nucleolar interior. We have now clarified that the internucleolar chromatin originates from invaginations, meaning the interface between this chromatin and the nucleolus is topologically part of the nucleolar periphery, suggesting that they represent heterochromatin invaginations from the nucleolar periphery rather than rDNA.

To directly test this, we performed DNA-FISH for rDNA and included the results as a new figure in the revised manuscript (Response Figure 2A). This analysis shows that the bulk DNA signals exhibit minimal colocalization with rDNA foci (Response Figure 2B). Instead, immunostaining for heterochromatin markers HP1 and H3K20me3 (Response Figure 2B–2D) indicates that these invaginations correspond to heterochromatic regions. Together, our findings extend the current model of heterochromatin and nucleolus-associated domains (NADs) beyond the nucleolar surface into the nucleolar interior, providing

new insight into the mechanistic bases of chromatin invaginations while remaining consistent with previous observations.

Response Figure 2: Intranucleolar chromatin shows heterochromatin signatures.

(A) Distinct localisation of rDNA and chromatin in the nucleolus. DNA-FISH against rDNA cells was performed in HeLa cells. DNA was stained with DAPI.

(B–D) Colocalisation of intranucleolar chromatin with heterochromatin markers. Immunostaining for HP1α, HP1β, and H4K20me3 was performed in HeLa cells. DNA was stained with DAPI.

In addition, we will comment on the existing literature in the revised version highlighting the challenges of detecting DNA in EM images. The reason why chromatin invaginations might have been missed could relate to the dense nature of nucleoli, differences in cell states or cell types, and variations in preparation protocols, which may introduce cellular stress.

3. Since this study is largely based on quantitative microscopy, the authors should verify three things regarding possible imaging artifacts: (a) Axial (Z) bleed-through, which can cause peripheral chromatin signals to appear within the nucleolus in 2D sections, particularly for irregularly shaped or multipart nucleoli. (b) Detector linearity should be verified for quantitative measurements. Notably, the detector is a PMT, which has difficulty imaging more than 10-fold ranges of overexpression, and thus, there is concern that the 32-fold differences may be outside of the linear range. Alternatively (and ideally), the users may have made a conversion table from one laser power to another, allowing for a higher dynamic range. (c) The authors need to confirm if there is spectral bleed-through, which could lead to apparent signal overlap between channels. Furthermore, it is not evident in the methods if the imaging is done sequentially or simultaneously, and thus, there may be substantial (if simultaneous) bleed-through from overexpressed Ki67 into the DNA channels (Figure 3). Additionally, sequentially acquired channels can have bleed at high overexpression levels. For both, please specify whether detector linearity and spectral separation were validated under the same imaging conditions. The method section currently does not describe this or the details for acquisition parameters, such as the filters that were used.

We thank the reviewer for raising these important technical points regarding our quantitative microscopy. We have performed additional validation experiments and updated the Methods section accordingly to address each concern.

- (a) Regarding the axial bleed-through, we fully agree that it is critical to distinguish true intranucleolar signals from potential axial bleed-through artefacts. As detailed in our response to Comment #1, we re-examined the 3D imaging data of the entire nucleus. The orthogonal views and Z-axis line

profile analysis (Response Figure 1) confirm that these signals are true chromatin invaginations rather than artifacts from peripheral chromatin projecting into the focal plane of 2D sections.

(b) We thank the reviewer for this suggestion. While we plan to perform serial dilution calibration to verify PMT linearity, we note that this is not essential for our conclusions, as the primary purpose of these measurements is to demonstrate a positive correlation with chromatin recruitment and irregular shape rather than to provide absolute quantification (Figure 3F–3I).

(c) To test the presence of spectral bleed-through by Ki-67 overexpression, we performed two key experiments to validate our setup:

1. We imaged cells highly overexpressing our EGFP-Ki-67 construct without fluorescence labelling of DNA and NPM1 (Respond Figure 2A). Under the exact same acquisition settings used for our experiments, we detected no significant signal in the DNA or NPM1 channels. This directly confirms the absence of spectral bleed-through from the Ki-67 channel.
2. To ensure the phenotype wasn't an artefact of the fluorescent tag or the bleed-through, we additionally used an EGFP-T2A-Ki-67 construct to overexpress Ki-67. This system co-expresses untagged Ki-67 and a separate, soluble EGFP reporter at equivalent levels. Cells with high soluble EGFP levels, indicating high levels of untagged Ki-67, fully recapitulated the nucleolar deformation and the excess chromatin enrichment in the nucleolus (Respond Figure 2B).

Together, these controls demonstrate that our observations reflect true effects of Ki-67 overexpression and not imaging artefacts. We will also update the Methods section in the revised manuscript to specify the acquisition parameters, including filter sets, laser powers, and PMT settings, to ensure full transparency and reproducibility.

Response Figure 3: Ki-67 overexpression directly causes nucleolar irregularity and chromatin enrichment within the nucleolus.

(A) Minimal spectral bleed-through from the overexpressed EGFP-Ki-67. HeLa cells expressing SNAP-NPM1 were transfected with EGFP-Ki-67. Images were acquired with the same settings with or without fluorescence labelling of NPM1 and DNA with SNAP-SiR and SPY555-DNA, respectively.

(B) Aberrant nucleolar morphology and chromatin enrichment is caused by Ki-67 itself. HeLa cells expressing SNAP-NPM1 were transfected with EGFP-T2A-Ki-67. Images were acquired following fluorescence labelling of NPM1 and DNA with SNAP-SiR and SPY555-DNA, respectively.

4. A recent study - Riback et al Mol Cell 2023 - suggests that nucleolar activity (locations of FCs) maintains the non-spherical shape of nucleoli and rules out bulk chromatin mechanics employing lamin and hp1alpha knockdowns. The authors should comment on whether they view their results as reconcilable with this alternative determinant of nucleolar shape.

We thank the reviewer for highlighting the study by Riback et al. (Mol Cell 2023) and will include a discussion of this study in a revised version.

From our perspective, the experiments presented to rule out a role for chromatin in the Riback et al. study are not conclusive. Depletion of factors such as lamin A or HP1 α induces widespread nuclear alterations, including shape abnormalities, rupture, and large-scale chromatin reorganization (Larrieu et al., Science; 2014 Strom et al., eLife, 2021), which are likely to cause indirect effects on nucleolar morphology. Such conditions may therefore obscure rather than clarify the specific contribution of chromatin to nucleolar structure. Moreover, chromatin softening upon these depletions could, in principle, make it even easier for Ki-67 (if still present) to recruit chromatin into nucleoli, potentially enhancing invaginations rather than simply producing nucleolar rounding. Thus, the experiments used to rule out a role for chromatin in Riback et al. cannot be considered conclusive.

In contrast, Ki-67 depletion specifically disrupts the chromatin–nucleolus anchor while having minimal or no effect on nuclear morphology (*data will be provided in the revised version*). Our findings indicate that Ki-67–mediated tethering of perinucleolar heterochromatin drives chromatin invaginations and thereby generates non-spherical nucleolar features (Response Figure 3B).

In support of the transcription-based shape regulation, Riback et al. showed that upon transcriptional inhibition, nucleoli round up. While we fully agree that transcriptional activity are important determinants of nucleolar morphology, rounding under these conditions could result from multiple factors. Notably, we also observe a loss of Ki-67 and chromatin from the nucleolar interior under these conditions, raising the possibility that the rounding upon transcription inhibition is, at least in part, due to the absence of Ki-67–mediated tethering (*data will be provided*).

Taken together, we propose a unified model in which transcription drives nucleolar assembly, while Ki-67–mediated chromatin tethering refines and stabilises its architecture. In this framework, transcription is sufficient to form nucleoli, but Ki-67 is required to deform them away from spherical symmetry, giving rise to invaginations and more complex morphologies. We will clarify this distinction and explicitly reconcile our results with the model proposed by Riback et al. in the revised Discussion.

To rule out cell type–specific differences, we will also perform Ki-67 depletion in HEK and U2OS cells, the same lines used in Riback et al., to directly test whether the effects on nucleolar shape are consistent across cell types.

5. The notion of Ki-67 as a pickering agent in contrast to a traditional surfactant is perplexing. Are the authors proposing that Ki-67 is forming a solid-like phase that localizes to the surface? Is the model not that Ki-67 binds chromatin and the nucleolus in contrast to solid-like aggregates of Ki-67, lowering the surface tension on the surface of chromatin? This would explain the FRAP (binding to chromatin strongly and with more specificity during interphase) and fit the truncation data. The authors should reconsider or, at a minimum, explain how chromatin fits in this Pickering agent picture (is chromatin aggregating?). The physical picture is elusive to this reviewer.

We will consult an expert and reconsider our model in the revised manuscript, if necessary. Our current thinking is the following:

We thank the reviewer for highlighting the need to clarify the physical model for Ki-67's function. We believe the “physical picture” need not be mutually exclusive: Ki-67 can act as a surfactant at a molecular scale while simultaneously exhibiting Pickering-like behavior at a mesoscopic, interface-stabilizing scale.

We propose that Ki-67 combines features of both a surfactant and a Pickering-like agent, acting as a molecular bridge that stabilises the chromatin-nucleolus interface. This model is based on two key characteristics. First, similar to a molecular surfactant, Ki-67 possesses distinct domains with affinities for both chromatin and nucleolar components, physically linking the two compartments and reducing interfacial tension between nucleolar material and perinucleolar chromatin,. Second, unlike traditional surfactants Ki-67 exhibits moderate mobility and likely multivalent RNA binding (Ma et al., Cell Discovery 2022), allowing it to form a more persistent viscoelastic interphase layer that resists coalescence and maintains invaginations. In this model, chromatin is not aggregating. Rather, Ki-67 tethers chromatin while interacting with nucleolar RNA/protein, producing a semi-solid interface that maintains non-spherical features, consistent with FRAP dynamics and truncation data.

Thus, Ki-67 can act as a surfactant at the nanoscale while simultaneously exhibiting Pickering-like behavior at the mesoscopic, interface-stabilizing scale. We will clarify this physical model in the revised manuscript, explicitly highlighting how Ki-67 differs from a classical surfactant and its dual surfactant/Pickering-like role at the chromatin–nucleolus interface.

6. To summarize, my core concern throughout the paper is the presentation and interpretation of the DNA signals reported within the nucleolar interior. Beyond the potential imaging artifacts discussed above, the current discussion section does not sufficiently address these findings. As written, the discussion focuses on the rim localization of Ki-67 and its possible interface role, while largely overlooking the implications of DNA detected within the GC. Likewise, summary Figure 8 depicts Ki-67-bound chromatin more as invaginations from the nuclear interior rather than actual chromatin localized within the GC (likely a more accurate representation of the data). I encourage the authors to address this discrepancy directly. Importantly, even if the interpretation of DNA within the nucleolar interior were revised or omitted, the main findings of the study would remain valuable and compelling.

We thank the reviewer for allowing us to clarify potential misunderstandings and for the helpful suggestions on how to further improve our manuscript. We fully agree that the DNA signal within the nucleolar interior require careful validation and discussion. To address this, we provide new data demonstrating that these signals correspond to chromatin invaginations into the nucleolar interior (see comment #1), that they are distinct from rDNA (see comment 2) and that the Ki-67 over-expression phenotype is independent of the spectral bleed-through (see comment #3). We will clarify this interpretation throughout the revised manuscript and add a discussion of its implications in the updated version.

Other major and minor concerns discussed figure by figure:

1. Figure 1B, Figure 3I - the results currently do not take into account possible differences in nucleolar sizes. Upon depletion of Ki-67 or other target genes from the screen, the nucleoli can change in size significantly, which would shift their aspect ratio closer to 1 as smaller nucleoli are more spherical due to the diffraction limit. The mechanism proposed for Ki-67 would be expected to shift the relationship between nucleolar area and aspect ratio (e.g. aspect ratio of nucleoli at the same average area) It would be important to confirm Ki-67 depletion and overexpression experiment conclusions to determine if Ki-67 changes the number of nucleoli or the surface properties.

We appreciate the reviewer's comment to improve our conclusion. We agree that changes in nucleolar size and its number could potentially confound the aspect ratio measurements, as smaller objects are inherently more spherical.

To address this, we have re-analysed our data from both the Ki-67 depletion and overexpression experiments. Our analysis confirms that altering Ki-67 levels has minimal differences in either the total nucleolar area per nucleus or the number of nucleoli between conditions (Response Figure 4). Because the nucleolar size remains largely unchanged regardless of Ki-67 depletion or its overexpression, the alteration of nucleolar shape is not a secondary consequence of size and number changes. This strengthens our conclusion that Ki-67 directly modulates the surface properties and shape of the nucleolus, independent of its size. We have added this important control analysis to the manuscript.

Response Figure 4: Changes of Ki-67 levels leads minimal effects on the number and area of nucleolus.

(A, B) Effect on the nucleolar area and its number by Ki-67 depletion. Re-analysis of data in Figure 3A–E for plotting the nucleolar area (A) and its number (B).

(C, D) Effect on the nucleolar area and its number by Ki-67 overexpression. Re-analysis of data in Figure 3F–G for plotting the nucleolar area (C) and its number (D).

In the screen setup, we applied a size threshold to filter out very small objects that could potentially obscure the phenotype. We will mention this in the figure legend and clarify it in the Methods section.

2. Figure 2F - the line scan for chromatin foci (1) is confusing. Ki-67 signal seems to be present asymmetrically on one side of the DNA region. At the same time NPM1 is strongly depleted in that area. The strongly asymmetric Ki-67 profile and strong depletion of NPM1 in that area rather suggest that this is, in fact, an invagination of the nuclear interior seen from a different optical slice. The nucleolar rim profile is, however, clear.

We thank the reviewer for their insightful comment regarding the line profile in Figure 2F. We agree with the interpretation that DNA signal corresponds to a chromatin invagination into the nucleolar body.

To confirm this, we re-examined the 3D confocal stacks. As shown in the orthogonal views (Response Figure 5), the structure is clearly a deep chromatin invagination. Crucially, Ki-67 is enriched along the entire surface of the invaginated chromatin, confirming its consistent interfacial localisation between chromatin and the nucleolus. We note that Ki-67 is patchy along chromatin surfaces in general, independent of the invagination, which we attribute to its gel-like, viscoelastic behavior. We have updated Figure S3 to include these clearer orthogonal-view images to illustrate this point effectively.

Response Figure 5: Interfacial localisation of Ki-67 along chromatin invagination. Orthogonal views at the chromatin foci in Figure 2F was shown. A yellow cross marks a single chromatin focus.

3. Figure 2G - we suggest using fire mode for colour scales, which would much improve the ability to see the difference in signal intensity. Figure 2H - please use microns on the X axis as the number of rings is an unclear measure (but presumably can be converted by the thickness per ring).

We appreciate the reviewer's feedback on the colour scales. In response, we have changed the figures to use the 'fire' colour map, which improves the visual distinction between signal intensities.

For the concentric ring analysis, it is important to note that the algorithm normalises the ring widths on a per-nucleolus basis, which is required when averaging many nucleoli because nucleolar segmentations vary widely in size. Consequently, the absolute thickness of individual rings cannot be extracted from this normalised data.

4. Figure 2H "Nucleolar Rim". The authors see a clear peak here for Ki-67; however, the rim and interior have identical fold changes in Figure 2D - why are these quantifications in conflict?

We thank the reviewer for the insightful question. The apparent discrepancy arises because the two figures quantify different aspects of the Ki-67 localisation. Figure 2H shows the local enrichment of Ki-67. The clear peak of Ki-67 between NPM1 and DNA signals in the nucleolar rim plot demonstrates the interfacial localisation of Ki-67. Conversely, Figure 2D displays the average intensity of Ki-67 across the entire nucleolar rim and its interior. The rim was defined to include the DNA signal around the nucleolus (see Fig. 2B), meaning that for Ki-67, a substantial portion of nucleoplasm (low signal) is also averaged. Combined with the patchy localisation in both regions, this results in similar average intensities despite the pronounced local enrichment (rings 20-30, Fig. 2H).

5. Figure 3F - see point about spectral bleed-through above. It would be useful to repeat this experiment with H2B-mCherry Ki-67-EGFP combination or, alternatively, not label NPM1 and instead use SiR DNA dye with Ki-67-EGFP to ensure there is no channel bleed-through at high intensities. Notably, 3G has a characteristic signature of bleed-through for partitioning measurements ($\text{relative ratio} = \frac{r_{\text{bleed}} \cdot K_{\text{no}} \cdot [\text{Ki-67 total}]}{([\text{DNA in nucleoplasm}] + r_{\text{bleed}} \cdot [\text{Ki-67 total}])}$).

We thank the reviewer for highlighting the potential issue of spectral bleed-through in Figures 3F,G. To address this concern, we have performed additional control experiments using the same acquisition settings but without fluorescence labelling of NPM1 and DNA (see Response Figure 3A). These experiments together with the additional experiment using the EGFP-T2A-Ki-67 construct (see Response

Figure 3B) confirmed that the Ki-67 overexpression directly causes the excess chromatin enrichment in the nucleolus and the deformation of nucleolar morphology.

6. Figure 4D - "No IAA treatment" control cells appear to also have a pronounced decrease in nucleolar aspect ratio over the timecourse of imaging, complicating the interpretation of results. This effect on control cells would suggest stress from improper incubator conditions or phototoxicity. Consider re-doing it with less channels to decrease phototoxicity. Figure S6 does not have the same issue, for example. Also, for Figure 4B-E please specify what kind of error the shading on the plots represents.

We thank the reviewer for pointing this out. We agree that the slight decrease in nucleolar aspect ratio observed in the "No IAA" control cells likely reflects general stress. While we have optimized imaging conditions to minimize these effects (as shown in Figure S6), prolonged imaging in the microscope incubator cannot perfectly maintain temperature and, more importantly, humidity compared to a standard cell culture incubator. Therefore, we attribute the observed baseline stress primarily to the limitations of the microscopy incubation rather than to imaging itself. Notably, control cells expressing wild-type Ki-67 (Fig. S6) are consistently less sensitive than EGFP-AID-Ki-67 cells, likely due to minimal basal degradation of Ki-67. Unfortunately, there is not much more we can do to further reduce this inherent stress.

Regarding the plots in Figures 4B, we will clarify in the figure legends that the shading represents the mean \pm SD for 4B and 4E, and the mean \pm SEM for 4D.

7. Figure 5C-E - please consider changing the plots for better visual readability. Using z-score and arranging the plots horizontally may help, as it is extremely hard to estimate the differences from the plots in their current form.

We thank the reviewer for their suggestions to improve the data visualisation. In the revised figure, we have rotated the plots to a horizontal orientation for better alignment with the corresponding images and have clarified the meaning of the green and grey dashed lines in the figure legend. While z-scores could standardize the data, we believe that reporting absolute measurements of nucleolar circularity and chromatin enrichment, along with medians, is more transparent and biologically interpretable.

Referee #2 (Report for Author)

The nucleolus is a biomolecular condensate whose morphology reflects its function. The authors performed a siRNA screen using nucleolar shape as a readout. One of their top hits is Ki-67, a protein well known for its role as a chromosome surfactant and for its connection to the nucleolus (localization at the nucleolar periphery and importance for nucleolar genesis). This includes outstanding work both by the present authors and by others.

In this study, the authors confirm the localization of Ki-67 at the interface between perinucleolar chromatin and the nucleolus. They confirm that Ki-67 depletion results in nucleolar rounding, accompanied by depletion of intranucleolar chromatin. This phenotype is observed both upon long-term depletion (siRNAs) and acute depletion (auxin degran). It is reversible, although the effect on shape was less obvious to see, in my opinion, in short-term experiments.

Conversely, overexpression of Ki-67 leads to recruitment of chromatin into the nucleolus, at the expense of chromatin outside. Interestingly, this does not seem to perturb internal nucleolar organization. The authors also perform domain mapping, extending prior work that had already attributed functions to specific regions of Ki-67. Finally, the authors report that microinjection of a restriction enzyme (to digest nuclear DNA) also causes nucleolar rounding.

Technically, the study is rather fine (I was less convinced by Fig 6). However, a major concern is that a significant fraction of the data presented has already been reported, in some cases nearly ten years ago (e.g. DOI: 10.7554/eLife.13722). Examples include:

- 1) the localization of Ki-67 at the nucleolar-chromatin interface (Results section #2),
- 2) its essential role in nucleologenesis during mitosis (perichromosomal formation; this should be added to the Introduction), and, in fact,
- 3) even the effect of Ki-67 depletion on nucleolar rounding (Results section #1; see Fig. 7A in Sobecki et al., eLife 2016).

The authors themselves have already contributed substantially to defining the surfactant nature of Ki-67, confirming its localization to the nucleolar rim and its role in nucleologenesis (DOI: 10.15252/msb.20209469). There were also connections established previously between surrounding chromatin and the nucleolus (e.g. 10.7554/eLife.47533).

It is felt that, altogether, the novelty element of this submission is not sufficient to meet the threshold of The EMBO Journal. I would recommend publication in a different venue, after simplification of redundant data and/or improved referencing of prior studies.

We thank the reviewer for their careful reading and for providing us with the opportunity to clarify the novelty of our work. We apologise if this was not sufficiently emphasised in the original manuscript. Our study provides several key findings that distinguish it from previous work:

- We performed the first high-throughput, image-based siRNA screen to systematically identify proteins, including Ki-67, that regulate nucleolar morphology, particularly those responsible for maintaining its characteristic irregular shape (Figure 1).
- While the study by Sobecki et al. was foundational for Ki-67's role in chromatin organisation, it does not document the function we report here as we found no explicit mention or quantification of nucleolar shape changes in their manuscript. Although their Figure 7A shows a single image of nucleolar staining (PES1 immunostaining) upon Ki-67 depletion suggestive of shape changes, it does not include any analysis or discussion of this phenotype.

- While Sobecki et al. previously described Ki-67's localisation to the nucleolar periphery, our work identifies that Ki-67 also localises to the nucleolar interior, surrounding chromatin invaginations (Figure 2). This observation clearly extends beyond previous observations.
- We demonstrate for the first time that Ki-67 expression levels directly dictate both nucleolar shape and the enrichment of chromatin into the nucleolus (Figure 3 and 4). Ki-67 depletion leads to nucleolar rounding and reduced chromatin enrichment, whereas its overexpression enhances nucleolar deformation and recruits excess chromatin. This Ki-67-dependent chromatin recruitment to the nucleolar interior represents a novel and intriguing finding. In response to Referee 1, we performed follow-up experiments showing that this chromatin is distinct from rDNA and bears heterochromatic marks (see Response Figure 2).
- Using a series of truncation mutants, we map the critical domains of Ki-67 required for its interfacial localisation and its function in determining nucleolar shape (Figure 5). In addition, we showed that Ki-67 exhibits a specific molecular orientation at the chromatin-nucleolar interface (Figure 6).
- Lastly, we show that depleting our top 20 candidate proteins phenocopies Ki-67 depletion (i.e., nucleolar rounding and reduced chromatin within the nucleolus). Notably, we found that the depletion of most of them causes a concomitant decrease in Ki-67 protein levels, suggesting Ki-67 as a key regulatory factor controlling nucleolar morphology (Figure 7).

Taken together, our study provides a novel mechanistic insight into nucleolar morphology regulation. We demonstrate that Ki-67 anchors chromatin to the nucleolar surface via its dual affinity domains, tightly linking nucleolar shape to genome organisation.

To ensure our novel findings are clear, we have revised the Introduction and Discussion sections to more explicitly place our findings in the context of the existing literature, including the foundational work of Sobecki et al.

Specific comments:

1) In Figure S2, the authors use holotomography to confirm, once more, nucleolar rounding in the absence of Ki-67. However, details on the technique (hardware, setup) are not provided.

If holotomography here is indeed equivalent to digital holographic microscopy (DHM), please reference Zorbas et al. (EMBO Reports 2024, DOI: 10.1038/s44319-024-00134-5), which used exactly the same AI approach (neural networks) for segmentation of the nucleolus.

That work extracted quantitative parameters such as "nucleolar optical volume" from OPL variations, providing a strong proxy for material state. Could similar quantifications be applied here to strengthen the analysis?

We thank the reviewer for their careful attention to our methodology and for the opportunity to clarify the distinction between the technique we employed and the one mentioned in the referenced paper. We apologize for not making this difference clearer in the original manuscript.

The holotomographic microscope (Nanolive) that we used employs a specialised implementation that is substantially different from the digital holographic microscopy (DHM) described in Zorbas, C. et al. Our method, holotomography (HT), is based on quantitative phase imaging but is specifically designed to reconstruct a full 3D refractive index (RI) tomogram of the sample. To achieve this, the system captures numerous holograms from multiple illumination angles and uses an algorithm to solve the inverse

scattering problem. This approach provides a detailed, label-free 3D reconstruction of the cell and its internal organelles. Conversely, the digital holographic microscopy (DHM) technique in the referenced paper generally refers to a method that captures a single hologram to numerically reconstruct a 2D quantitative phase or thickness map.

While using different techniques, we employed a similar, yet independently developed, machine-learning-based segmentation algorithm (Shabanov, A. et al., 2021). We will also cite the work by Zorbas et al. in the revised manuscript to acknowledge its relevance and clarify the methodological distinction.

Moreover, we appreciate the suggestion to extract additional quantitative parameters. We have quantified refractive indices upon Ki-67 depletion, which directly report on local material density, and we will include these analyses in the revised manuscript.

2) The claim that internal nucleolar organization is not affected by Ki-67 enrichment of intranucleolar chromatin seems counterintuitive. Such enrichment should logically alter nucleolar architecture. Electron microscopy or DHM-based assessment of material state could help clarify this point.

We appreciate the reviewer's suggestion and the opportunity to clarify this point. We agree that the enrichment of chromatin within the nucleolus could disrupt its internal organisation. We now clarify in response to referee 1 that Ki-67 does not primarily co-localize with rDNA within the nucleolus, but rather with heterochromatic regions that invaginate into the nucleolar interior (see Response Figure 2). Consequently, the enrichment of intranucleolar chromatin by Ki-67 reflects these heterochromatin invaginations, which are distinct from the core nucleolar components. This spatial segregation explains the minimal impact on internal nucleolar architecture (Figure S3) and ribosome biogenesis (Sobecki et al., 2017).

We appreciate the reviewer's suggestion regarding the assessment of nucleolar material state. To address this, we have quantified local refractive indices, which provide a direct readout of nucleolar density, and will include these measurements in the revised manuscript.

Clear examples of redundancies:

-The conclusion of Fig. 1 of the present submission is already in Fig. 7A of Sobecki et al. 2016.
-The conclusion of Fig. 2 is also in Sobecki et al. 2016, which explicitly stated (p.12): "...During interphase, we found that Ki-67 localised at the cortical periphery of the GC, visualised using PES1. Ki-67 formed a boundary between the perinucleolar heterochromatin (clearly visible as a 'ring' in the DAPI staining) and the GC (Figure 7A, Figure 7-figure supplement 1)."

-Etc.

We thank the reviewer for this detailed feedback and for highlighting the need for a clearer distinction between our findings and the foundational work of Sobecki et al. While Sobecki et al. made important observations linking Ki-67 to the nucleolus and surrounding chromatin organization, its role in nucleolar shape regulation, chromatin invaginations, and the underlying mechanism has not been described previously.

- Regarding our Figure 1 and Sobecki et al. Fig. 7A: While Sobecki et al. perform Ki-67 depletion and show the nucleolus using PES1 as a marker protein, they do not focus on the nucleolar shape. We found no explicit mention or quantification of nucleolar shape changes in their manuscript. Rather, the study concludes "Ki-67 did not affect the gross structure of the nucleolus, as determined by PES1 staining". The novelty of our Figure 1 is thus based on Ki-67's role and additionally we provide other candidates from our unbiased screen setup.

- Regarding our Figure 2 and the observations from Sobecki et al. 2016: We agree that Ki-67's localization to the nucleolar periphery is not novel, and we explicitly note this in the manuscript: "Several studies have described its localisation to the region surrounding nucleoli, known as the nucleolar periphery or rim" (citing Sobecki et al. and two other studies). The novelty lies in the internal Ki-67 signal, which is difficult to detect by immunostaining. Using live-cell imaging, we also show that Ki-67 localizes to the nucleolar interior, specifically at the interface where chromatin forms deep invaginations. This intranucleolar localization, beyond the outer rim, represents a key novel finding that underpins our mechanistic model.

Referee #3 (Report for Author)

This excellent manuscript begins with the description of a screen for modulators of nucleolar shape. Although novel, at first it seemed only moderately exciting to me. However, one of the hits (Ki-67) revealed an interesting link between nucleolar shape and chromatin, and the authors did a wonderful job connecting the dots by providing substantial new mechanistic insights, including convincing evidence for a role for Ki-67 in tethering chromatin to the nucleolus and thereby affecting nucleolar shape, plus beautiful super-resolution data showing that the long Ki-67 molecules are oriented within the nucleolus-chromatin interface.

Although Ki-67 had previously already been implicated in nucleoli and the positioning of (hetero)chromatin, this study provides important new pieces of the puzzle.

The writing is very clear and nicely compact, the figures are beautiful and effective, the experimental design is rigorous, and all data are supported by solid statistical analysis.

Altogether this is a very rounded story (pun intended). Although one can think of follow-up experiments to address new questions raised by the interesting findings, I do not think these are necessary for the current manuscript.

However, one concern is that all results are based on a single cell type (HeLa cells) only. I would recommend that a few key experiments are repeated in at least one other cell type. This will be important to verify that the findings are not restricted to a single cell lineage.

=====
Reviewed by Bas van Steensel, review task received: 28 July 2025; completed 17 Aug 2025 (apologies for the delay; I was on vacation). It is my standard policy to sign and date *all* of my manuscript review reports, regardless of my comments and recommendations. All correspondence about this manuscript should go via the editor. PLEASE DO NOT REMOVE THIS NOTE =====

We thank the reviewer for their positive assessment of the writing and experimental quality of our study, as well as for the helpful suggestions to further improve the manuscript. We agree that demonstrating the generality of our findings would significantly strengthen our conclusions.

To address this, we are conducting preliminary experiments in immortalized retinal pigment epithelial cells (RPE-1). Early results appear highly consistent with our initial findings: depletion of Ki-67 by siRNAs induces nucleolar rounding and loss of chromatin enrichment within the nucleolus. These data will be incorporated into the revised manuscript and referenced in the Results section. We believe this addition will substantially strengthen the broad applicability of our proposed mechanism. In the revised version, we also plan to extend the analysis to Ki-67 overexpression and to an additional cell line (such as U2OS or HEK).

Dr. Sara Cuylen-Häring
European Molecular Biology Laboratory
Cell Biology and Biophysics
Meyerhofstr. 1
Heidelberg
Germany

10th Oct 2025

Re: EMBOJ-2025-121939
Ki-67 shapes the nucleolus by anchoring chromatin via its amphiphilic properties

Dear Sara,

Thank you again for your comprehensive point-by-point response for your recent EMBO Journal submission. In light of your revision plans and the preliminary new data you laid out during our meeting, I would be happy to formally invite you to revise your study along the proposed lines. As we discussed, it shall be important to clarify the origin of the observed nucleolar chromatin inclusions, to include the additional evidence for their heterochromatic nature, and to add some discussion on their possible functional significance. Likewise, it will be important to provide the data from additional cell lines/types, and to dispel concerns about potential imaging artifacts and other experimental uncertainties. Regarding queries about previous studies on nucleolar shape/function, on Ki-67 connections to heterochromatin/nucleoli, and about Ki-67 roles as pickering agent, it would be valuable to carefully discuss and diligently clarify your thinking about these issues.

It is our policy to allow only a single round of (major) revision, so please keep me updated should there be any unexpected problems with the revisions, or should you require an extension beyond the default 3-months deadline. As always, competing manuscript published elsewhere during the course of this revision will not affect our eventual decision on your study. Finally, please note the detailed information and guidelines on how to prepare a revision below (and in our online Guide to Authors) - closely adhering to them shall greatly facilitate the editorial process at the time of resubmission.

Thank you again for the opportunity to consider this work, and I look forward to receiving your revision in due time.

With kind regards,

Hartmut

9) To facilitate reproducibility and cross-laboratory adoption of methodologies, please structure the Materials & Methods section as outlined in our guide to authors, including a completed Reagents and Tools Table that can be downloaded from our author guidelines as well (<https://www.embopress.org/page/journal/14602075/authorguide#structuredmethods>).

10) Digital image enhancement is acceptable practice, as long as it accurately represents the original data and conforms to community standards. If a figure has been subjected to significant electronic manipulation, this must be clearly noted in the figure legend and/or the 'Materials and Methods' section. The editors reserve the right to request original versions of figures and the original images that were used to assemble the figure. Finally, we generally encourage uploading of numerical as well as gel/blot image source data; for details see: embopress.org/page/journal/14602075/authorguide#sourcedata

In the interest of ensuring the conceptual advance provided by the work, we recommend submitting a revision within 3 months (8th Jan 2026). Please discuss the revision progress ahead of this time with the editor if you require more time to complete the revisions. Use the link below to submit your revision:

Link Not Available

Referee #1

In Schichler, Hayashi et al., the authors define proliferation marker Ki-67 as one of the proteins regulating nucleolar shape in proliferating cells from a high-throughput screen of nucleolar shape. Using auxin-inducible degron and dose-controlled overexpression, the authors show a clear causal link between Ki-67 and nucleolar shape. Following their previous works, they further establish that the amphiphilic properties of Ki-67 are necessary for its correct interphase localization. This is notable as it is unclear which proteins and mechanisms anchor perinucleolar chromatin, despite nucleolar-associated domains (NAD) being described for many years. This paper brings new knowledge on the potential mechanisms of NAD tethering to the nucleolus by Ki-67. Strikingly, the authors also suggest the presence of heterochromatin within the nucleolar interior, a notion that is both confusing due to the long history of DNA being difficult to identify in the nucleolus in EM and alarming regarding data interpretation. Overall, the manuscript will be of high interest given the well-established connection between nucleolar shape and human physiology, disease, and the general focus of elucidating the determinants of condensate form and function in cells. Currently, the role of Ki-67 and nucleolar shape is well established in this study; however, multiple problems in the manuscript remain, such as the notion that Ki-67 brings chromatin into the interior of the nucleolus, along with others discussed below. Thus, I am not in support of the manuscript being published in its current form.

We thank the reviewer for recognizing the significance of our findings on Ki-67 and for the constructive suggestions to enhance our manuscript. We are particularly grateful for the comments and questions regarding the origin of intranucleolar chromatin, which represents a novel and intriguing aspect of our work. In response, we have performed additional experiments to rule out potential imaging artefacts and further strengthen the manuscript (see below).

1. The detection of chromatin signals within the nucleolus is intriguing but also raises concerns about potential imaging artifacts. Are the authors certain that the measured signals originate from within the GC, rather than from Z-bleed of chromatin located above or below the focal plane, given the irregular shape of nucleoli? While the observed differences between conditions are striking, Z-bleed artifacts would likely be less prominent in highly spherical nucleoli than in complex or multipart structures. A full 3D stack of the nucleus would be needed to prove whether these DNA signals are truly coming from the interior of the nucleolus (ideally with and without 3D deconvolution). Alternatively, these DNA and Ki-67 signals might represent chromatin invaginations into the nucleolus that are captured in the given optical section.

We thank the reviewer for this valuable point. We recognize that the term “chromatin foci” may have been misleading, as chromatin signals within nucleoli actually represent chromatin invaginations into the nucleolus, as illustrated in our model (Revised Figure 9). In cross-sections, these invaginations can appear as discrete foci, and we emphasized them because the associated Ki-67 signal is readily detected in their vicinity. However, 3D reconstructions show that they remain connected to the nuclear periphery rather than forming isolated fragments. For example, the Z-axis profile of one such region shows three distinct chromatin peaks – below, inside and above the nucleolus – indicating a chromatin fiber invaginating into the nucleolus while remaining connected to the nucleolar periphery (Response Figure 1; Revised Figure S4).

This 3D evidence strongly supports our conclusion that the chromatin signal within nucleoli represent invaginated chromatin fibers connected to the nuclear periphery. Although these invaginations are topologically part of the nucleolar periphery, they form striking fibers that extend deep into the nucleolar interior. We will clarify this point and consistently use the term “chromatin invagination” throughout the revised manuscript.

Response Figure 1: Chromatin invagination into the nucleolus.

(A) A 3D confocal stack of HeLa cells expressing H2B-mCherry. Orthogonal views reveal chromatin invaginations within the nucleolus, with a yellow cross marking a single chromatin focus.

(B) Chromatin signal along the z-axis. H2B intensity profile measured through the centre of the yellow cross in (A).

2. Related to the detection of chromatin within the nucleolar interior, the manuscript lacks sufficient discussion and literature context to support the interpretation of these findings. What types of heterochromatic DNA sequences do the authors propose could be enriched inside the nucleolus? The discussion currently cites Ki-67 interacting with heterochromatic, late-replicating sequences at the nucleolar rim. While this explains the presence of Ki-67 at the periphery, it does not account for the colocalization of both Ki-67 and DNA within the nucleolar interior. To our knowledge, the only DNA normally found inside the nucleolus is rDNA, which is already challenging to visualize even by EM, and truly heterochromatic sequences are not reported in the nucleolar interior. Indeed, the notion is perplexing based on the overwhelming difficulty of identifying DNA in EM images of nucleoli, which has led to the location of pre-rRNA transcription being controversial (PMCID: PMC2173404) - why do the authors think this was missed? As it stands, the claim about chromatin present in the nucleolar interior is unsupported.

We thank the reviewer for this valuable point and for highlighting the need for careful discussion and literature context regarding DNA detected within the nucleolar interior. We have now clarified that the intranucleolar chromatin originates from invaginations (Response Figure 1; Revised Figure S4), meaning this intranucleolar chromatin is topologically part of the nucleolar periphery, suggesting that it represents heterochromatin invaginations from the nucleolar periphery rather than rDNA.

To directly test this, we performed DNA-FISH for rDNA and included the results as a new Figure in the revised manuscript (Response Figure 2A; Revised Figure S5A). This analysis shows that the bulk DNA signals exhibit minimal colocalization with rDNA regions. Instead, immunostaining for heterochromatin markers (HP1 α , HP1 β , H3K9me3, and H4K20me3) indicates that these invaginations correspond to heterochromatic regions (Response Figure 2B–2E; Revised Figure S5B–S5E). Together, our findings extend the current model of heterochromatin and nucleolus-associated domains (NADs) beyond the nucleolar surface into the nucleolar interior, providing new insight into the organisation of heterochromatin within and around the nucleolus.

Response Figure 2: Intranucleolar chromatin shows heterochromatin signatures.

(A) Distinct localisation of rDNA and chromatin within the nucleolus. DNA-FISH against rDNA in HeLa cells, with DNA counterstained using DAPI.

(B–D) Colocalisation of intranucleolar chromatin with heterochromatin markers. Immunostaining for HP1α, HP1β, and H4K20me3 in HeLa cells, with DNA counterstained using DAPI.

We believe these chromatin invaginations are difficult to detect in EM studies for several reasons. First, in conventional EM, nucleoli appear as highly electron-dense structures after standard heavy-metal staining, attributed to their RNA-rich composition. This density can obscure underlying DNA structures. Second, large invaginations of compact heterochromatin, while theoretical detectable, would likely require targeted screening of many sections, as the probability of capturing them in a single random ultrathin section is low. Specialized DNA staining approaches, such as ChromEMT (Ou, H. D. et al. Science 2017), could also help visualize these structures, though to date, ChromEMT studies have not focused on nucleolar DNA to our knowledge. To overcome the limitations and uncertainties of staining, cryo-electron tomography of vitrified cells offers a label-free approach to examine chromatin organization within the nucleolus *in situ*. This would require combining cryo-correlative fluorescence microscopy with nucleolar and chromatin markers, along with robust computational methods for localizing nucleosomes, which are not yet fully developed.

3. Since this study is largely based on quantitative microscopy, the authors should verify three things regarding possible imaging artifacts: (a) Axial (Z) bleed-through, which can cause peripheral chromatin signals to appear within the nucleolus in 2D sections, particularly for irregularly shaped or multipart

nucleoli. (b) Detector linearity should be verified for quantitative measurements. Notably, the detector is a PMT, which has difficulty imaging more than 10-fold ranges of overexpression, and thus, there is concern that the 32-fold differences may be outside of the linear range. Alternatively (and ideally), the users may have made a conversion table from one laser power to another, allowing for a higher dynamic range. (c) The authors need to confirm if there is spectral bleed-through, which could lead to apparent signal overlap between channels. Furthermore, it is not evident in the methods if the imaging is done sequentially or simultaneously, and thus, there may be substantial (if simultaneous) bleed-through from overexpressed Ki67 into the DNA channels (Figure 3). Additionally, sequentially acquired channels can have bleed at high overexpression levels. For both, please specify whether detector linearity and spectral separation were validated under the same imaging conditions. The method section currently does not describe this or the details for acquisition parameters, such as the filters that were used.

We thank the reviewer for raising these important technical points regarding our quantitative microscopy. We have performed additional validation experiments and updated the Methods section accordingly to address each concern.

- (a) Regarding the axial bleed-through, we fully agree that it is critical to distinguish true intranucleolar signals from potential axial bleed-through artefacts. As detailed in our response to Comment #1, we re-examined the 3D imaging data of the entire nucleus. The orthogonal views and Z-axis line profile analysis (Response Figure 1) confirm that these signals are true chromatin invaginations rather than artifacts from peripheral chromatin projecting into the focal plane of 2D sections.
- (b) We thank the reviewer for this suggestion. Our primary purpose of these measurements is to demonstrate a positive correlation with chromatin recruitment and irregular shape rather than to provide absolute quantification. Nevertheless, to assess detector linearity under our imaging conditions, we imaged ATTO488 standards prepared in the same medium and acquired with the identical settings as for cells. Per-concentration means showed a linear relationship with concentration (ordinary least squares, $I = \alpha + \kappa C$ with $R^2 = 0.9997$, $\alpha = -0.00118$, $\kappa = 9.6 \times 10^{-6}$; Response Figure 3A). We then computed the deviation of each dye level from this fit (Response Figure 3B). Concentrations from 0.25 to 25 μM were within $\pm 10\%$ of the fitted line, which we take as the verified linear operating range of the PMT for our settings. The cellular GFP intensities used in our analyses fall predominantly within this verified linear window (Response Figure 3C), suggesting that our intensity measurements reliably reflect relative EGFP-Ki-67 levels without being significantly affected by detector nonlinearity.

Response Figure 3. PMT linearity and operational range under experimental settings.

(A) Intensity–concentration relationship for ATTO488. Different ATTO488 solutions in imaging medium were acquired with the same settings used for the Ki-67 overexpression experiments. Per-concentration mean intensities are plotted against concentration. The dashed line shows the ordinary least-squares (OLS) fit $I = \alpha + \kappa C$ ($R^2=0.9997$, $\alpha = -0.00118$, $\kappa = 9.6 \times 10^{-6}$ intensity· nM^{-1}). The gray shaded band means $\pm 10\%$ around the fit. Points are coloured by whether they fall within (blue) or outside (red) the $\pm 10\%$ criterion.

(B) Percent deviation from the OLS fit vs concentration. Dashed lines mark $\pm 10\%$. Concentrations from 0.25–25 μM fall within $\pm 10\%$, defining the verified detector-linear window under these settings (grey region).

(C) Cell intensity range relative to dye standards. The histogram shows the cellular EGFP-Ki-67 intensities in its overexpression experiments, with vertical dashed lines indicating the dye means for each concentration. The cell distributions are mostly between 0.25 and 25 μM ATTO488, which is within the verified linear window.

(c) To test the presence of spectral bleed-through by Ki-67 overexpression, we performed two key experiments to validate our results:

1. We imaged cells highly overexpressing our EGFP-Ki-67 construct without fluorescence labelling of DNA and NPM1 (Response Figure 4A, Revised Figure S14). Under the exact same acquisition settings used for our experiments, we detected no significant signal in the DNA or NPM1 channels. This directly confirms the absence of spectral bleed-through from the Ki-67 channel.
2. To ensure the phenotype wasn't an artefact of the fluorescent tag or the bleed-through, we additionally used an EGFP-T2A-Ki-67 construct to overexpress Ki-67. This system co-expresses untagged Ki-67 and a separate, soluble EGFP reporter at equivalent levels. Cells with high soluble EGFP levels, indicating high levels of untagged Ki-67, fully recapitulated the nucleolar deformation and the excess chromatin enrichment in the nucleolus (Response Figure 4B, Revised Figure S9D).

Response Figure 4: Ki-67 overexpression directly causes nucleolar irregularity and chromatin enrichment within the nucleolus.

(A) Minimal spectral bleed-through from the overexpressed EGFP-Ki-67. HeLa cells expressing SNAP-NPM1 were transfected with EGFP-Ki-67. Images were acquired with the same settings with or without fluorescence labelling of NPM1 and DNA with SNAP-SiR and SPY555-DNA, respectively.

(B) Aberrant nucleolar morphology and chromatin enrichment is caused by Ki-67 itself. HeLa cells expressing SNAP-NPM1 were transfected with EGFP-T2A-Ki-67. Images were acquired following fluorescence labelling of NPM1 and DNA with SNAP-SiR and SPY555-DNA, respectively.

Together, these controls demonstrate that the observed effects arise from Ki-67 overexpression and are not attributable to imaging artefacts. We will also update the Methods section in the revised manuscript to specify the acquisition parameters, including filter sets, laser powers, and PMT settings, to ensure full transparency and reproducibility.

4. A recent study - Riback et al Mol Cell 2023 - suggests that nucleolar activity (locations of FCs) maintains the non-spherical shape of nucleoli and rules out bulk chromatin mechanics employing lamin and hp1alpha knockdowns. The authors should comment on whether they view their results as reconcilable with this alternative determinant of nucleolar shape.

We thank the reviewer for drawing our attention to the study by Riback et al. (Riback, J. A. et al. Mol. Cell 2023) and will include a discussion of this study in a revised version.

In support of transcription-based shape regulation, Riback et al. showed that nucleoli round up upon transcriptional inhibition. We fully agree that transcriptional activity is an important determinant of nucleolar morphology and we similarly observed nucleolar rounding upon transcriptional stalling, as evidenced by nucleolar cap formation in several of our top candidates involved in ribosome biogenesis (see Revised Figure S1). However, rounding under these conditions could result from multiple factors as nucleoli are drastically reorganized. Notably, under these conditions we observe a loss of Ki-67 from the nucleolar interior, consistent with previous studies (Sobecki, M. et al. Elife 2016; Kill, I. R. J. Cell Sci. 1996), and a concomitant reduction of intranucleolar chromatin, which to our knowledge has not been reported previously (Response Figure 5). This raises the possibility that the rounding upon transcription inhibition is, at least in

Response Figure 5: Transcription inhibition induces loss of Ki-67 and chromatin from the nucleolar interior.

HeLa cells endogenously expressing EGFP-Ki-67 and stably expressing SNAP-NPM1 were stained with SPY555-DNA dye. SNAP-NPM1 was labelled with SNAP-silicon rhodamine (SiR). RNA polymerase I transcription was inhibited by 5 nM actinomycin D (Act D) for 2 h. part, due to the absence

of Ki-67-mediated tethering.

Riback et al. investigated the role of bulk chromatin mechanics using lamin A or HP1 α depletion. These perturbations can induce widespread nuclear alterations, including nuclear shape abnormalities, rupture, and large-scale chromatin reorganization (Larrieu, D. et al. Science 2014; Strom, A. R. et al. Elife 2021), which could indirectly affect nucleolar morphology and make the specific contribution of chromatin more difficult to isolate. In contrast, Ki-67 depletion specifically disrupts the chromatin–nucleolus anchor while having minimal effect on overall nuclear morphology (Response Figure 6A and 6B; Revised Figure S6A, and S6B). Nevertheless, it strongly affects nucleolar circularity (Revised Figure 3B) and chromatin enrichment (Revised Figure 3C and D), suggesting that Ki-67 is required for tethering perinucleolar heterochromatin and generating non-spherical nucleolar features. Taken together, we propose a unified model in which transcription drives nucleolar assembly, while Ki-67-mediated chromatin tethering refines and stabilises its architecture. In this framework, transcription is sufficient to form nucleoli, but Ki-67 is required to deform them away from spherical symmetry, giving rise to invaginations and more complex morphologies. We will clarify this distinction and explicitly reconcile our results with the model proposed by Riback et al. in the revised Discussion.

Response Figure 6: Minimal effects of Ki-67 depletion on the nuclear morphologies.

(A, B) Quantification of nuclear morphologies following Ki-67 depletion. Nuclear area (A) and its circularity (B) were measured using data in Revised Figure 3. Bars indicate median. * $P < 0.05$ with Kruskal–Wallis test followed by Dunn’s test, compared to siControl.

5. The notion of Ki-67 as a pickering agent in contrast to a traditional surfactant is perplexing. Are the authors proposing that Ki-67 is forming a solid-like phase that localizes to the surface? Is the model not that Ki-67 binds chromatin and the nucleolus in contrast to solid-like aggregates of Ki-67, lowering the surface tension on the surface of chromatin? This would explain the FRAP (binding to chromatin strongly and with more specificity during interphase) and fit the truncation data. The authors should reconsider or, at a minimum, explain how chromatin fits in this Pickering agent picture (is chromatin aggregating?). The physical picture is elusive to this reviewer.

We thank the reviewer for highlighting the need to clarify Ki-67’s physical model. We agree that the physical picture must be stated more carefully. Our intention was to convey that Ki-67 exhibits features of a classical surfactant, with distinct domains binding both chromatin and nucleolar components, while also adopting a more persistent, semi-solid-like interfacial layer, with less than 35% FRAP recovery compared to approximately 90% in prometaphase (Saiwaki et al., 2005). This reduced mobility in interphase is likely due to both increased chromatin affinity through dephosphorylation (Hernandez-Armendariz, A. et al. Mol. Cell 2024) and multivalent interactions including RNA binding (Ma, K. et al. Cell Discov. 2022). We, however, agree that the Pickering terminology may be misleading, particularly because chromatin is already a polymeric, viscoelastic network and the interface is not a classical fluid–fluid boundary. At present, there are insufficient data to propose a detailed physical model, as key parameters such as 2D lateral dynamics, nucleolar surface tension, and fusion dynamics of intra-nucleolar chromatin foci remain unknown. We will therefore revise the text to emphasize Ki-67’s amphiphilic character, molecular extension, and dynamic mobility without invoking a Pickering-like model.

6. To summarize, my core concern throughout the paper is the presentation and interpretation of the DNA signals reported within the nucleolar interior. Beyond the potential imaging artifacts discussed above, the current discussion section does not sufficiently address these findings. As written, the discussion focuses on the rim localization of Ki-67 and its possible interface role, while largely overlooking the implications of DNA detected within the GC. Likewise, summary Figure 8 depicts Ki-67-bound chromatin more as invaginations from the nuclear interior rather than actual chromatin localized within the GC (likely a more accurate representation of the data). I encourage the authors to address this discrepancy directly. Importantly, even if the interpretation of DNA within the nucleolar interior were revised or omitted, the main findings of the study would remain valuable and compelling.

We thank the reviewer for allowing us to clarify potential misunderstandings and for the helpful suggestions on how to further improve our manuscript. We fully agree that the DNA signal within the nucleolar interior require careful validation and discussion. To address this, we provide new data demonstrating that these signals correspond to chromatin invaginations into the nucleolar interior (see comment #1), that they are distinct from rDNA and instead co-localize with heterochromatin marks (see comment #2) and are not due to spectral bleed-through, as evidenced by the ability of Ki-67 overexpression to recruit substantial amounts of chromatin into nucleoli (see Comment #3). We will clarify this interpretation throughout the revised manuscript and add a discussion of its implications in the updated version.

Other major and minor concerns discussed Figure by Figure:

1. Figure 1B, Figure 3I - the results currently do not take into account possible differences in nucleolar sizes. Upon depletion of Ki-67 or other target genes from the screen, the nucleoli can change in size significantly, which would shift their aspect ratio closer to 1 as smaller nucleoli are more spherical due to the diffraction limit. The mechanism proposed for Ki-67 would be expected to shift the relationship between nucleolar area and aspect ratio (e.g. aspect ratio of nucleoli at the same average area) It would be important to confirm Ki-67 depletion and overexpression experiment conclusions to determine if Ki-67 changes the number of nucleoli or the surface properties.

We appreciate the reviewer's comment to improve our conclusion. We agree that changes in nucleolar size and its number could potentially confound the aspect ratio measurements, as smaller objects are inherently more spherical.

Response Figure 7: Quantification of nucleolar number in RNAi screen.

Individual data points represent the median percentage of cells with a single nucleolus in siControl wells or under target knockdown conditions. Knockdown of the indicated proteins (magenta) induced nucleolar cap formation (Revised Figure S1). Boxplots show the median, interquartile range, and 1.5× interquartile range. The dashed line indicates 4× SD above the overall median of the control samples.

To assess whether small nucleolar size might have affected aspect ratio measurements in our screen, we quantified the percentage of cells with more than three nucleoli. The resulting plot (Response Figure 7) shows that most siRNA conditions exhibit a decrease in nucleolar number compared to controls, indicating that nucleolar shape can be reliably measured. The few candidates with increased nucleolar number correspond to hits classified as nucleolar cap formation, a condition that induces true nucleolar rounding (Riback et al. 2023; Revised Figure S1). If the rounding phenotype were largely driven by very small nucleoli (e.g., due to diffraction-limited effects), one would expect numerous false positives across the screen. However, visual inspection of the top 20 candidates confirms genuine rounding, and our analysis pipeline does not simply select cells with small nucleoli (Revised Figure 1C), supporting the conclusion that the observed nucleolar rounding reflects a true morphological change rather than a size-dependent artifact. Collectively, these analyses confirm that the screen reliably measures nucleolar aspect ratio.

To address this concern specifically for Ki-67, we have re-analysed our data from both the Ki-67 depletion and overexpression experiments. Our analysis confirms that altering Ki-67 levels has minimal differences in either the total nucleolar area per nucleus or the number of nucleoli between conditions (Response Figure 8; Revised Figure S6C, D). Because the nucleolar size remains largely unchanged regardless of Ki-67 depletion or its overexpression, the alteration of nucleolar shape is not a secondary consequence of size and number changes. This strengthens our conclusion that Ki-67 directly modulates the surface properties and shape of the nucleolus, independent of its size. We have added this

Response Figure 8: Minimal effects of Ki-67 depletion on the nucleolar size and its number.

(A, B) Quantification of nucleolar size and its number following Ki-67 depletion. Nucleolar area per nucleus (A) and nucleolar number per nucleus (B) were quantified using data in Revised Figure 3. Bars indicate median values. * P < 0.05 with Kruskal–Wallis test followed by Dunn’s test, compared to siControl.

important control analysis to the manuscript.

2. Figure 2F - the line scan for chromatin foci (1) is confusing. Ki-67 signal seems to be present asymmetrically on one side of the DNA region. At the same time NPM1 is strongly depleted in that area. The strongly asymmetric Ki-67 profile and strong depletion of NPM1 in that area rather suggest that this is, in fact, an invagination of the nuclear interior seen from a different optical slice. The nucleolar rim profile is, however, clear.

We thank the reviewer for their insightful comment regarding the line profile in Figure 2F. We agree with the interpretation that DNA signal corresponds to a chromatin invagination into the nucleolar body.

To confirm this, we re-examined the 3D confocal stacks. As shown in the orthogonal views (Response Figure 9, Revised Figure S4A), the structure is clearly a deep chromatin invagination. Crucially, Ki-67 is enriched along the entire surface of the invaginated chromatin, confirming its consistent interfacial localisation between chromatin and the nucleolus. We note that Ki-67 is patchy along chromatin surfaces in general, independent of the invagination, which we attribute to its gel-like, viscoelastic behaviour. We have updated Figure S4 to include these clearer orthogonal-view images to illustrate this point effectively.

Response Figure 9: Interfacial localisation of Ki-67 along a chromatin invagination. Orthogonal views at the chromatin focus in Revised Figure 2F. A yellow cross marks a single chromatin focus.

3. Figure 2G - we suggest using fire mode for colour scales, which would much improve the ability to see the difference in signal intensity. Figure 2H - please use microns on the X axis as the number of rings is an unclear measure (but presumably can be converted by the thickness per ring).

We appreciate the reviewer's feedback on the colour scales. In response, we have changed the Figures to use the 'fire' colour map, which improves the visual distinction between signal intensities.

For the concentric ring analysis, it is important to note that the algorithm normalises the ring widths on a per-nucleolus basis, which is required when averaging many nucleoli because nucleolar segmentations vary widely in size. Consequently, the absolute thickness of individual rings cannot be extracted from this normalised data.

4. Figure 2H "Nucleolar Rim". The authors see a clear peak here for Ki-67; however, the rim and interior have identical fold changes in Figure 2D - why are these quantifications in conflict?

We thank the reviewer for the insightful question. The apparent discrepancy arises because the two Figures quantify different aspects of the Ki-67 localisation. Revised Figure 2H shows the local enrichment of Ki-67. The clear peak of Ki-67 between NPM1 and DNA signals in the nucleolar rim plot demonstrates the interfacial localisation of Ki-67. Conversely, Revised Figure 2D displays the average intensity of Ki-67 across the entire nucleolar rim and its interior. The rim was defined to include the DNA signal around the nucleolus (see Revised Figure 2B), meaning that for Ki-67, a substantial portion of nucleoplasm (low signal) is also averaged. Combined with the patchy localisation in both regions, this results in similar average intensities despite the pronounced local enrichment (rings 20-30, Revised Figure 2H).

5. Figure 3F - see point about spectral bleed-through above. It would be useful to repeat this experiment with H2B-mCherry Ki-67-EGFP combination or, alternatively, not label NPM1 and instead use SiR DNA dye with Ki-67-EGFP to ensure there is no channel bleed-through at high intensities. Notably, 3G has a characteristic signature of bleed-through for partitioning measurements (relative ratio = $r_bleed * K_no * [Ki-67\ total] / ([DNA\ in\ nucleoplasm] + r_bleed * [Ki-67\ total])$).

We thank the reviewer for highlighting the potential issue of spectral bleed-through in original Figures 3F, 3G (now Revised Figures 4A, B). To address this concern, we have performed additional control experiments using the same acquisition settings but without fluorescence labelling of NPM1 and DNA (see Response Figure 4A). These experiments together with the additional experiment using the EGFP-T2A-Ki-67 construct (see Response Figure 4B) confirmed that the Ki-67 overexpression directly causes the excess chromatin enrichment in the nucleolus and the deformation of nucleolar morphology.

6. Figure 4D - "No IAA treatment" control cells appear to also have a pronounced decrease in nucleolar aspect ratio over the timecourse of imaging, complicating the interpretation of results. This effect on control cells would suggest stress from improper incubator conditions or phototoxicity. Consider re-doing it with less channels to decrease phototoxicity. Figure S6 does not have the same issue, for example. Also, for Figure 4B-E please specify what kind of error the shading on the plots represents.

We thank the reviewer for pointing this out. We agree that the slight decrease in nucleolar aspect ratio observed in the "No IAA" control cells likely reflects general stress. While we have optimized imaging conditions to minimize these effects (as shown in Revised Figure S11), prolonged imaging in the microscope incubator cannot perfectly maintain temperature and, more importantly, humidity compared to a standard cell culture incubator. Therefore, we attribute the observed baseline stress primarily to the limitations of the microscopy incubation rather than to imaging itself. Notably, control cells expressing wild-type Ki-67 (Revised Figure S11) are consistently less sensitive than EGFP-AID-Ki-67 cells, likely due to minimal basal degradation of Ki-67. Unfortunately, there is not much more we can do to further reduce this inherent stress.

Regarding the plots in Revised Figures 5, we will clarify in the Figure legends that the shading represents the mean \pm SD for 5B and 5D, and the mean \pm SEM for 5C.

7. Figure 5C-E - please consider changing the plots for better visual readability. Using z-score and arranging the plots horizontally may help, as it is extremely hard to estimate the differences from the plots in their current form.

We thank the reviewer for their suggestions to improve the data visualisation. In the revised Figure, we have rotated the plots to a horizontal orientation for better alignment with the corresponding images and have clarified the meaning of the green and grey dashed lines in the Figure legend. While z-scores could standardize the data, we believe that reporting absolute measurements of nucleolar circularity and chromatin enrichment, along with median values, provides a clearer and more biologically intuitive representation.

Referee #2 (Report for Author)

The nucleolus is a biomolecular condensate whose morphology reflects its function. The authors performed a siRNA screen using nucleolar shape as a readout. One of their top hits is Ki-67, a protein well known for its role as a chromosome surfactant and for its connection to the nucleolus (localization at the nucleolar periphery and importance for nucleolar genesis). This includes outstanding work both by the present authors and by others.

In this study, the authors confirm the localization of Ki-67 at the interface between perinucleolar chromatin and the nucleolus. They confirm that Ki-67 depletion results in nucleolar rounding, accompanied by depletion of intranucleolar chromatin. This phenotype is observed both upon long-term depletion (siRNAs) and acute depletion (auxin degran). It is reversible, although the effect on shape was less obvious to see, in my opinion, in short-term experiments.

Conversely, overexpression of Ki-67 leads to recruitment of chromatin into the nucleolus, at the expense of chromatin outside. Interestingly, this does not seem to perturb internal nucleolar organization. The authors also perform domain mapping, extending prior work that had already attributed functions to specific regions of Ki-67. Finally, the authors report that microinjection of a restriction enzyme (to digest nuclear DNA) also causes nucleolar rounding.

Technically, the study is rather fine (I was less convinced by Fig 6). However, a major concern is that a significant fraction of the data presented has already been reported, in some cases nearly ten years ago (e.g. DOI: 10.7554/eLife.13722). Examples include:

- 1) the localization of Ki-67 at the nucleolar-chromatin interface (Results section #2),
- 2) its essential role in nucleologenesis during mitosis (perichromosomal formation; this should be added to the Introduction), and, in fact,
- 3) even the effect of Ki-67 depletion on nucleolar rounding (Results section #1; see Fig. 7A in Sobecki et al., eLife 2016).

The authors themselves have already contributed substantially to defining the surfactant nature of Ki-67, confirming its localization to the nucleolar rim and its role in nucleologenesis (DOI: 10.15252/msb.20209469). There were also connections established previously between surrounding chromatin and the nucleolus (e.g. 10.7554/eLife.47533).

It is felt that, altogether, the novelty element of this submission is not sufficient to meet the threshold of The EMBO Journal. I would recommend publication in a different venue, after simplification of redundant data and/or improved referencing of prior studies.

We thank the reviewer for their careful reading and for providing us with the opportunity to clarify the novelty of our work. We apologise if this was not sufficiently emphasised in the original manuscript. Our study provides several key findings that distinguish it from previous work:

- We performed the first live-cell high-throughput, image-based siRNA screen to systematically identify proteins, including Ki-67, that regulate nucleolar morphology, particularly those responsible for maintaining its characteristic irregular shape (Revised Figure 1).
- While the study by Sobecki et al. was foundational for Ki-67's role in chromatin organisation, it does not document the function we report here as we found no explicit mention or quantification of nucleolar shape changes in their manuscript. Although their Figure 7A shows a single image of nucleolar staining (PES1 immunostaining) upon Ki-67 depletion suggestive of shape changes, it does not include any analysis or discussion of this phenotype.

- While Sobecki et al. previously described Ki-67's localisation to the nucleolar periphery, our work identifies that Ki-67 also localises to the nucleolar interior, surrounding chromatin invaginations (Revised Figure 2; Revised Figure S4A). This observation clearly extends beyond previous observations.
- We demonstrate for the first time that Ki-67 expression levels directly dictate both nucleolar shape and the enrichment of chromatin into the nucleolus (Revised Figure 3–5). Ki-67 depletion leads to nucleolar rounding and reduced chromatin enrichment, whereas its overexpression enhances nucleolar deformation and recruits excess chromatin. This Ki-67-dependent chromatin recruitment to the nucleolar interior represents a novel and intriguing finding. In response to Referee 1, we performed follow-up experiments showing that this chromatin is distinct from rDNA and bears heterochromatic marks (see Response Figure 2; Revised Figure S5).
- Using a series of truncation mutants, we map the critical domains of Ki-67 required for its interfacial localisation and its function in determining nucleolar shape (Revised Figure 6). In addition, we showed that Ki-67 exhibits a specific molecular orientation at the chromatin-nucleolar interface (Revised Figure 7).
- Lastly, we show that depleting our top 20 candidate proteins phenocopies Ki-67 depletion (i.e., nucleolar rounding and reduced chromatin within the nucleolus). Notably, we found that the depletion of most of them causes a concomitant decrease in Ki-67 protein levels, suggesting Ki-67 as a key regulatory factor controlling nucleolar morphology (Revised Figure 8).

Taken together, our study provides a novel mechanistic insight into nucleolar morphology regulation. We demonstrate that Ki-67 anchors chromatin to the nucleolar surface via its dual affinity domains, tightly linking nucleolar shape to genome organisation.

To ensure our novel findings are clear, we have revised the Introduction and Discussion sections to more explicitly place our findings in the context of the existing literature, including the foundational work of Sobecki et al.

Specific comments:

1) In Figure S2, the authors use holotomography to confirm, once more, nucleolar rounding in the absence of Ki-67. However, details on the technique (hardware, setup) are not provided.

If holotomography here is indeed equivalent to digital holographic microscopy (DHM), please reference Zorbas et al. (EMBO Reports 2024, DOI: 10.1038/s44319-024-00134-5), which used exactly the same AI approach (neural networks) for segmentation of the nucleolus.

That work extracted quantitative parameters such as "nucleolar optical volume" from OPL variations, providing a strong proxy for material state. Could similar quantifications be applied here to strengthen the analysis?

We thank the reviewer for their careful attention to our methodology and for the opportunity to clarify the distinction between the technique we employed and the one mentioned in the referenced paper. We apologize for not making this difference clearer in the original manuscript.

The holotomographic microscope (Nanolive) that we used employs a specialised implementation that is substantially different from the digital holographic microscopy (DHM) described in Zorbas, C. et al. Our method, holotomography (HT), is based on quantitative phase imaging but is specifically designed to reconstruct a full 3D refractive index (RI) tomogram of the sample. To achieve this, the system captures

numerous holograms from multiple illumination angles and uses an algorithm to solve the inverse scattering problem. This approach provides a detailed, label-free 3D reconstruction of the cell and its internal organelles. Conversely, the digital holographic microscopy (DHM) technique in the referenced paper generally refers to a method that captures a single hologram to numerically reconstruct a 2D quantitative phase or thickness map.

While using different techniques, we employed a similar, yet independently developed, machine-learning-based segmentation algorithm (Shabanov, A. et al. bioRxiv 2021). We will also cite the work by Zorbas et al. in the revised manuscript to acknowledge its relevance and clarify the methodological distinction.

Moreover, we appreciate the suggestion to extract additional quantitative parameters. To address this, we measured the nucleolar refractive index in wild-type and Ki-67 knockout cells (Response Figure 10). Our analysis reveals minimal differences between the two cell lines, indicating that Ki-67-dependent chromatin enrichment within the nucleolus contributes little to changes in overall nucleolar density.

2) The claim that internal nucleolar organization is not affected by Ki-67 enrichment of intranucleolar chromatin seems counterintuitive. Such enrichment should logically alter nucleolar architecture. Electron microscopy or DHM-based assessment of material state could help clarify this point.

We appreciate the reviewer's suggestion and the opportunity to clarify this point. We agree that the enrichment of chromatin within the nucleolus could disrupt its internal organisation. However, we now clarify in response to referee 1 that intranucleolar chromatin surrounded by Ki-67 does not primarily co-localize with rDNA within the nucleolus (see Response Figure 2A; Revised Figure S5A), but rather with heterochromatic regions that invaginate into the nucleolar interior (see Response Figure 2B–2E; Revised Figure S5B–S5E). Further supporting this, while Ki-67 overexpression accumulates chromatin in the nucleolus, cells still exhibit a multi-layered nucleolar architecture (Response Figure 11). Together, the enrichment of intranucleolar chromatin by Ki-67 reflects these heterochromatin invaginations, which are distinct from the core nucleolar components. This spatial segregation explains the minimal impact on internal nucleolar architecture (Revised Figure S3) and ribosome biogenesis (Sobecki et al., 2017).

Response Figure 11: Multi-layered nucleolar architecture is preserved under Ki-67 overexpression.

HeLa cells endogenously tagged with Halo-UBF and stably expressing SNAP-NPM1 and H2B-mNeongreen were transfected with EBFP2-Ki-67 expression plasmids. SNAP-NPM1 was labelled with SNAP- SiR. Images were acquired on an Olympus iXplore SPIN SR using a UPLSAPO $\times 100/1.35$ NA Silicon oil immersion objective.

Clear examples of redundancies:

-The conclusion of Fig. 1 of the present submission is already in Fig. 7A of Sobecki et al. 2016.
 -The conclusion of Fig. 2 is also in Sobecki et al. 2016, which explicitly stated (p.12): "...During interphase, we found that Ki-67 localised at the cortical periphery of the GC, visualised using PES1. Ki-67 formed a boundary between the perinucleolar heterochromatin (clearly visible as a 'ring' in the DAPI staining) and the GC (Figure 7A, Figure 7-Figure supplement 1)."

-Etc.

We thank the reviewer for this detailed feedback and for highlighting the need for a clearer distinction between our findings and the foundational work of Sobecki et al. While Sobecki et al. made important observations linking Ki-67 to the nucleolus and surrounding chromatin organization, its role in nucleolar shape regulation, chromatin invaginations, and the underlying mechanism has not been described previously.

- Regarding our Revised Figure 1 and Sobecki et al. Fig. 7A: While Sobecki et al. perform Ki-67 depletion and show the nucleolus using PES1 as a marker protein, they do not focus on the nucleolar shape. We found no explicit mention or quantification of nucleolar shape changes in their manuscript. Rather, the study concludes "Ki-67 did not affect the gross structure of the nucleolus, as determined by PES1 staining". The novelty of our Revised Figure 1 lies in demonstrating Ki-67's specific role in shaping nucleolar architecture, while also highlighting additional candidate regulators identified through our unbiased screen.
- Regarding our Revised Figure 2 and the observations from Sobecki et al. 2016: We agree that Ki-67's localization to the nucleolar periphery is not novel, and we explicitly note this in the manuscript: "Several studies have described its localisation to the region surrounding nucleoli, known as the nucleolar periphery or rim" (citing Sobecki et al. and two other studies). The novelty lies in the internal Ki-67 signal, which is difficult to detect by immunostaining. Using live-cell imaging, we also show that Ki-67 localizes to the nucleolar interior, specifically at the interface where chromatin forms deep invaginations (Revised figure S4A). This intranucleolar localization, beyond the outer rim, represents a key novel finding that underpins our mechanistic model.

Referee #3 (Report for Author)

This excellent manuscript begins with the description of a screen for modulators of nucleolar shape. Although novel, at first it seemed only moderately exciting to me. However, one of the hits (Ki-67) revealed an interesting link between nucleolar shape and chromatin, and the authors did a wonderful job connecting the dots by providing substantial new mechanistic insights, including convincing evidence for a role for Ki-67 in tethering chromatin to the nucleolus and thereby affecting nucleolar shape, plus beautiful super-resolution data showing that the long Ki-67 molecules are oriented within the nucleolus-chromatin interface.

Although Ki-67 had previously already been implicated in nucleoli and the positioning of (hetero)chromatin, this study provides important new pieces of the puzzle.

The writing is very clear and nicely compact, the Figures are beautiful and effective, the experimental design is rigorous, and all data are supported by solid statistical analysis.

Altogether this is a very rounded story (pun intended). Although one can think of follow-up experiments to address new questions raised by the interesting findings, I do not think these are necessary for the current manuscript.

However, one concern is that all results are based on a single cell type (HeLa cells) only. I would recommend that a few key experiments are repeated in at least one other cell type. This will be important to verify that the findings are not restricted to a single cell lineage.

=====
Reviewed by Bas van Steensel, review task received: 28 July 2025; completed 17 Aug 2025 (apologies for the delay; I was on vacation). It is my standard policy to sign and date *all* of my manuscript review reports, regardless of my comments and recommendations. All correspondence about this manuscript should go via the editor. PLEASE DO NOT REMOVE THIS NOTE =====

We thank the reviewer for their positive assessment of the writing and experimental quality of our study, as well as for the helpful suggestions to further improve the manuscript. We agree that demonstrating the generality of our findings would significantly strengthen our conclusions.

To address this, we performed Ki-67 depletion and overexpression in human osteosarcoma cell lines (U2-OS) and human immortalized retinal pigment epithelial cells (RPE-1). In both cell lines, the results are highly consistent with our initial findings in HeLa cells. Depletion of Ki-67 by two different siRNAs induces nucleolar rounding and loss of chromatin enrichment within the nucleolus (Response Figure 12). Conversely, overexpression of Ki-67 induced irregular nucleolar shapes and excess chromatin enrichment within the nucleolus (Response Figure 13). We incorporated these data into the revised manuscript (Revised Figure S7 and S10). We believe this addition will substantially strengthen the broad applicability of our proposed mechanism.

Response Figure S12: Ki-67 depletion induces nucleolar rounding and chromatin removal from nucleoli in U2-OS and RPE-1 cells.

(A) Live imaging of U2-OS cells stably expressing EGFP-NPM1 following Ki-67 depletion. Cells were transfected with a non-targeting control siRNA (siControl) or two different Ki-67 siRNAs, followed by confocal imaging 48 h post-transfection. DNA was stained with SPY555-DNA before imaging. A single z-slice is shown.

(B) Quantification of nucleolar roundness. Nucleoli were segmented based on NPM1 signals to measure its circularity. Circularity values of the largest nucleolus per nucleus were shown. Bars indicate median values.

(C, D) Quantification of chromatin enrichment in the nucleolar interior and its rim. Relative DNA signal intensities were calculated as Figure 3C and 3D for the nucleolar interior (D) or the nucleolar rim (E). Bars indicate mean values.

(E) Live imaging of RPE-1 cells stably expressing SNAP-NPM1 following Ki-67 depletion. Cells were transfected with a non-targeting control siRNA (siControl) or two different Ki-67 siRNAs, followed by confocal imaging 48 h post-transfection. SNAP-NPM1 and DNA was stained with SiR-SNAP and SPY555-DNA before imaging. A single z-slice is shown.

(F) Quantification of nucleolar roundness. Nucleoli were segmented based on NPM1 signals to measure its circularity. Circularity values of the largest nucleolus per nucleus were shown. Bars indicate median values.

(G, H) Quantification of chromatin enrichment in the nucleolar interior and its rim. Relative DNA signal intensities were calculated as Figure 3C and 3D for the nucleolar interior (H) or the nucleolar rim (G). Bars indicate mean values.

For (B–D), $n = 122$ nuclei (siControl), 116 nuclei (siKi-67#1), 92 nuclei (siKi-67#2), 2 experiments. **** $P < 0.0001$ and * $P < 0.05$ with Kruskal–Wallis test followed by Dunn’s test, compared to siControl.

For (F–H), $n = 82$ nuclei (siControl), 90 nuclei (siKi-67#1), 88 nuclei (siKi-67#2), 2 experiments. **** $P < 0.0001$ and * $P < 0.05$ with Kruskal–Wallis test followed by Dunn’s test, compared to siControl.

Response Figure S13: Ki-67 overexpression induced irregularly shaped nucleolus and excess loading of chromatin into nucleoli in U2-OS and RPE-1 cells.

(A) Live imaging of Ki-67 overexpressing cells (Ki-67 OE) in U2-OS cells. Cells stably overexpressing EGFP-NPM1 were transfected with an Halo-Ki-67 plasmid. Halo-Ki-67 and DNA was labelled with Halo-JF646 and DNA was stained with SPY555-DNA. Mock refers to cells treated with the transfection reagent only.

(B, C) Ki-67 expression level-dependent chromatin enrichment in the nucleolar interior (B) and its rim (C). Relative DNA signal intensities, calculated as described for the DNA signal intensities in Figure 3D and 3E, are plotted against the total Halo-Ki-67 intensity in the nucleus.

(D) Ki-67 expression level-dependent increase in the irregularity of nucleolar shape. Median circularity of segmented nucleoli from NPM1 signals per nucleus is plotted against the total intensity of Halo-Ki-67 in the nucleus.

(E) Live imaging of Ki-67 overexpressing cells (Ki-67 OE) in RPE-1 cells. Cells stably overexpressing SNAP-NPM1 were transfected with an EGFP-Ki-67 plasmid. DNA was stained with SPY555-DNA. Mock refers to cells treated with the transfection reagent only.

(F, G) Ki-67 expression level-dependent chromatin enrichment in the nucleolar interior (F) and its rim (G). Relative DNA signal intensities, calculated as described for the DNA signal intensities in Figure 3D and 3E, are plotted against the total EGFP-Ki-67 intensity in the nucleus.

(H) Ki-67 expression level-dependent increase in the irregularity of nucleolar shape. Median circularity of segmented nucleoli from NPM1 signals per nucleus is plotted against the total intensity of EGFP-Ki-67 in the nucleus.

For (B–D), n = 38 nuclei (Mock); n = 24 nuclei (Ki-67 OE), 2 experiments.

For (F–H), n = 33 nuclei (Mock); n = 28 nuclei (Ki-67 OE), 3 experiments.

Dr. Sara Cuylen-Häring
European Molecular Biology Laboratory
Cell Biology and Biophysics
Meyerhofstr. 1
Heidelberg
Germany

23rd Feb 2026

Re: EMBOJ-2025-121939R
Ki-67 shapes the nucleolus by anchoring chromatin via its amphiphilic properties

Dear Sara,

Thank you for submitting your revised manuscript to our editorial office. Two of the original referees have now assessed it once more, and I am happy to say were both satisfied with the revisions and responses to the first round of comments. Referee 1 still retains a minor interpretational point, which I feel would be good to consider and accommodate in the discussion section of the final version. Other than that, there are only various remaining editorial issues that I need to ask you for at this point:

- Please adjust the order of the manuscript sections, and also make sure to use the correct section headers: Title page with complete author information, Abstract, Introduction, Results, Discussion, Methods, Data Availability, Acknowledgements, Disclosure and Competing Interests Statement, References, Main Figure Legends, Tables, Expanded Figure Legends.
- Please carefully go through the reference list, which contains several entries lacking full citation information such as volume or page/eLocator numbers.
- For preprint citations: Please change in-text reference format according to "preprint: XX et al, 2025"; and in the reference list, authors-year-title followed by name of the preprint platform plus DOI + preprint label - e.g. "bioRxiv doi: 1234/002.dj123 [PREPRINT]"
- Please rename the Conflict of Interest section into "Disclosure and Competing Interests Statement", in accordance with our updated Guide to Authors (<https://www.embopress.org/competing-interests>)
- As we are switching from a free-text author contribution statement towards a more formal statement based on Contributor Role Taxonomy (CRediT) terms, please remove the present Author Contribution section and instead specify each author's contribution(s) directly in the Author Information page of our submission system during upload of the final manuscript. See <https://casrai.org/credit/> for more information.
- Please move the currently separate AI declaration into the Methods section.
- In the legend for Figure 2, please explicitly state that various panels contain additional analyses of the micrographs presented in panel A, rather than different images.
- Please turn '(Supplementary)Table S1' into an Expanded View Dataset - renaming it into 'Dataset EV1' both in the file and in all in-text call-outs (including those made in the Reagents & Tools table).
- Please refer to our author guide (<https://link.springer.com/journal/44318/submission-guidelines#cms-Expanded-View-data>) regarding 'supplementary' figures, and consider re-organizing the current figures and supplemental figures. The simplest solution shall be to convert Figures S1-S14 into 'Appendix Figures S1-S14' (making sure to use this naming when referring to them in the text), and combining them in a single 'Appendix' PDF, with each figure legend directly below the respective figure; such a combined file should be headed by a title page stating 'Appendix for [article title]' and containing a Table of Contents listing the Appendix Figure numbers and respective pages. Alternatively, you may promote some (5-8) of these figures to 'Expanded View Figures', in which case they should remain uploaded as separate files and with their legends remaining in the main text (these EV Figures would be typeset and directly viewable in the HTML version of the article). If you choose to do so, just make sure to adjust all figure numbering and referencing (for EV figures and Appendix Figures) accordingly.
- Please provide suggestions for a short 'blurb' text prefacing and summing up the conceptual aspect of the study in two sentences (max. 250 characters), followed by 3-5 one-sentence 'bullet points' with brief factual statements of key results of the paper; they will form the basis of an editor-written 'Synopsis' accompanying the online version of the article. Please may also

upload a synopsis image, which can be used as a "visual title" for the synopsis section of your paper, although we might alternatively simply display a downsized version of the current Figure 9 for this purpose. The image should be in JPG format and in the modest dimensions of (exactly) 550 pixels wide and 300-600 pixels high.

- For the provided Figure Source Data, please make sure to upload them as one folder/archive file per MAIN figure. Source Data for Appendix or Expanded View Figures can be uploaded in a single archive file (combining respective subfolders).

- Finally, during routine pre-acceptance checks, our data editors have raised the following queries regarding figures, data, and legends; I would appreciate if you briefly answered to them in the cover letter of your final submission, and made the requested text modifications with changes/additions highlighted via the "Track changes" option, to facilitate our final checking"

1) Please note that the exact p values are not provided in the legends of figures 1F, 3B-D; 8E, F.

2) Please note that the box plots need to be defined in terms of minima, maxima, centre, bounds of box and whiskers, and percentile in the legends of figures 8A-C

3) Please note that information related to n is missing in the legends of figures 2C-E

4) Please note that the scale bar is missing for figure 2B

I am returning the manuscript to you for a final round of minor revision, solely to allow you to make these modifications and upload the revised files. Once we will have received them, we should be ready to swiftly proceed with formal acceptance and production of the manuscript.

With kind regards,

Hartmut

*** PLEASE NOTE: All revised manuscript are subject to initial checks for completeness and adherence to our formatting guidelines. Revisions may be returned to the authors and delayed in their editorial re-evaluation if they fail to comply to the following requirements. As a first step please read our guidelines for revised submissions:
<https://link.springer.com/journal/44318/submission-guidelines#cms-Revised-submissions>

1) Every manuscript requires a Data Availability section (even if only stating that no deposited datasets are included). Primary datasets or computer code produced in the current study have to be deposited in appropriate public repositories prior to resubmission, and reviewer access details provided in case that public access is not yet allowed.

- size of the scale bars that are mandatory for all micrograph panels

- the statistical test used to generate error bars and P-values

- the type error bars (e.g., S.E.M., S.D.)

- the number (n) and nature (biological or technical replicate) of independent experiments underlying each data point

- Figures may not include error bars for experiments with $n < 3$; scatter plots showing individual data points should be used instead.

4) Each main and each Expanded View (EV) figure should be uploaded as individual production-quality files (preferably in .eps, .tif, .jpg formats). For suggestions on figure preparation/layout, please refer to our Figure Preparation Guidelines:
<https://media.springernature.com/original/springer-cms/rest/v1/content/27825798/data/v1>

6) Please complete our Author Checklist, and make sure that information entered into the checklist is also reflected in the manuscript; the checklist will be available to readers as part of the Review Process File.

8) Please note that supplementary information at EMBO Press has been superseded by the 'Expanded View' for inclusion of

additional figures, tables, movies or datasets; with up to five EV Figures being typeset and directly accessible in the HTML version of the article.

9) To facilitate reproducibility and cross-laboratory adoption of methodologies, please structure the Materials & Methods section as outlined in our guide to authors, including a completed Reagents and Tools Table.

10) Digital image enhancement is acceptable practice, as long as it accurately represents the original data and conforms to community standards. If a figure has been subjected to significant electronic manipulation, this must be clearly noted in the figure legend and/or the 'Materials and Methods' section. The editors reserve the right to request original versions of figures and the original images that were used to assemble the figure. Finally, we generally encourage uploading of numerical as well as gel/blot image source data.

In the interest of ensuring the conceptual advance provided by the work, we recommend submitting a revision within 3 months (24th May 2026). Please discuss the revision progress ahead of this time with the editor if you require more time to complete the revisions. Use the link below to submit your revision:

Link Not Available

Referee #1:

The authors have addressed all of my major and minor concerns. Thus, I am now strongly in favor of publication. The manuscript presents strong evidence and a mechanistic picture of the role of Ki67 and invagination of chromatin as a modulator of nucleolar shape.

The only further suggestion I had for the authors is to reduce the impression that Ki67 is the only determinant of nucleolar shape in their discussion. This is with respect to their response to major concern 4 of my critique (and reflected in their discussion). I think the authors might want to lean more toward the idea that the interplay between Ki67-driven chromatin organization and rRNA transcription defines nucleolar shape as the physical model in Riback et al., 2023 does seem to show that FC location is sufficient to predict nucleolar shape across multiple cell lines ($R^2 \sim 0.96$, Figure 3B of Riback et al 2023). I reiterate (for the editor) that this is a suggestion to the authors, as it is a discussion point, and I am strongly in favor of publishing the manuscript.

Referee #3:

The new data for U2-OS and RPE-1 cells are compelling and strengthen the manuscript further. Other additional data (rDNA FISH; further imaging to confirm invaginations; technical checks of imaging and analyses) and clarification of the interpretations - particularly regarding invaginations - are also helpful. The work is of very high quality and provides multiple new and interesting insights. So I recommend publication in EMBO J.

=====
Reviewed by Bas van Steensel, review task received: 23 Jan 2026; completed 29 Jan 2026. It is my standard policy to sign and date *all* of my manuscript review reports, regardless of my comments and recommendations. All correspondence about this manuscript should go via the editor. PLEASE DO NOT REMOVE THIS NOTE =====

The authors have addressed all review and editorial requests.